

# Who's who in *Magelona:* phylogenetic hypotheses under Magelonidae Cunningham & Ramage, 1888 (Annelida: Polychaeta)

Kate Mortimer[1], Kirk Fitzhugh[2], Ana Claudia dos Brasil[3] and Paulo Lana[4]

[1] Natural Sciences, Amgueddfa Cymru–National Museum Wales, Cardiff, Wales, United Kingdom
[2] Natural History Museum of Los Angeles County, Los Angeles, CA, United States of America
[3] Departamento de Biologia Animal, Instituto de Ciências Biológicas e da Saúde, Universidade Federal Rural do Rio de Janeiro, Seropédica, Rio de Janeiro, Brazil
[4] Centro de Estudos do Mar, Universidade Federal do Paraná, Pontal do Sul, Paraná, Brazil

Corresponding author
Kate Mortimer,
Katie.Mortimer@museumwales.ac.uk

## ABSTRACT

Known as shovel head worms, members of Magelonidae comprise a group of polychaetes readily recognised by the uniquely shaped, dorso-ventrally flattened prostomium and paired ventro-laterally inserted papillated palps. The present study is the first published account of inferences of phylogenetic hypotheses within Magelonidae. Members of 72 species of *Magelona* and two species of *Octomagelona* were included, with outgroups including members of one species of Chaetopteridae and four of Spionidae. The phylogenetic inferences were performed to causally account for 176 characters distributed among 79 subjects, and produced 2,417,600 cladograms, each with 404 steps. A formal definition of Magelonidae is provided, represented by a composite phylogenetic hypothesis explaining seven synapomorphies: shovel-shaped prostomium, prostomial ridges, absence of nuchal organs, ventral insertion of palps and their papillation, presence of a burrowing organ, and unique body regionation. *Octomagelona* is synonymised with *Magelona* due to the latter being paraphyletic relative to the former. The consequence is that Magelonidae is monotypic, such that *Magelona* cannot be formally defined as associated with any phylogenetic hypotheses. As such, the latter name is an empirically empty placeholder, but because of the binomial name requirement mandated by the International Code of Zoological Nomenclature, the definition is identical to that of Magelonidae. Several key features for future descriptions are suggested: prostomial dimensions, presence/absence of prostomial horns, morphology of anterior lamellae, presence/absence of specialised chaetae, and lateral abdominal pouches. Additionally, great care must be taken to fully describe and illustrate all thoracic chaetigers in descriptions.

# INTRODUCTION

*Felix qui potuit rerum cognoscere causas* –Virgil, *Georgics, Vol. 1, Books I–II*
The Magelonidae, known commonly as the shovel head worms, gain their name from their uniquely flattened, spade-shaped prostomia used in burrowing through soft sediments. They comprise a relatively small family of marine annelids, with members distributed among 77 species, although many more individuals to which new species hypotheses will refer are likely. Magelonids are generally found at depths of less than 100 m, although members of deeper water species at depths of 1,000–4,000 m have been recorded (*Hartman, 1971*; *Aguirrezabalaga, Ceberio & Fiege, 2001*). They predominately burrow through sands and muds, although members of several tubicolous species are known (*Mills & Mortimer, 2019*; *Mortimer, 2019*). Individuals feed using two slender palps (*Mortimer & Mackie, 2014*), which are unique amongst polychaetes in being papillated and ventrally inserted.

The unusual morphology of magelonids has often led to difficulties in relating them to other annelid groups. *Johnston (1865)*, puzzled by their peculiar external form, placed them at the end of his catalogue, under the family Maeadae, based on the new genus and species *Maea mirabilis*. Later, *McIntosh (1877)* included magelonids within the Spionidae, after synonymising *Maea* (*Johnston, 1865*) with *Magelona* Müller, 1858. Whilst *Cunningham & Ramage (1888)* erected the family Magelonidae stating "We have formed a special family for it, as it cannot be admitted into the Spionidae, with which it is most nearly allied, or into any other family." McIntosh, who published extensively on magelonid morphology (*McIntosh, 1877*; *McIntosh, 1878*; *McIntosh, 1879*; *McIntosh, 1911*), suggested similarities with spioniforms such as members of *Prionospio* Malmgren, 1867, and *Heterospio* Ehlers, 1874, but also with the chaetopterid *Spiochaetopterus* Sars, 1853. He additionally noted that the way the "proboscis" (cf. **Character descriptions** below regarding terminology) operated and the structure of the "snout and circulatory organs" are features *sui generis*. However, the placement of magelonids with spioniform polychaetes has continued until semi-recently. The latest studies have proposed placement alongside the Chaetopteridae, Sipuncula, and Oweniidae (*Struck et al., 2015*; *Weigert et al., 2014*; *Helm et al., 2018*), or as sister taxon of all other annelids along with the Oweniidae (Palaeoannelida) (*Weigert & Bleidorn, 2016*). These phylogenetic considerations will be discussed more fully later in this paper.

Whilst the systematic position of the family may have received some attention, phylogenetic relationships within have received far less. Perhaps the relatively uniform bodies of magelonids have been a contributing factor. Currently the family contains two genera: the type genus *Magelona* F. Müller, 1858, and the monotypic *Octomagelona* (*Aguirrezabalaga, Ceberio & Fiege, 2001*), differentiated by the number of thoracic chaetigers. All previously introduced generic names for the group have been synonymised (*Maea Johnston, 1865*), *Rhynophylla* Carrington, 1865, *Meredithia Hernández-Alcántara & Solís-Weiss, 2000*). *Brasil (2003)* inferred phylogenetic hypotheses among members of the family based on external morphological characters, confirming the monophyletic status of the group. Her analysis concluded that both *Octomagelona* and *Meredithia* were paraphyletic, but no further conclusions could be made. The present study aims to expand the unpublished analysis performed by *Brasil (2003)* to include additional taxa and characters.
## MATERIAL AND METHODS

### Methodological considerations

What is and is not contained in the present study is based on several interrelated principles that initially might not appear consistent with some of today's thinking regarding biological systematics. These principles are not of our making, but rather are firmly established perspectives within philosophy of science, and familiar to many fields of science beyond systematics. The last several decades have witnessed systematics engaging in a growing tendency toward developing insular views that are at odds with basic tenets of logic, reasoning, and scientific inquiry. For these reasons, we regard it as imperative to present justifications for what is and is not included in this study from a methodological perspective. The disinterested reader can ignore this section without damage, but the information provided does offer a useful summary for those more inclined toward a much-needed critical thinking in systematics.

#### *The goal of scientific inquiry*

As systematics is a field of science and subfield of evolutionary biology, it should cohere with the acknowledged goal of scientific inquiry. That goal is to not only describe phenomena but also pursue causal understanding of what is encountered (*Hanson, 1958*; *Hempel, 1965*; *Rescher, 1970*; *Popper, 1983*; *Popper, 1992*; *Salmon, 1984b*; *VanFraassen, 1990*; *Strahler, 1992*; *Mahner & Bunge, 1997*; *Hausman, 1998*; *Thagard, 2004*; *Nola & Sankey, 2007*; *Regt, Leonelli & Eigner, 2009*; *Hoyningen-Huene, 2013*; *Potochnik, 2017*; *Potochnik, 2020*; *Anjum & Mumford, 2018*; *Currie, 2018*). In the context of systematics that pursuit includes describing the characteristics of organisms, in the form of differentially shared characters, as well as exploring possible past causal events that account for those characters, either as matters of proximate and/or ultimate causes *sensu Mayr (1961)*, *Mayr (1993)*, (*Fitzhugh, 2012* and *Fitzhugh, 2016a*). As aptly expressed by *Uller & Laland* (*2019*: 1), "Scientific inference typically relies on establishing causation. This is also the case in evolutionary biology, a discipline charged with providing historical accounts of the properties of living things, as well as an understanding of the processes that explain the origin of those properties."

Whilst it is fashionable to speak of "*the* phylogeny" of a group of organisms, this is something of a misnomer. Inferring phylogenetic hypotheses from a set of observed character data only achieves the objective of explaining those data, with implicit acknowledgement that no phylogenetic inference can exhaustively account for all potentially observable characters of organisms. Phrases of the form "*the* phylogeny of $X$" incorrectly connote that a final solution is being offered, when in fact such would be impossible. Emphasising the term phylogeny has the undesirable consequence of suggesting that this is the objective of systematics, thus overshadowing the pursuit of causes, which is the real aim of scientific inquiry. The same critique applies to the commonly used phrase "molecular phylogeny." There can be no "phylogeny" of molecules, just as there can be no separate phylogenies

for any arbitrary subdivisions of classes of characters that are in need of being explained.

The causal objective in systematics is the arena within which taxa are considered. Taxa are best regarded not as either class constructs, mere groupings, things, entities, or ontological individuals, but rather as the variety of explanatory hypotheses routinely and purposely inferred. These hypotheses include species and phylogenetic hypotheses, albeit there are other classes of explanatory hypotheses that in their own right deserve to be called taxa (Fig. 1; cf. *Hennig, 1966*: fig. 6) given that they as well are inferred causal accounts (*Fitzhugh, 2012*; *Fitzhugh, 2016a*; see also *Fitzhugh, 2005b*; *Fitzhugh, 2006a*; *Fitzhugh, 2008a*; *Fitzhugh, 2009*; *Fitzhugh, 2013*; *Fitzhugh, 2015*). Consider for instance the phylogenetic and specific "relations" shown in Fig. 1. Each clearly illustrates that individuals along the top of the diagram are what exist in the present, and lines connecting to lower individuals indicate past tokogenetic events. This diagram also implies past, albeit quite vague, causes such as novel character origin and fixation events among individuals in populations, as well as population splitting events. To say Fig. 1 shows several "lineages" is just an imprecise way of referring to the totality of past causal events that account for what are observed of the organisms in the present. A further consequence is that taxa are neither described, discovered, nor delimited; they are inferred as reactions to what we observe of the properties of organisms. What *are* described are individuals at specific moments in their life history (*Fitzhugh, 2012*; *Fitzhugh, 2016a*, see also *Fitzhugh, 2008a*; *Fitzhugh, 2009*; *Fitzhugh, 2013*; *Fitzhugh, 2015*; operational examples include (*Fitzhugh, 2010b*; *Nogueira, Fitzhugh & Rossi, 2010*; *Nogueira, Fitzhugh & Hutchings, 2013*; *Nogueira et al., 2017*; *Nogueira et al., 2018*; *Fitzhugh et al., 2015*); what *Hennig (1966)* termed semaphoronts (Fig. 1; cf. *Hennig, 1966*: fig. 6). The interplay between the descriptive and the explanatory will become apparent later (cf. **Phylogenetic inference = abduction**).

The present study seeks to infer taxa in the form of phylogenetic hypotheses, which are subsumed under the more inclusive (composite) phylogenetic hypothesis formally called Magelonidae. We acknowledge that species hypotheses have been previously and separately inferred and are not the focus of this paper. Whilst this distinction between phylogenetic and specific hypotheses has been the typical and inferentially appropriate approach in systematics, exceptions can be found among some publications (*e.g.*, *Nygren et al., 2018*; *Shimabukuro et al., 2019*; *Radashevsky et al., 2020*), in which species hypotheses are simultaneously inferred with phylogenetic hypotheses that only causally account for sequence data, after which morphological characters are incorrectly introduced in a post hoc manner. Notwithstanding the fact that the requirement of total evidence (RTE; *Fitzhugh, 2006b*; see **Sequence data and explanatory hypotheses**, below) is violated, such inferences have questionable merits for the fact that explaining shared nucleotides or amino acids requires, at a minimum, discriminating between causes such as genetic drift, and selection *via* downward causation (see **Sequence data and explanatory hypotheses**, below). The nature of phylogenetics algorithms is such that they are agnostic as to detailed causal parameters, requiring that one must give specific consideration to drift or selection, which can determine whether or not sequence data can be directly explained *via* phylogenetic inferences. This is an issue that has been largely ignored in discussions

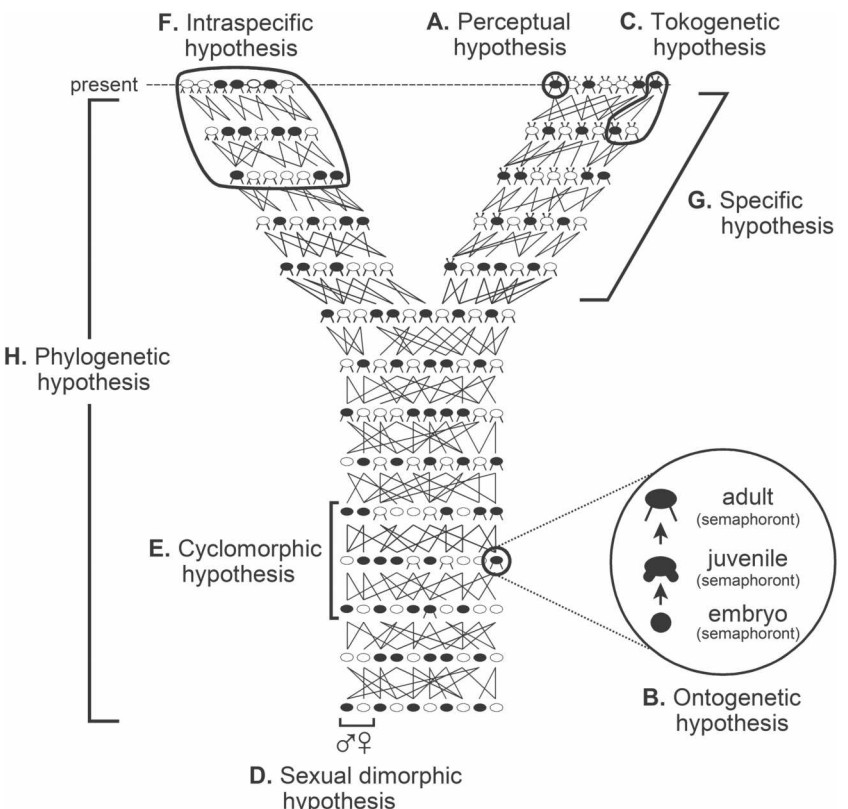

**Figure 1** Classes of hypotheses in systematics. Relations between the eight classes (A–H) of explanatory hypotheses commonly inferred in biological systematics (adapted from *Fitzhugh, 2013*: fig. 1, *Fitzhugh, 2016b*: fig. 1; based on *Hennig, 1966*: fig. 6).

of phylogenetic inference (*Fitzhugh, 2016b*). Once again, the objective of inferring these different classes of hypotheses is to pursue causal understanding of the properties of organisms.

## Characters and observation statements

Scientific inquiry is in reaction to the objects and events we perceive in the universe. Conceptualising and then communicating our perceptions require accurate conveyance of observation statements. Systematics has had a long tradition of speaking of organismal observations in terms of characters and states (*Sokal & Sneath, 1963*; *Sneath & Sokal, 1973*). *Fitzhugh (2006c)* remarked on the fact that the character/state distinction as well as other, similar perspectives in systematics are not accurate representations of observation statements, especially compared to the established views in epistemology, where observation statements are presented as relations between subjects and predicates (*Strawson, 1971*; *Alston, 1993*; *Audi, 1998*). Consider for instance, the statement, "These chaetae are bidentate." The terms chaetae and bidentate do not refer to character and state, respectively. That one can observe objects they call chaetae is because of the properties or characters of those objects. Instead of character and state, the observation statement refers

to the *subject* chaetae, and the *predicate* bidentate is applied to that subject, reflecting a particular property or character perceived of the object as subject (*Hanson, 1958*; *Mahner & Bunge, 1997*; *Strawson, 1971*; *Gracia, 1988*; *Armstrong, 1997*).

Whilst observation statements of organisms are often of intrinsic properties, there also are relational or extrinsic characters (*Findlay, 1936*; *Sider, 1996*; *Armstrong, 1997*; *Francescotti, 2014*; *Allen, 2016*). A common example in systematics involves sequence data, *e.g.*, "Individuals to which species hypothesis *X-us x-us* refer have [subject] nucleotide G at [predicate] position 546." Being in a particular position is relative to other objects. The character/state distinction ignores this relational aspect, leading to nonsensical characterisations, such as, "…if nucleotide A is observed to occur at position 139 in a sequence, 'position 139' is the character [sic] and 'A' is the state assigned to that character" (*Swofford et al., 1996*: 412). Obviously, a position cannot be a character of an object since the position is dependent upon other objects.

The traditional emphasis on "character coding" leading to compilations of observations in the form of the data matrix has treated columns as "characters" and each cell as a "state." *Fitzhugh (2006c*: fig. 1) pointed out, however, that if observations are to be accurately implied by entries in a matrix then each cell represents a complete observation statement of subject-predicate relations, such that each column denotes each subject (Fig. 2A). This also accommodates relational characters (*Fitzhugh, 2016b*). But with the emphasis on causal inquiry in systematics, a data matrix is not only a codified representation of observation statements. The matrix must also imply the why-questions that inquiry seeks to answer in the form of explanatory hypotheses known as taxa. This necessitates knowing the formal structure of those questions and how they, like observation statements, are implied by the matrix. This will be addressed next.

### Basis for phylogenetic inference: why-questions

The inferences of explanatory hypotheses vis-à-vis taxa do not appear *ex nihilo*. As a significant role of scientific inquiry is the pursuit of causal understanding, there is a vital conceptual link between our observations and explanatory hypotheses. That link exists through the implicit or explicit why-questions that are prompted by observations, such that the hypotheses we infer are intended as answers to those questions. The formal structure of why-questions has relevance to systematics for the fact that those questions are implicitly present in a character data matrix provided to a computer algorithm that infers phylogenetic hypotheses (*Fitzhugh, 2006c*; *Fitzhugh, 2016b*).

Whilst we tend to think of why-questions as having the form, "Why (is) *q* (the case)?," such a structure is an incomplete representation. The form of why-questions is instead what is often referred to as contrastive, "Why (is) *q* (the case) in contrast to *p*?" (*Salmon, 1984b*; *Sober, 1984*; *Sober, 1986*; *Sober, 1994*; *Salmon, 1989*; *VanFraassen, 1990*; *Lipton, 2004*; *Fitzhugh, 2006a*; *Fitzhugh, 2006b*; *Fitzhugh, 2006c*; *Fitzhugh, 2016b*; *Lavelle, Botterill & Lock, 2013*). A contrastive why-question contains observation statement

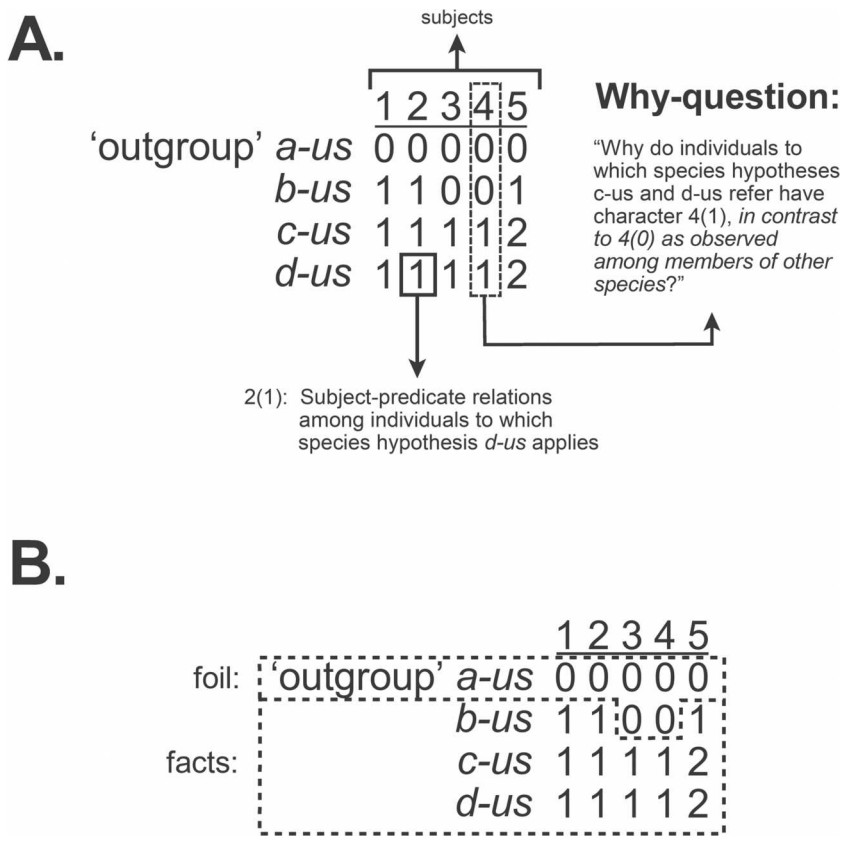

**Figure 2 Observation statements, why-questions, and data matrix.** A data matrix used to infer phyloge-netic hypotheses implies the why-questions to which inferences lead to answers as phylogenetic hypothe-ses, implied by cladograms (Fig. 1). (A) Observation statements, as subject-predicate relations, and con-trastive why-questions implied by a data matrix. (B) The contrastive form of why-questions is maintained in a data matrix through the inclusion of outgroups, distinguishing foil and fact. Modified from *Fitzhugh (2016b*: fig. 3).

*q*, the *fact* to be explained, and *p*, the *foil*, which is usually a condition ordinarily expected or previously explained. Asking a why-question often occurs because we are faced with a situation that is unexpected or surprising (the *fact*), and it is that element of surprise, against what is expected (the *foil*), that leads to the desire for an explanation. The fact/foil distinction associated with implied why-questions can be found in the data matrices used to infer phylogenetic hypotheses, when "outgroup" (= *foil*) and "ingroup" (= *fact*) taxa are designated (*Fitzhugh, 2006c*; *Fitzhugh, 2016b*) (Fig. 2B).

It is often acknowledged (*Bromberger, 1966*; *Sober, 1986*; *Sober, 1988*; *Barnes, 1994*; *Marwick, 1999*; *Sintonen, 2004*; *Schurz, 2005*) that why-questions include the presupposition that fact and foil statements must be true, otherwise there is no basis for asking such questions. This also follows from the common-sense notion that we assume the truth of our observation statements; otherwise, scientific inquiry would not be feasible. This assumption has special implications for explaining sequence data

when rates of substitution are involved, which will be addressed later (see **Sequence data and explanatory hypotheses**). Explaining a fact entails also explaining the foil, and while those explanations are due to separate causal events, they should be due to the same type of cause (*Sober, 1986*; *Barnes, 1994*; see also *Cleland, 2001*; *Cleland, 2002*; *Cleland, 2009*; *Cleland, 2011*; *Cleland, 2013*; *Tucker, 2004*; *Tucker, 2011*; *Turner, 2007*; *Jeffares, 2008*). Designations of outgroup and ingroup accommodate this requirement (*Fitzhugh, 2006c*).

## Forms of reasoning

In speaking of inferring phylogenetic hypotheses, or any other class of taxon for that matter, it is necessary to acknowledge the type of reasoning used to produce those hypotheses as well as the reasoning available for empirically testing them. This provides a clear basis for establishing which, if any, of the currently available phylogenetic inference procedures are logically and scientifically sound.

Three forms of reasoning are often recognised: abduction, deduction, and induction. Among formal treatments of logic, abduction, if acknowledged at all, is typically subsumed under induction (*e.g.*, *Kneale & Kneale, 1964*; *Salmon, 1967*; *Salmon, 1984a*; *Copi & Cohen, 1998*), since the purview of logic is to identify and discriminate logically valid (deductive: true premises give a true conclusion) from invalid or fallacious (inductive: true premises do not guarantee true conclusions) arguments. For purposes of explicating the processes of scientific inquiry, however, it is appropriate to regard abduction as distinct from induction. Operationally each form of reasoning plays a respective role in different stages of inquiry, where those stages include at a minimum the inferences of theories and hypotheses, and the subsequent empirical testing of those propositions. The stage of inquiry most associated with systematics is abduction (*Peirce, 1878*; *Peirce, 1931*; *Peirce, 1932*; *Peirce, 1933a*; *Peirce, 1933b*; *Peirce, 1934*; *Peirce, 1935*; *Peirce, 1958a*; *Peirce, 1958b*; *Hanson, 1958*; *Achinstein, 1970*; *Fann, 1970*; *Reilly, 1970*; *Curd, 1980*; *Nickles, 1980*; *Thagard, 1988*; *Josephson & Josephson, 1994*; *Baker, 1996*; *Hacking, 2001*; *Magnani, 2001*; *Magnani, 2009*; *Magnani, 2017*; *Psillos, 2002*; *Psillos, 2007*; *Psillos, 2011*; *Godfrey-Smith, 2003*; *Norton, 2003*; *Walton, 2004*; *Gabbay & Woods, 2005*; *Aliseda, 2006*; *Schurz, 2008*; *Park, 2017*), *i.e.,* the inferences of hypotheses as the means of causally accounting for the various differentially shared characters observed among organisms (*Fitzhugh, 2005a*; *Fitzhugh, 2005b*; *Fitzhugh, 2006a*; *Fitzhugh, 2006b*; *Fitzhugh, 2008a*; *Fitzhugh, 2008b*; *Fitzhugh, 2008c*; *Fitzhugh, 2009*; *Fitzhugh, 2010a*; *Fitzhugh, 2013*; *Fitzhugh, 2014*; *Fitzhugh, 2015*; *Fitzhugh, 2016a*; *Fitzhugh, 2016b*; *Fitzhugh, 2016c*; *Fitzhugh, 2016d*; *Fitzhugh, 2021*). Subsequent empirical assessments of explanatory hypotheses rely on deducing predictions of expected consequences—potential test evidence—if the causal claims in hypotheses are true. The act of testing, which involves determining manifestations of test evidence, is inductive. The operational relations between ab-, de-, and induction received extensive attention in the 19th and early 20th centuries by the polymath, Charles Sanders Peirce (1835–1914) (*Peirce, 1931*; *Peirce, 1932*; *Peirce, 1933a*; *Peirce, 1933b*; *Peirce, 1934*; *Peirce, 1935*; *Peirce, 1958a*; *Peirce, 1958b*), but full appreciation of Peirce's ideas did not become realised until

the second half of the 20th century (*Hanson, 1958*; *Fann, 1970*; *Reilly, 1970*; *Thagard, 1988*; *Josephson & Josephson, 1994*; *Magnani, 2001*; *Magnani, 2009*; *Magnani, 2017*; *Psillos, 2002*; *Psillos, 2011*; *Walton, 2004*; *Gabbay & Woods, 2005*; *Aliseda, 2006*; *Schurz, 2008*; *Park, 2017*). As noted by *Peirce (1932*: 2.106),

> Abduction[…]is merely preparatory. It is the first step of scientific reasoning, as induction is the concluding step[…] Abduction makes its start from the facts, without, at the outset, having any particular [hypothesis] in view, though it is motivated by the feeling that a [hypothesis] is needed to explain the surprising facts. Induction makes its start from a hypothesis which seems to recommend itself, without at the outset having any particular facts in view, though it feels the need of facts to support the [hypothesis]. Abduction seeks a [hypothesis]. Induction seeks for facts.

The "facts" Peirce refers to in relation to abduction are the various characteristics of organisms in need of explanation, whereas the "facts" associated with induction are test evidence. Discussions of relations between these stages of inquiry in systematics can be found in *Fitzhugh (2005a)*, *Fitzhugh (2006a)*, *Fitzhugh (2006b)*, *Fitzhugh (2008b)*, *Fitzhugh (2010a)*, *Fitzhugh (2012)*, *Fitzhugh (2013)*, *Fitzhugh (2016a)*, *Fitzhugh (2016c)* and *Fitzhugh (2016d)*.

Whilst systematics in recent decades, especially beginning with the school of thought called cladistics, has laid great emphasis on the testing of hypotheses, usually at the expense of conflating testing with hypothesis inference (reviewed by *Fitzhugh, 2016c*; *Fitzhugh, 2016d*). Actual testing by way of induction of the variety of hypotheses implied by cladograms is rarely performed. Instead, the tendency is to merely replace results of previous abductions with new abductions as new observed effects become available, which is neither an act of testing nor a basis for claiming that previous hypotheses have been "defeated" or overturned. Associated with confusing hypothesis inference with testing has been the misconception that character data used to infer hypotheses also provide evidential support for those hypotheses, leading to false claims that phylogenetic hypotheses are "robust" or "strongly supported" by the characters used to infer those hypotheses. We will address this misunderstanding later (see ***Explanatory hypotheses and the myth of evidential support***).

In the context of phylogenetic hypotheses, examples of each of the stages of inquiry can be represented by the following abbreviated forms, where the premises lie above the double (non-deductive) or single (deductive) line, and the conclusion(s) allowed by those premises below the line(s):

(1) **Abduction– inferring hypotheses as answers to why-questions:**
- background knowledge, $b$
- theory(ies) $t$, such as "common ancestry"
- observed effects, as differentially shared characters, $e$

---

- explanatory hypotheses, *e.g.*, cladograms, $h$

(2) **Deduction– predictions of consequences given the truth of hypotheses:**
- background knowledge, $b$
- theory(ies) $t$ relevant to the observed effects
- specific causal conditions presented in explanatory hypothesis *via* (1)
- proposed conditions needed to perform test

---

- observed effects $e$, originally prompting $h$ [cf. (1)]
- *predicted test evidence, i.e., effects related as closely as possible with the specific causal conditions of the hypothesis*

(3) **Induction– hypothesis testing:**
- background knowledge, $b$
- theory(ies) relevant to observed effects, $t$
- test conditions performed
- confirming/disconfirming evidence, $e_2$ [observations of predicted test evidence in (2), or alternative observations]

---

- $h$ is confirmed/disconfirmed.

**Table 1  Forms of questions leading to ab-, de-, and inductive inferences, and respective answers provided by those inferences.**

| Question | Inferential Reaction | Conclusion (= answer) |
|---|---|---|
| 'Why is $Y$ the case in contrast to $X$?' | Abduction, cf. (1) | Explanatory hypothesis |
| 'What consequence(s) are expected if hypothesis $h$ is true?' | Deduction, cf. (2) | Potential test evidence |
| 'Should hypothesis $h$ be accepted as true?' | Induction, cf. (3) | Hypothesis confirmation / disconfirmation |

A useful way to think of these inferences is as reactions to particular questions. The form of each question determines what type of inference is used to produce a conclusion that serves as an answer. Respective questions and inferences are shown in Table 1.

Several points of logic need to be highlighted regarding the three types of reasoning, as they are relevant to inferences of phylogenetic hypotheses (in fact all taxa, for that matter). For any inference to be deemed *factually correct*, premises must be assumed to be true. A deduction is *valid* only if true premises guarantee a true conclusion, which is established by particular rules of logic that can be found in standard logic textbooks. Because of those rules, the content of a deductive conclusion is already present in the premises, such that information in the conclusion cannot go beyond what is already stated in the premises (*Salmon, 1984a*). Such reasoning is said to be non-ampliative. By its very nature, deduction cannot introduce new ideas. All non-deductive reasoning is ampliative, thus does not satisfy the criterion of logical validity. This makes abduction a fundamentally important mode of reasoning in everyday life as well as all of science. Abduction is the only type of reasoning from which new ideas are conceived. With true premises, abductive and inductive

reasoning do not guarantee true conclusions. The content of those conclusions will extend beyond what is provided in the premises. Abductive conclusions can be regarded as merely plausible: "By plausibility, I mean the degree to which a theory [or hypothesis] ought to recommend itself to our belief independently of any kind of [test] evidence other than our instinct urging us to regard it favorably" (*Peirce, 1958b*: 8.223). Inductive conclusions, on the other hand, are probable given that they are determined by available test evidence.

### Phylogenetic inference = abduction

Representing phylogenetic inference in a more complete form compared to what is shown in (**1**), the following example of abduction is somewhat closer to the actual abductive structure leading to phylogenetic hypotheses. The basis for this abduction would be to answer the why-question, "Why do semaphoronts to which specific hypotheses *x-us* and *y-us* refer have ventrolateral margins with appendages in contrast to smooth as seen among individuals to which other species hypotheses (*a-us*, *b-us*, etc.) refer?" (*Fitzhugh, 2006c*; *Fitzhugh, 2012*; *Fitzhugh, 2013*; *Fitzhugh, 2015*; *Fitzhugh, 2016a*; *Fitzhugh, 2016b*):

(4)   • *Background knowledge, b*
  • *Phylogenetic theory*: If character $x(0)$ exists among individuals of a reproductively isolated, gonochoristic or cross-fertilising hermaphroditic population and character $x(1)$
  originates by mechanisms $a, b, c \ldots n$, and becomes fixed within the population by mechanisms $d, e, f \ldots n$ ($=$ ancestral species hypothesis), followed by event(s) $g, h, i \ldots n$, wherein the population is divided into two or more reproductively isolated populations, then individuals to which descendant species hypotheses refer would exhibit $x(1)$.
  • *Observations (effects)*: Individuals to which specific hypotheses *x-us* and *y-us* refer have ventrolateral margins with appendages in contrast to smooth as seen among individuals
  to which other species hypotheses (*a-us*, *b-us*, etc.) refer.

  ────────────────────────────────────────

  • *Causal conditions (phylogenetic hypothesis X-us)*: Ventrolateral margin appendages originated by some unspecified mechanism(s) within a reproductively isolated population with smooth ventrolateral margins, and the appendage condition became fixed
  in the population by some unspecified mechanism(s) ($=$ ancestral species hypothesis), followed by an unspecified event(s) that resulted in two or more reproductively isolated populations.

*Background knowledge b*, as auxiliary theories and hypotheses, is generally not listed as a premise, but instead is established knowledge accepted as true that is needed for the other premises in an inference. Background knowledge is specified here for reference later when considering abduction in relation to sequence data. The major premise, *Phylogenetic theory*, is intentionally vague regarding causal specifics. This reflects the near-complete absence of causal information that goes into the development of phylogenetic algorithms as well as the lack of explanatory content in phylogenetic hypotheses implied by cladograms (*Fitzhugh, 2012*; *Fitzhugh, 2013*; *Fitzhugh, 2016b*). The theory as presented here is intentionally limited to gonochoristic or cross-fertilising hermaphroditic organisms (*Fitzhugh, 2013*)

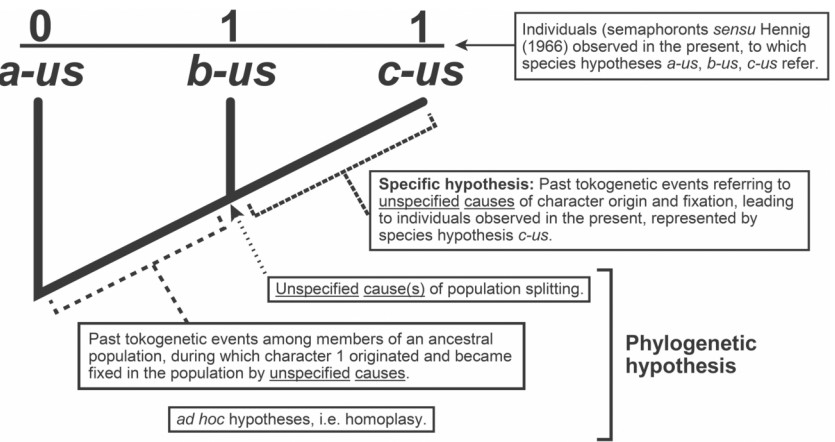

**Figure 3** **Explanatory hypotheses implied by cladograms.** Cladograms imply two general classes of explanatory hypotheses (see also Fig. 1)- specific and phylogenetic- causally accounting for differentially shared characters. Inferences of specific hypotheses are distinct from phylogenetic inferences (cf. *Fitzhugh, 2005b*; *Fitzhugh, 2009*; *Fitzhugh, 2013*; *Fitzhugh, 2015*), but both typically offer vague explanations with largely unspecified causes. Modified from *Fitzhugh (2016b*: fig. 2).

since population-splitting events ("speciation") are not relevant to asexually reproducing, obligate parthenogenetic, or self-fertilising hermaphroditic organisms. The theory states a common cause, which is consistent with the necessary presupposition that observation statements are true, mentioned earlier; if we assume the truth of our observations, then we should pursue explanations of those observations in a manner that maximises the truth element as much as possible in those explanatory accounts. The minor premise in this example, *Observations (effects)*, presents only one set of observation statements, as subject-predicate relations, *i.e.,* ventrolateral body margins. This premise can be expanded to include other relevant characters to be explained, as is typical of the standard character data matrix. A data matrix implies both why-questions as well as the totality of observation statements comprising this second premise (*Fitzhugh, 2006c*; *Fitzhugh, 2016b*). The conclusion, *Causal conditions*, is the result of applying the *Phylogenetic theory* to *Observations* of effects. Such conclusions, often represented by cladograms, serve as answers to the why-questions. The limited causal details provided in the conclusion is a consequence of what little is offered in the theory in terms of causal particulars. For this reason, cladograms are largely devoid of explanatory information. As part of the ampliative nature of abduction, the premises can lead to a single conclusion or multiple, mutually exclusive conclusions. Any valid deduction, on the other hand, being non-ampliative, will only produce a single conclusion.

Whilst we often refer to a cladogram as a phylogenetic "hypothesis," they are in fact composite constructs, implying a minimum of three classes of hypotheses: previously inferred species hypotheses, hypotheses of character origin/fixation among members of ancestral populations, and population splitting events ("speciation") (Fig. 3). A fourth, ad hoc class of hypothesis is typically invoked, called homoplasy (*Lankester, 1870*).

To manifest abductive inferences in a phylogenetics computer algorithm that emulate what is presented in (**4**), the inferential form would be akin to the following (cf. *Josephson & Josephson, 1994*),

(5)  •*d* is a collection of (presumed true) observation statements of differentially-shared characters
•hypotheses, implied by cladograms $X_1, X_2 \ldots X_n$, are possible composite explanations of *d*
•select subset of cladograms with score $S_n$ (minimal "tree length" = minimise ad hoc hypotheses of homoplasy)

────────────────────────────

•cladograms with score $S_n$ are most plausible, mutually exclusive explanations of *d*.

The algorithm functions in a manner that produces conclusions *as if* the common cause *Phylogenetic theory* in (**4**) were operative. Notice as well that the first premise in (**5**) contains the assumption that observation statements regarding shared characters are true; the necessary presupposition of truth associated with implied why-questions in data matrices, if one accepts that the intent of these inferences is to explain observations. The phrase in the conclusion, "cladograms with score $S_n$ are most plausible," is intentional. Plausibility indicates hypotheses are worthy of further consideration:

> By Plausible, I mean a theory [or hypothesis] that has not yet been subjected to any test, although more or less surprising phenomena have occurred which it would explain if it were true, is in itself of such a character as to recommend it for further examination or, if it be *highly* plausible, justify us in seriously inclining toward belief in it, as long as the phenomena be inexplicable otherwise (*Peirce, 1932*: 2.662, emphasis original; *Kapitan, 1997*: 481).

Epistemically, a plausible hypothesis is far weaker than a hypothesis that has been successfully tested with the introduction of test evidence, *i.e.,* (**2**), (**3**); a distinction that is too often ignored in the systematics literature because of the general misunderstanding of the meaning of evidential support (see ***Phylogenetic hypotheses and the myth of evidential support***).

At first sight, the algorithmic representation of abduction in (**5**) might appear to be what has often been called "parsimony analysis." The difficulty with this perspective is that parsimony is not a form of reasoning that produces explanatory hypotheses. That responsibility resides with abduction. Parsimony is a comparative measure, where for instance of two hypotheses that predict the same results, the hypothesis that requires fewer assumptions is the simpler of the two. The third premise in (**5**) invokes this principle. That intended action of selection, based on minimising ad hoc hypotheses of homoplasy, is implemented for the sole purpose of serving as a surrogate for the application of the *Phylogenetic theory* in (**4**). This is not tantamount to a parsimony analysis or method. It is an instance of abductive reasoning.

It was noted earlier that abduction is the only type of reasoning that produces explanatory hypotheses; it is the only form of reasoning that introduces new ideas. The depiction of

abduction in (**4**), or its algorithmic form in (**5**), is not only at odds with "parsimony" being an inference procedure but also contrary to claims in systematics that so-called "likelihood" or "Bayesian" methods can infer explanatory hypotheses. With his introduction of the likelihood principle, *Fisher (1934*; see also *Hacking, 1965*; *Hacking, 2001*; *Edwards, 1972*) was clear that it is only relevant to considerations of test evidence *subsequent* to inferring hypotheses. Evidence *e* in representations of likelihood, *e.g.*, $p(h|e) = l(h|e)$, refer to *test evidence* [cf. (**2**), (**3**)]. Probability *p* quantifies anticipation of an outcome of testing, whereas likelihood *l* quantifies trust in the hypothesis as a consequence of testing. Characters used to abductively infer phylogenetic hypotheses [cf. (**1**), (**4**), (**5**)] do not constitute test evidence, and likelihood does not provide the means to infer explanatory hypotheses (*contra e.g.*, *Felsenstein, 1981*; *Felsenstein, 2004*; *Swofford et al., 1996*; *Huelsenbeck & Crandall, 1997*; *Haber, 2011*).

The intent of Bayes Theorem is to quantify changes in belief in hypotheses following the introduction of test evidence (*Salmon, 1967*; *Howson & Urbach, 1993*; *Hacking, 2001*; *contra e.g.*, *Huelsenbeck & Ronquist, 2001*; *Huelsenbeck et al., 2001*; *Archibald, Mort & Crawford, 2003*; *Felsenstein, 2004*; *Ronquist, van der Mark & Huelsenbeck, 2009*). Bayesianism is inductive *sensu* [**3**]; it has no relevance to abduction. The attempt to force Bayes' Theorem into phylogenetic inference (*e.g.*, *Huelsenbeck et al., 2001*: fig. 1) is represented as,

$$p(tree|characters) = \frac{p(characters|tree) \cdot p(tree)}{p(characters)}$$

This is not a meaningful implementation of the Theorem (*Fitzhugh, 2010a*; *Fitzhugh, 2012*; *Fitzhugh, 2016a*; *Fitzhugh, 2016c*; *Fitzhugh, 2016d*). The supposed prior probability, $p(tree)$, refers to all possible "tree topologies" for taxa considered. Such topologies are nothing but branching diagrams devoid of the empirical content typical of cladograms that imply the several classes of causal events discussed earlier (Fig. 2). Tree diagrams disconnected from characters explained by those diagrams cannot be assigned prior probabilities. More problematic is the use of differentially shared characters as "evidence" to determine posterior probabilities. Character data cannot serve as test evidence for the hypotheses intended to explain those data (see ***Phylogenetic hypotheses and the myth of evidential support***). As with "parsimony" and "likelihood" methods, the "Bayesian" approach in systematics fails to live up to its name.

In sum, neither "parsimony," "likelihood," nor "Bayesianism" are valid forms of reasoning or methods for inferring explanatory hypotheses. Unfortunately, wider recognition of the fundamental importance of abductive reasoning did not take hold among scientists until *Hanson (1958)* pointed out its significance in his *Patterns of Discovery*, and later acknowledgment of its importance in artificial intelligence (*Thagard, 1988*) and science in general were embraced (*Fann, 1970*; *Reilly, 1970*; *Thagard, 1988*; *Josephson & Josephson, 1994*; *Magnani, 2001*; *Magnani, 2009*; *Magnani, 2017*; *Psillos, 2002*; *Psillos, 2011*; *Walton, 2004*; *Gabbay & Woods, 2005*; *Klienhans, Buskes & De Regt, 2005*; *Aliseda, 2006*; *Schurz, 2008*; *Klienhans, Buskes & De Regt, 2010*; *Park, 2017*; *Anjum & Mumford, 2018*). In the absence of awareness about abduction, systematics since the 1960's was left with an inferential void that was filled with incorrect applications under the false dichotomy of

parsimony *versus* induction vis-à-vis statistics, where the latter led to incorrectly applying likelihood and Bayesianism. What is actually being performed in phylogenetic inference under these labels is abductive reasoning, albeit in a most peculiar and not entirely philosophically or scientifically acceptable manner.

Regarding abductive inferences of phylogenetic hypotheses to explain sequence data, there are two significant issues to consider. One issue will be discussed here, while the other will be addressed later (see ***Sequence data and explanatory hypotheses***). The treatment of sequence data has often been seen as requiring the assumption of rates of substitution as part of the inferences of phylogenetic hypotheses, along with "branch lengths" (*i.e.,* number of substitutions or character changes that have occurred "along" the branches of phylogenetic trees) (*Swofford et al., 1996*; *Felsenstein, 2004*; *Schmidt & von Haeseler, 2009*). Once we acknowledge that trees are only graphic devices for implying the various classes of explanatory hypotheses accounting for observed characters (Fig. 2), it becomes apparent that those trees cannot have properties like lengths of branches. The need to consider substitution rates has warranted the development of so-called "model-based" approaches (*Jukes & Cantor, 1969*; *Felsenstein, 1978*; *Felsenstein, 2004*; *Farris, 1983*; *Swofford et al., 1996*; *Sullivan & Joyce, 2005*; *Rindal & Brower, 2011*; *Kapli, Yang & Telford, 2020*), under the guises of "likelihood" and "Bayesian," *versus* the "model-free" method of "parsimony analysis." It should be apparent by now that this distinction, just like the methods, is without merit. If we accept that phylogenetic inferences are abductive, then the inclusion of substitution rates within abduction would have the following form (*Fitzhugh, 2021*):

(6)  *Background knowledge* (partim): Observation statements assumed true, per alignments of sequence data

*Theories*: (a) Rates of sequence substitution, applied to sequence data;

(b) Common cause Phylogenetic theory of common ancestry, cf. (4), per Background knowledge

*Observation statements*: Represented in data matrix as why-questions

*Phylogenetic hypotheses*:Diagrammatically implied by graphic devices called cladograms.

Notice that this inference relies on theories that are contradictory. The theory of rates of sequence substitution is at odds with the *Background knowledge* of true observation statements of aligned sequences, and by extension also at odds with the implied why-questions one would ask for which the abduction seeks answers using a common-cause theory. Correcting for this contradiction would necessitate removing the theory of substitution rates as a premise in the abduction and placing it within the *Background knowledge*, such that originally aligned sequence data would be modified to take substitution rates into consideration. By doing this, revised observation statements could be presented, which re-establish assumptions that observation statements are true. The *Background knowledge* in (**6**) would then have the following revised form *Fitzhugh (2021)*,

(7)  *Background knowledge, revised*:
   (a) observation statements of differentially shared characters, are initially assumed true, per alignments of sequence data;
   (b) rates of sequence substitution taken into consideration;
   (c) per (b), the assumption of truth of observation statements in (a) is not necessarily correct;
   (d) per (b) and (c), observation statements are revised such that apparent shared nucleotides are renamed as *different* observation statements where applicable to re-establish true statements.

With the revised *Background knowledge* in (7), the ensuing abductive inference would then have the form shown in (4), which again provides answers to why-questions by way of common cause; there is no need to consider substitution rates within the inference. The inherent difficulty of accomplishing step (7)(d), however, will likely preclude even proceeding forward with causally accounting for those sequence data given that there are no empirical criteria for discerning identical nucleotides or amino acids as not being identical. This is a matter that has been overlooked in systematics. The alternative solution is to simply accept the truth of shared characters obtained through alignment, in which case abduction would again be as shown in (4) and neither substitution rates nor branch lengths would be considered.

We have emphasised that cladograms or phylogenetic trees satisfy the goal of scientific inquiry if they are interpreted as composite explanatory hypotheses (Fig. 3). Whilst the explanatory nature of cladograms is sometimes explicitly acknowledged (*e.g.*, *Farris, 1983*; *Swofford et al., 1996*; *Schmidt & von Haeseler, 2009*; *Kapli, Yang & Telford, 2020*), often the greater focus of phylogenetic studies is on just obtaining "trees," with little to no causal considerations given (*e.g.*, *Felsenstein, 1981*; *Felsenstein, 2004*; *Sullivan & Joyce, 2005*; *Kapli, Yang & Telford, 2020*). Consequences of this emphasis on trees as opposed to explanations includes an erroneous method called character mapping (*Fitzhugh, 2014*). Mapping most often involves obtaining phylogenetic trees using sequence data, then "optimising" other characters, usually morphological, onto these diagrams, from which are determined vague evolutionary conclusions regarding the mapped characters. This approach has become popular among polychaete phylogenetic studies (*e.g.*, *Struck et al., 2011*; *Borda et al., 2012*; *Glasby, Schroeder & Aguado, 2012*; *Goto et al., 2013*; *Weigert et al., 2014*; *Aguado et al., 2015*; *Andrade et al., 2015*; *Struck et al., 2015*; *Goto, 2016*; *Weigert & Bleidorn, 2016*; *Kobayashi et al., 2018*; *Nygren et al., 2018*; *Langeneck et al., 2019*; *Shimabukuro et al., 2019*; *Radashevsky et al., 2020*; *Martín et al., 2020*; *Tilic et al., 2020*; *Gonzalez et al., 2021*). Much credence has been given to mapping, albeit without foundation. For instance, *Kapli, Yang & Telford (2020)* state that,

Mapping heritable character states (phenotypic or genotypic) onto a tree is the basis of different evolutionary analyses: it allows us, for example, to make inferences [sic] about character homology and also to gain insights into character loss and convergent evolution[….] Whereas trees were initially based to a great extent on morphological

characters, biological molecules - nucleic acids and proteins - provide a far more powerful [sic] and plentiful source of information for reconstructing trees.

Mapping is flawed for a simple reason. That phylogenetic inference is abductive means inferring hypotheses, implied by a cladogram, for one set of characters produces conclusions relevant only to those characters. Any subsequent "mapping" of additional characters on that cladogram is not a legitimate form of inference, abductive or otherwise. The premises of a "mapping inference" make this apparent:

(8)  • tree topology, (*a-us* (*b-us* (*c-us* (*d-us*, *e-us*))))
     • observed effects, as differentially shared characters, *e*
     ___________________________________________
     • ???

As a premise, a tree topology is an empirically empty statement since it would have been previously inferred to account for characters other than those in the second premise. Attempting to apply that first premise to the second is pointless since this cannot lead to any interpretable conclusion—certainly not explanatory hypotheses. As such it is impossible to speak of explanations of mapped characters. Like the past and present uncritical advocacy of phylogenetic methods called "parsimony," "likelihood," and "Bayesian," the popularity of character mapping is a further testament to the lack of consideration of formal reasoning in systematics.

### Phylogenetic hypotheses and the myth of evidential support

The present study does not assess the veracity of either the Magelonidae or less inclusive phylogenetic hypotheses in terms of evidential "support," nor are there attempts to use popular methods such as the bootstrap or Bremer analysis to assert that hypotheses are either "strongly supported" or "robust." Such perspectives have become all too common in systematics research, including those on annelids (*e.g., Struck et al., 2011; Borda et al., 2012; Weigert et al., 2014; Aguado et al., 2015; Andrade et al., 2015; Struck et al., 2015; Weigert & Bleidorn, 2016; Gonzalez et al., 2018; Nygren et al., 2018; Shimabukuro et al., 2019; Martín et al., 2020; Stiller et al., 2020; Tilic et al., 2020; Gonzalez et al., 2021*). Reasons for not speaking of support in the present study follow from the basics of reasoning described above. More detailed treatments of misconceptions regarding evidence of, and support for phylogenetic hypotheses can be found in *Fitzhugh (2012), Fitzhugh (2016b), Fitzhugh (2016c)*.

First, to correctly speak of *evidence* means to refer to the premises that allow for a certain conclusion (*Longino, 1979; Salmon, 1984a; Achinstein, 2001*). Whilst that evidence is said to "support" a given conclusion, it is only inductive, *i.e.,* test evidence or support (cf. [3]) that matters in the pursuit of empirical evaluation, whether in matters of everyday life or during scientific inquiry. An emphasis on test evidence certainly permeates all of evolutionary biology and by extension it is understandable that it would be of concern in systematics. But this requires being clear about (1) what is meant by the test evidence,

(2) if that evidence is being used in the proper context, and (3) whether or not bootstrap and Bremer "support" values are meaningfully interpretable in relation to phylogenetic hypotheses.

Recall that the examples of ab-, de-, and induction in (1)–(3) draw clear distinctions between premises and conclusions derived from those premises. Those premises constitute the *evidence for* the respective conclusions [see also (4)–(6)]. Beyond basic matters of logic, we need to be cognisant of the way evidence is interpreted in the separate stages of inquiry. Evidence in relation to abduction is unremarkable in the sense that conclusions must be as they are because of the premises [cf. (1), (4), (5)] (*Fitzhugh, 2010a*; *Fitzhugh, 2012*; *Fitzhugh, 2016a*; *Fitzhugh, 2016c*; *Fitzhugh, 2016d*). As well, abductive conclusions cannot be changed or "defeated" with the introduction of additional effects to be explained. Adding those effects as premises only results in the inference of a new set of hypotheses that have no relevance to previously inferred hypotheses.

A related misconception is that statistical consistency applies to phylogenetic inference (*e.g.*, *Felsenstein, 1978*; *Felsenstein, 2004*; *Swofford et al., 1996*; *Heath, Hedtke & Hillis, 2008*; *Assis, 2014*; *Brower, 2018*). When correctly interpreted, consistency is the view that as more and more *test evidence* is introduced in support of a hypothesis, the closer one is supposedly getting to a true proposition. The mistaken interpretation of consistency in systematics has led to the popular yet erroneous view that sequence data are beneficial for discerning "phylogenies" because of the vast increase in characters that are available (*e.g.*, *Goto, 2016*; *Tilic et al., 2020*). For example, in relation to phylogenetic inference, *Felsenstein (2004*: 107, 121, respectively) states,

> "An estimator is consistent if, as the amount of data gets larger and larger (approaching infinity), the estimator converges to the true value of the parameter with probability 1…"

> "The inconsistency of parsimony [sic] has been the strongest challenge to its use. It becomes difficult to argue that parsimony methods have logical and philosophical priority, if one accepts that consistency is a highly desirable property."

Contrary to claims that consistency is relevant to phylogenetic inference, it has been acknowledged since the early 20th century that continued additions of effects to abductive inferences provides no indication that a true, as opposed to plausible, set of explanatory hypotheses has been attained (*Peirce, 1902*; *Peirce, 1932*: 2.774–777; *Rescher, 1978*; *Fitzhugh, 2012*; *Fitzhugh, 2016b*). As new effects in need of explanation are sequentially added to abductive premises, each subsequent conclusion will only offer a new set of hypotheses that have no relevance to previously inferred hypotheses. The emphasis on consistency in systematics is yet another consequence of abduction not being recognised in lieu of a misplaced statistical (inductive) mindset.

Deduction serves to conclude predictions of potential test evidence [cf. (2)] that might be later sought during the process of testing. Much like premises-as-evidence in abduction, the premises in deduction do not warrant any special attention in terms of "support." Where evidence is of importance is during the act of testing, *i.e.,* induction [cf. (3)]. For instance,

a conclusion that test evidence supports or confirms a hypothesis can be subsequently revised if additional support through further testing is obtained, or additional test evidence might eventually lead to a conclusion of disconfirmation. As noted by *Lipton (2005)*, the process of testing puts hypotheses at risk of being disconfirmed since there is no guarantee predicted test evidence, *via* (**2**) and (**3**), will be found. Obtaining contrary evidence is always a possibility. It is test evidence that matters when speaking of support. Abductive "support" lacks that qualification, and character data are not test evidence.

Because an emphasis on causality has been largely wanting in recent decades in relation to inferences of taxa-as-explanatory-hypotheses, references to support for phylogenetic hypotheses has centered on support [sic] for "groups" or "clades" within cladograms (*Fitzhugh, 2012*). Such claims of support are misleading for two reasons. First, as mentioned above, "support" for any abductive conclusion is trivial since the conclusion could not be otherwise given the premises. In other words, unlike test evidence associated with induction, observations of characters do not present any risk to the hypotheses explaining those characters. To assert, for example, that characters "support" some phylogenetic hypothesis verges on circularity: the conclusion(s) is/are determined by the premises and the premises support the conclusion(s). Second, our concern is not for groups, but rather various explanatory hypotheses. As all phylogenetic hypotheses are composite ([Fig. 3]), an emphasis on groups is both artificial and denies consideration of the variety of explanatory accounts implied by cladograms. Support in relation to groups is epistemically meaningless.

There is a third difficulty, faced by the two popular "support" measures for phylogenetic hypotheses [sic] (*Efron, 1979*; *Efron & Gong, 1983*; *Felsenstein, 1985*; *Felsenstein, 2004*; *Bremer, 1988*; *Bremer, 1994*; *Efron & Tibshirani, 1993*; *Davis, 1995*; *Efron, Halloran & Holmes, 1996*; *Holmes, 2003*; *Soltis & Soltis, 2003*; *Fitzhugh, 2006a*; *Fitzhugh, 2012*; *Fitzhugh, 2016c*; *Lemoine et al., 2018*). Neither method provides epistemically meaningful values regarding evidential support, contrary to the popularity of their use, or claims that they provide a basis to say phylogenetic hypotheses are "robust." The bootstrap was originally developed to test statistical hypotheses through a process of random resampling from an original set of random samples taken from a population. The intent is to test the hypothesis that a statistical parameter, inferred from the original samples, has the hypothesised value, without having to perform additional sampling from the population. It is important to note that the bootstrap approach is designed to test statistical, *not* explanatory hypotheses. Phylogenetic hypotheses are not statistical constructs, as clearly shown in (**4**) and (**5**). Like explanatory hypotheses, however, hypotheses of statistical estimates are inferred by way of abduction. Such estimates are generalisations that account for observed instances as representative of the population from which the sample was drawn. For example,

(**9**)    ●Some balls in this bag are red
         ●25% of balls in this random sample are red
         ———————————————————————————————
         ●*Hypothesis*: 25% of all balls in this bag are red.

Testing the hypothesis would proceed by predicting what should be observed if additional random samples are taken. Such an inference could not be deductive, but rather what *Peirce (1932)*: 2.268) called a "statistical deduction,"

**(10)** • *Hypothesis to be tested*: 25% of all balls in this bag are red
• A random sample of balls will be taken from the bag

• *Predicted test evidence*: 25% of balls in the sample will be red.

Notice that the prediction of test evidence in **(10)** is different from what is shown for the explanatory hypothesis in **(2)**. Statistical hypotheses are tested by seeking the same class of data used to originally infer the hypotheses, whereas testing explanatory hypotheses requires test evidence that is different from the effects originally used to infer the hypotheses (*Fitzhugh, 2016c*; *Fitzhugh, 2016d*).

As suggested by *Felsenstein (1985)* and *Felsenstein (2004)*, the bootstrap method applied to phylogenetic hypotheses involves randomly sampling characters from an original character data matrix to create contrived data sets of the same size from which new cladograms are produced. Keep in mind that these cladograms, as "bootstrap replicates," are *not* explanatory hypotheses; they are merely branching diagrams derived from artificial data sets. Yet, it is claimed that the more often "clades" (actually only branching patterns) are obtained among these "replicates" that match the original groupings indicates higher "support" for those clades. This claim is, however, entirely incorrect. The "bootstrap replicates" have no epistemic relation to the original phylogenetic hypothesis(es) under consideration, such that any measure of "support" would be impossible to ascertain. Along with the statistical, *i.e.,* inductive, mindset that was introduced into phylogenetic inference in the 1970's, the bootstrap method applied to phylogenetic hypotheses was largely an exercise in filling a perceived methodological void. Had the abductive nature of systematics been recognised early on, and character data not confused with test evidence, the state of the field might have avoided so many scientifically questionable schemes. The only support relevant to cladograms is valid test evidence produced in the process of testing the various classes of hypotheses within a (composite) phylogenetic hypothesis [Fig. 3; cf. **(2)**, **(3)**]. But as noted earlier, such testing virtually never occurs, making the need for talk of support both gratuitous and positively deceptive.

Just as the bootstrap provides results that cannot be interpreted as support for phylogenetic hypotheses, the Bremer Index produces values that are equally impossible to defend. Like the bootstrap, Bremer Index is claimed to determine support for groups, as opposed to any particular hypotheses implied by cladograms. This support is based on the extent to which groups or "clades" are present in cladograms of ever-increasing "length" beyond the original minimum-length tree. The greater the tree length with a particular group still present is supposed to represent greater "support" for or resiliency by that group. The problem is that trees of greater length can only be interpreted as sets of explanatory hypotheses that have no epistemic relation to the originally inferred hypotheses. The

consequence is that values provided by the Bremer Index offer no indication of support. Once again, in the absence of actually testing hypotheses, as indicated in (**2**)–(**3**), no evidential test support is possible for hypotheses.

Finally, an oblique yet equally specious approach to garnering support comes from comparisons of cladograms inferred from different sets of characters, sometimes referred to as taxonomic congruence (*Fitzhugh, 2014*; *Fitzhugh, 2016c*). The idea is that if tree topologies are identical, or nearly so, this justifies a sense of greater confidence or belief in those topologies (not the implied explanatory hypotheses). Just as was noted earlier with regard to the error of character mapping, comparisons of cladograms are an inferentially indefensible approach. Cladograms inferred to explain separate sets of characters are distinct compilations of explanatory constructs that have no epistemic relevance to each other. A popular, somewhat related approach is to speak of comparisons of past and present cladogram topologies inferred from updated compilations of observations, remarking on the degree of similarities between groups in those topologies. The same basic criticism against taxonomic congruence applies here. For example, previous abductive inferences using data set $D_1$ give composite phylogenetic hypothesis $H_1$. With subsequent characters added, $D_{1+2}$, a new composite phylogenetic hypothesis, $H_2$, is inferred. It might seem reasonable to speak of topological similarities and differences between $H_1$ and $H_2$ but this is a mistake. Again, $H_1$ and $H_2$ are distinct sets of explanatory hypotheses from different sets of abductive premises. Just as there are no meaningful comparisons to be made between the actual explanatory hypotheses implied by the different cladograms (cf. Figure 2), there are no relevant conclusions to be made regarding topological similarity or difference. In the absence of recognising the nature of abduction or associated intent of those inferences, focusing on phylogenetic tree comparisons offers no scientifically meaningful evaluation.

### *Sequence data and explanatory hypotheses*

The pursuit of sequence data was not given consideration in this study. The reasoning behind this decision will be outlined in this subsection. A more complete account is presented in *Fitzhugh (2016b*; see also *Nogueira et al., 2017*: 683–684, *Lovell & Fitzhugh, 2020*: 270, and *Fitzhugh, 2021*). Contrary to a claim made by a reviewer of an earlier draft of this paper, what is presented in this section should not be interpreted as the assertion that we "do not believe in sequence data." Rather, our intent is to point out inherent and significant difficulties that generally preclude causally accounting for differentially shared nucleotides or amino acids *via* inferences of phylogenetic hypotheses.

The line of argument developed in this more inclusive section—establishing the basis for systematics—follows from the fact that the goal of scientific inquiry is to continually acquire causal understanding of phenomena, in the form of differentially shared characters of organisms, that we encounter as well as describe. The pursuit of that understanding begins with reactions to those effects, in the form of implicit or explicit why-questions, and subsequently inferring answers to those questions by way of abductive reasoning. As discussed earlier, abduction produces hypotheses that posit possible past causal events/conditions that account for observed effects, as shown in (**1**), (**4**), and (**5**). And, almost all actions in systematics are centered around abduction as opposed to testing

hypotheses *via* induction (*Fitzhugh, 2005a*; *Fitzhugh, 2006a*; *Fitzhugh, 2006b*; *Fitzhugh, 2008a*; *Fitzhugh, 2009*; *Fitzhugh, 2010a*; *Fitzhugh, 2012*; *Fitzhugh, 2013*; *Fitzhugh, 2014*; *Fitzhugh, 2015*; *Fitzhugh, 2016a*; *Fitzhugh, 2016b*; *Fitzhugh, 2016c*; *Fitzhugh, 2016d*).

We pointed out earlier that composite phylogenetic hypotheses provide almost no causal specifics among the four classes of hypotheses implied by cladograms (Fig. 3). This is a consequence of the fact that phylogenetics computer algorithms emulate a vague form of abduction [e.g., (4), (5)]. Consider again what the *Phylogenetic theory* in (4) offers regarding character origin/fixation:

> character $x(1)$ originates by [unstated] mechanisms $a, b, c \ldots n$, and becomes fixed within the population by [unstated] mechanisms $d, e, f \ldots n$ (= ancestral species hypothesis), followed by [unstated] event(s) $g, h, i \ldots n$, wherein the population is divided into two or more reproductively isolated populations.

The cause of character fixation in an ancestral population could at a minimum be assumed to be natural selection or genetic drift. In the absence of explicit causal specifics being presented in the *Phylogenetic theory*, it would have to be assumed that either selection or drift are reasonable causal alternatives. This equivalence is plausible for explaining phenotypic characters but cannot be applied to sequence data. For selection to operate, whether purifying or directional, variation among heritable traits within a population must be identified that directly result in differential fitness among individuals. Nucleotides and amino acids lack emergent properties that could *directly* manifest fitness differences (for important nuances, cf. *Linquist, Doolittle & Palazzo, 2020*). Whilst there are tests that can indicate whether selection has occurred in relation to sequence data, such as the McDonald–Kreitman test (*McDonald & Kreitman, 1991*; *Sawyer, 1994*; *Hey, 1999*; *Hurst, 2002*; *Biswas & Akey, 2006*; *Zhang & Yu, 2006*; *Koonin, 2012*; *Petrov, 2014*), these tests cannot discriminate at what organisational level selection might have been directed. It is at higher organisational levels that phenotypic characters can result in fitness variation. If selection is a causal factor, it comes into play due to phenotypic effects at these higher levels. Alternatively, since there is no *direct* selection for sequence data, phenotypic characters have the potential to lead to differential fitness and intergenerational selection that can influence the occurrences of those sequence data that produce the selected characters. The result is a form of *indirect* selection, termed downward causation by *Campbell (1974*: 180),

> "Where natural selection operates through life and death at a higher level of organisation, the laws of the higher-level selective system determine in part the distribution of lower-level events and substances."

Since the 1970's the importance of downward causation has become increasingly recognised (*e.g.*, *Vrba & Eldredge, 1984*; *Salthe, 1985*; *Lloyd, 1988*; *Auletta, Ellis & Jaeger, 2008*; *Ellis, 2008*; *Ellis, 2012*; *Ellis, 2013*; *Ellis, Noble & O'Connor, 2011*; *Jaeger & Calkins, 2011*; *Laland et al., 2011*; *Martínez & Moya, 2011*; *Davies, 2012*; *Okasha, 2012*; *Walker, Cisneros & Davies, 2012*; *Griffiths & Stotz, 2013*; *Martínez & Esposito, 2014*; *Walker, 2014*; *Mundy, 2016*; *Callier, 2018*; *Pouyet et al., 2018*; *Salas, 2019*; *Yu et al., 2020*), but its relevance
to how sequence data are explained *via* phylogenetic hypotheses has only recently been considered (*Fitzhugh, 2016b*).

If there can be no direct selection for sequence data, the issue then becomes one of deciding to explain individual nucleotides or amino acids either directly *via* drift or indirectly through downward causation. If the latter, it is first necessary to associate those sequence data with the higher-level phenotypic characters to which selection is the hypothesised causal condition. Those associated sequence data would then be excluded from phylogenetic inferences since they would already be explained in conjunction with higher-level characters (*Fitzhugh, 2016b*). On the other hand, it might be argued that one could avoid the issue altogether by assuming all sequence data should be explained by drift. This would be unrealistic as it necessitates that selection at higher organisational levels never occurs. In the absence of being able to discriminate between drift *versus* selection by downward causation, the only sensible option is to forgo altogether attempts to causally account for shared nucleotides or amino acids by way of phylogenetic inference. Otherwise, to explain shared nucleotides or amino acids with phylogenetic hypotheses inferred under an entirely agnostic perspective [*i.e.*, the *Phylogenetic theory* in (**4**)] would be epistemically and scientifically unwarranted.

The inherent limitations for inferring explanatory hypotheses for sequence data have associated relevant consequences for mapping morphological characters on cladograms only inferred for sequence data. As discussed earlier (***Phylogenetic inference =abduction***; see also *Fitzhugh, 2014*), mapping is an epistemically unfounded approach to explaining characters. This problem is compounded by the fact that not discriminating between genetic drift and natural selection as possible causal factors when explaining sequence data yet again precludes rational acceptance of phylogenetic hypotheses accounting for mapped characters.

Finally, it might appear at first sight that the exclusion of sequence data on either the basis of downward causation or inability to distinguish drift from selection is at odds with the requirement of total evidence (RTE; cf. *Fitzhugh, 2006*, for a discussion of the RTE in relation to systematics). Such is not the case. The RTE is a recognised maxim for all non-deductive reasoning (the requirement is automatically satisfied in deduction), wherein rational acceptance of a conclusion is based on considering all relevant evidence that can affect support (either negative or positive) for that conclusion (*Carnap, 1950*; *Barker, 1957*; *Hempel, 1962*; *Hempel, 1965*; *Hempel, 1966*; *Hempel, 2001*; *Salmon, 1967*; *Salmon, 1984a*; *Salmon, 1984b*; *Salmon, 1989*; *Salmon, 1998*; *Sober, 1975*; *Fetzer, 1993*; *Fetzer & Almeder, 1993*): "All that the requirement of total evidence says is that one's confidence in a hypothesis must be proportional to the support that that hypothesis receives from one's evidence…" (*Neta, 2008*: 91). Nearly all of the philosophical literature on the RTE has focused on test evidence in relation to induction [cf. (**3**)], which is understandable given that, for instance, to exclude known test evidence that could enhance or compromise acceptance of a theory or hypothesis would be less than rational.

Whilst abductive reasoning has almost never figured into discussions of the RTE, the non-deductive nature of abduction necessitates its attention (*Fitzhugh, 2006b*). Consider

the following example. For members of three species, *a-us*, *b-us*, and *c-us*, there are characters distributed among subjects 1–3 (cf. *Fitzhugh, 2006c*):

| Subjects: | 1 | 2 | 3 |
|-----------|---|---|---|
| Outgroup  | 0 | 0 | 0 |
| *a-us*    | 0 | 1 | 1 |
| *b-us*    | 1 | 0 | 1 |
| *c-us*    | 1 | 1 | 0 |

If the characters among members of *a-us*, *b-us*, and *c-us* are explained separately, there would be three sets of explanatory hypotheses (outgroup is excluded), represented here in parenthetical form:

Subject 1: (*a-us* (*b-us*, *c-us*))
Subject 2: (*b-us* (*a-us*, *c-us*))
Subject 3: (*c-us* (*a-us*, *b-us*)).

The three sets of hypotheses are contradictory, and while each is supported by abductive evidence [cf. (**1**)], choosing among them based on the information provided would be problematic. This is the sort of situation the RTE is intended to address, as well as those instances that involve other degrees of data partitioning in relation to inferring phylogenetic hypotheses and character mapping. The nature of the abductive inferences involves explaining features of semaphoronts, such that the observed characters are part of the integrated network that makes up those individuals. The various phylogenetic theories applied to those characters [cf. (**1**)] assume that past causes operated on individuals, not distinct characters. Thus, explaining the characters under each subject would be relevant to explaining the other characters. The plausibility for any of the phylogenetic hypotheses would be based on an abductive inference where all the characters are treated as part of the premises.

Related to the topic highlighted in this subsection, it would not be a violation of the RTE to exclude from phylogenetic inferences those sequence data for which no empirical basis is available for discriminating drift from selection-via-downward-causation for sequence data. The difficulty faced is not a matter that falls under the purview of the RTE, but rather an inability to engage in abductive reasoning for a particular class of characters.

## Sequence data are not more objective than other classes of observations

Consider the following statements,

> "Molecular data are more objective and subject to considerably more rigor than morphological data. DNA sequence contains four easily identified and mutually exclusive character states[….] Morphological and embryological character definitions and scoring of character states are far more subjective, and most characters have been repeatedly used without critical evaluation, calling into question the utility of morphological cladistic studies that span Metazoa[….]" (*Halanych, 2004*: 230).

"As we move forward with the deep-animal tree [sic], we should not employ morphology and developmental data to reconstruct the tree. There is too much historical baggage and subjectivity with these data. We should reconstruct the tree with molecular data and then use that tree to independently interpret the morphology and development in light of that tree[….]" (*Halanych, 2016*: 325).

"We present the view that rigorous and critical anatomical studies of fewer morphological characters, in the context of molecular phylogenies, is a more fruitful approach to integrating the strengths of morphological data with those of sequence data[....] We argue[...] that a main constraint of morphology-based phylogenetic inference concerns the limited number of unambiguous characters available for analysis in a transformational framework" (*Scotland, Olmstead & Bennett, 2003*: 539).

Taken at face value, each statement appears damning of all or most observations of features of organisms other than sequence data in relation to inferring phylogenetic hypotheses. We have seen throughout this section, however, that the disconnect between principles of scientific inquiry and recent views on phylogenetic inference have led to perspectives that cannot be defended. This outcome extends to the notion that sequence data are somehow "more objective," "more fruitful," or more reliable than other observations. Asserting that one class of observations is more effective than other classes presumes that the objective of phylogenetic inference is to obtain "phylogenies" or "trees," which is at odds with pursuing explanations of all relevant observations. If the objective of systematics is causal explanation, thus consistent with all of scientific inquiry, then the pursuit of "phylogenies" or "trees" is not only contrary to that objective but has led to erroneous inclinations in addition to the largely contrived objective/subjective dichotomy. These erroneous pursuits include, but are not limited to (1) inferring phylogenetic hypotheses for separate classes of characters, especially sequence data *versus* morphology, and drawing comparisons between tree "topologies" (cf. **Phylogenetic hypotheses and the myth of evidential support**), and (2) inferring phylogenetic hypotheses for sequence data, then "mapping" morphological characters on the "tree" and making evolutionary claims regarding the latter characters (cf. **Phylogenetic inference =abduction**). To address the claim that sequence data are more objective or less ambiguous requires that we first consider what is meant by scientific objectivity. Then there are several additional issues in the above quotes that need to be addressed.

As might be expected, objectivity has multiple meanings. Useful overviews can be found in *Daston & Galison (2007)*, *Gaukroger (2012)*, *Reiss & Sprenger (2017)*, and *Wilson (2017)*. Common interpretations of objectivity range from the view that judgements should be (a) free of prejudice or bias, (b) free of assumptions, and/or (c) an accurate representation of, or "faithfulness" to facts (*Gaukroger, 2012*). Each meaning can be applied to everyday life as well as within fields of science. Regarding freedom from prejudice or bias, this is to speak of limiting levels of distortion beyond what is perceived. Freedom from assumptions would be impossible to accomplish given the theory-laden nature of observation statements and other types of propositions: "Every instance of scientific inquiry, every study, rests on a vast submerged set of political, moral, and ultimately metaphysical assumptions" (*Wilson,*

*2017*). Scientific objectivity has often been interpreted along the lines of "faithfulness to facts" (*Daston & Galison, 2007*; *Reiss & Sprenger, 2017*), where we attempt to minimise arbitrary judgements; we want to answer questions on the basis of the most appropriate evidence we have available. Objectivity occurs in degrees—there is no absolute objectivity, which is why objectivity cannot be equated with truth, which is absolute. Ultimately, "While we can strive for objectivity, inquiry is inherently subjective" (*Spencer, 2020*).

*Reiss & Sprenger (2017)* distinguish "product" and "process" objectivity. Product objectivity regards the products of scientific inferences, such as theories, laws, and observation statements, as accurate representations of the external world. Process objectivity refers to processes and methods that are not dependent on social or ethical values, or the individual biases of the scientist. It is product objectivity that is of concern when it comes to assessing the quotes given above, claiming sequence data are in some way more objective or less ambiguous than other classes of observations.

There is nothing in the conception of product objectivity that would support the claim that observation statements of morphological features are less objective than sequence data, much less that such a contrived distinction warrants the exclusion of an entire class of observations from the goal of systematics. Indeed, that biologists hold different interpretations of what they perceive of the properties of organisms is not tantamount to proffering less objective observation statements. Those statements are themselves products of abductive reasoning (*Peirce, 1935*; *Hoffmann, 1995*; *Burton, 2000*; *Magnani, 2009*; *Anjum & Mumford, 2018*), constrained by one's background knowledge, and like all hypotheses, are open to empirical evaluation. If anything, it can be argued that sequence data are less than objective relative to other classes of characters given the fact that attempted phylogenetic inferences explaining differentially shared nucleotides or amino acids typically operate under the assumption that rates of substitution must be imposed within abductive inferences (cf. *Phylogenetic inference =abduction*). That assumption carries with it the implication that observation statements of shared characters must be treated as potentially false, hence the need to interject substitution rates within inferences, as indicated by the all-too-common emphasis on branch lengths. Considering the rule for valid reasoning—that we should assume the truth of premises, discussed earlier—the treatment of sequence data with "likelihood" and "Bayesian" methods pales against any criterion of objectivity.

Rather than focusing on the false argument from objectivity as a basis to eliminate observations of morphological characters in need of explanation, systematists should embrace as much as possible all relevant observations per the goal of scientific inquiry discussed earlier and the requirement of total evidence (*Fitzhugh, 2006b*). Two notable exceptions to considering explaining differentially shared characters by way of phylogenetic hypotheses come from conditions discussed earlier: (1) why-questions addressing some characters might not be amenable to being answered by phylogenetic hypotheses, but instead one of the other classes of explanatory hypotheses applied in systematics (Fig. 1); (2) epistemic limitations to explaining sequence data because of an inability to discern explanations due to genetic drift *versus* selection by way of downward causation (cf. *Sequence data and explanatory hypotheses*). Pursuing causality is the intent—not

producing "trees" or "phylogenies," which are typically devoid of causal considerations in relation to sequence data.

## Outgroup and ingroup taxa

Outgroup and ingroup taxa considered in thus study include the following:

| | | |
|---|---|---|
| outgroups – | Chaetopteridae: | *Phyllochaetopterus limicolus* Hartman, 1960 |
| | Spionidae: | *Spio filicornis* (Müller, 1776 |
| | | *Prionospio ehlersi* Fauvel, 1928 |
| | | *P. lighti* Maciolek, 1985 |
| | | *Laonice cirrata* (Sars, 1851) |
| ingroup – | *Magelona* (72 species) | |
| | *Octomagelona* (2 species) | |

*P. limicolus*– *Blake (1996a)*
*S. filicornis*– *Meißner, Bick & Bastrop (2011)*
*P. ehlersi*–*Blake (1996b)*
*P. lighti*–*Blake (1996b)*
*Laonice cirrata*–*Blake (1996b)*

The choice of outgroups ideally would follow from more inclusive phylogenetic studies from which the immediate sister groups to Magelonidae have been ascertained. In this regard, the merits of recent phylogenetic hypotheses are questionable. In her review of the Magelonidae, *Mortimer (2019)* summarised past studies of phylogenetic relationships of magelonids to other groups within Polychaeta. As would be expected for a group as moderately disparate as magelonids, earlier conclusions were that they exhibit close relations with other spioniforms. Recent phylogenetic studies based exclusively on morphological (*Capa, Parapar & Hutchings, 2012*; *Chen et al., 2020*) or sequence characters (*Weigert et al., 2014*; *Helm et al., 2018*) have, however, allied Magelonidae with Oweniidae, while *Weigert et al.*'s (*2016*) use of mitochondrial sequences had Magelonidae and Chaetopteridae as sister taxa. Each approach has aspects that call into question the respective conclusions. For instance, *Capa, Parapar & Hutchings (2012*; see also *Capa, Parapar & Hutchings, 2019*) cautiously suggested a Magelonidae-Oweniidae clade based on the absence of nuchal organs, fusion of pro- and peristomium, presence of a ventral buccal organ, and monociliated cells. The absence of nuchal organs is not exclusive to magelonids and oweniids (*Purschke, 1997*; *Purschke, 2005*). *Mortimer & Mackie (2014)* and *Mortimer (2019)* have shown that pro- and peristomium are not entirely fused, and the magelonid "burrowing organ" is not homologous to the oweniid buccal organ [(*Mortimer, 2019*); see also **Character descriptions**, *Burrowing organ* (**subject 14**), below]. Monociliated cells have only been recorded from among members of *Magelona mirabilis* (*Johnston, 1865*) (*Bartolomaeus, 1995*, N.B. fig. 1A in this publication is not *M. mirabilis*; *Patrick, Helm & Bartolomaeus, 2019*). As part of their description of a fossil magelonid, *Chen et al. (2020*:

fig. 8A; see also *Parry et al., 2016*) presented phylogenetic hypotheses also suggesting Oweniidae and Magelonidae are exclusive sister taxa, but their inferences are based on an overly broad taxon coverage and the compilation of characters is inadequate to address phylogenetic relationships of magelonids to other polychaetes. As discussed in the previous section (**Methodological considerations** -*Sequence data and explanatory hypotheses*), the inherent problems associated with attempting to explain shared nucleotides or amino acids undermine the plausibility of phylogenetic hypotheses inferred from those premises. With the emphasis on an extensive treatment of magelonid morphological characters in this study, the greatest number of characters is shared with members of Spionidae, making them an appropriate outgroup. Members of several genera and species were included to accommodate variation within Spionidae. An additional spioniform representative from the Chaetopteridae, *Phyllochaetopterus limicolus*, was also included.

Ideally, all currently recognised magelonid taxa would be included in inferences of the phylogenetic hypotheses. This, however, relies on several factors. Initially, taxa were chosen based on access to specimens, preferably type material, some of which had been observed previously during descriptions and re-descriptions by the present authors. Specimens were borrowed from the following institutions:

| | |
|---|---|
| BMNH | Natural History Museum, London |
| MB | Museu Nacional de História Natural, Lisboa |
| MNCN | Museo Nacional de Ciencias Naturales, Madrid |
| MNHM | Muséum National d'Histoire Naturelle, Paris |
| NHMG | Gothenburg Museum of Natural History |
| NHMLAC | Natural History Museum of Los Angeles County |
| NMW | Amgueddfa Cymru- National Museum Wales |
| PMBC | Phuket Marine Biological Center |
| SMF | Senckenberg Research Institute and Natural History Museum, Frankfurt |
| USNM | Smithsonian National Museum of Natural History |
| ZMBN | University Museum of Bergen |
| ZMHB | Museum für Naturkunde der Humboldt-Universität zu Berlin |
| ZMUC | Zoological Museum, University of Copenhagen |
| ZUEC | Museu de Zoologia, Universidade de Campinas |

Where specimens were not available for observation, the remaining taxa were chosen based on the level of information and detail in the original descriptions. It is only comparatively recently that the importance of illustrating and fully describing all thoracic chaetigers has been realised for the treatment of taxa for example, and further characters defined. Thus, many of the earliest descriptions lack much of the detail required for the current analysis. All ingroup taxa included in the analyses are listed in Table 2, including which specimens were observed and the institution from which they were borrowed. Table 2 also indicates whether observations were based solely on published material. Species not included within the analysis: *Magelona agoensis* Kitamori, 1967, *M. americana Hartman, 1965*, *M. capax Hartman, 1965*, *M. capensis* Day, 1961, *M. japonica* Okuda, 1937, *M. kamala Nateewathana & Hylleberg, 1991*, *M. koreana* Okuda, 1937, *M. longicornis* Johnston, 1901, *M. methae Nateewathana & Hylleberg, 1991*, *M. mickiminni Nateewathana & Hylleberg,*

*1991*, *M. noppi* *Nateewathana & Hylleberg, 1991*, *M. pectinata* *Nateewathana & Hylleberg, 1991*, *M. petersenae* *Nateewathana & Hylleberg, 1991*, *M. pettiboneae lanceolata* *Jones, 1963*, *M. rosea* Moore, 1907 and *M. sachalinenis* Buzhinskaja, 1985.

## Character descriptions

A total of 176 characters distributed among 79 subjects (Table 3; Table S1) were used to infer phylogenetic hypotheses. Descriptions of characters are presented here.

### Characters of the anterior end: prostomium (subjects 1–9)

The form of the magelonid prostomium (**subject 1**) is considered a synapomorphy of the group, the name "shovel head worms" having been coined due to its peculiar shape. The dorso-ventrally flattened, shovel-shaped prostomium [character 1(1), Figs. 4 and 5] is a feature not observed among members of the outgroups or any other polychaete group. Nuchal organs (**subject 2**) have not been recorded among adult members of any magelonid species [character 2(0)] but are present among all members of the outgroups [character 2(1)].

The distal margin of the magelonid prostomium (**subject 3**) can vary from smooth [character 3(0), Figs. 4A and 4F], crenulate [character 3(1), Figs. 4B and 4C] or medially indented [character 3(2), Fig. 4D]. *Uebelacker & Jones (1984)* stated that the presence of a crenulated prostomial margin is an excellent character for separation of species, although the degree of crenulation present shows great intraspecific variability. The crenulations may differ in number and form, for example, the distinct, triangular crenulations of *Magelona* sp. L of *Uebelacker & Jones (1984)* or *M. crenulifrons* (Fig. 4B), in comparison to the numerous minute crenulations observed for *M. longicornis* or *M. wilsoni* (Fig. 4C). The crenulations among members of some species are difficult to discern and this is perhaps the reason they have been previously overlooked in some descriptions and erroneously reported as absent in others. The prostomial distal margin (**subject 4**) may additionally vary in shape, from triangular [character 4(0), Fig. 4E], to rounded [character 4(1), Figs. 4A and 4F] or straight [character 4(2), Figs. 4G, 4H, 5C and 5E]. As subjects 3 and 4 relate directly to the shovel-shaped prostomium, characteristic of magelonids, it does not apply to any members of the outgroups.

Horns are the distal projections of the prostomium (**subject 5**) that occur among members of some species of magelonids [character 5(1)], such as *Magelona pacifica* (Fig. 5A), or *M. montera* (Fig. 5B). *Uebelacker & Jones (1984)* drew attention to this character for the initial separation of magelonid species. When present, horns can vary in size/shape (**subject 6**), from rudimentary [character 6(0), Figs. 4G, 4H, 5C and 5E], such as those observed among members of *M. variolamellata* and *M. alleni* (often described as a squared anterior margin), or conspicuous structures as seen among members of species such as *M. posterelongata* [character 6(1), Fig. 5D]. In the latter case, these are often very distinct from the distal prostomial margin. Prostomial horns do not occur in the outgroups, thus subjects 5 and 6 are not applicable.

Several authors have identified dimensions of the prostomium (**subject 7**) as an important feature in the separation of magelonid species (*Hartman, 1944*; *Wilson, 1958*;

**Table 2  Sources from which character data were obtained for this study.** Museum abbreviations are explained in the text.

| Species | Material |
| --- | --- |
| *M. alexandrae* Magalhães, Bailey-Brock & Watling, 2018 | Original description plus photos provided by the authors |
| *M. alleni Wilson, 1958* | Holotype: BMNH 1958.5.2.1<br>Paratypes: BMNH 1958.5.2.2-10 |
| *M. annulata* Hartmann-Schröder, 1962 | Original description |
| *M. anuheone* Magalhães, Bailey-Brock & Watling, 2018 | Original description plus photos provided by the authors |
| *M. berkeleyi Jones, 1971* | Original description |
| *M. californica Hartman, 1944* | Original description |
| *M. cepiceps* Mortimer & Mackie, 2006 | Holotype: NMW.Z.2000.020.0209<br>Paratype: NMW.Z.2000.020.0208 |
| *M. cerae* Hartman & Reish, 1950 | Original description |
| *M. cincta* Ehlers, 1908 | Holotype (ZMHB 4531) |
| *M. cinthyae* Magalhães, Bailey-Brock & Watling, 2018 | Original description plus photos provided by the authors |
| *M. conversa Mortimer & Mackie, 2003* | Holotype: NMW.Z.2000.020.0001<br>Paratypes: NMW.Z.2000.020.0002-0007 |
| *M. cornuta* Wesenberg-Lund, 1949 | Holotype: (ZMUC–POL–969; two abdominal slide preparations ZMUC–POL–963) |
| *M. crenulata Bolívar & Lana, 1986* | Original description |
| *M. crenulifrons* Gallardo, 1968 | Paratypes: (ZMUC–POL–1416- 1421) |
| *M. dakini Jones, 1978* | Original description |
| *M. debeerei Clarke et al., 2010* | Original description |
| *M. equilamellae* Harmelin, 1964 | Syntypes: SMF 4675 |
| *M. falcifera Mortimer & Mackie, 2003* | Holotype: NMW.Z.2000.020.0009<br>Paratypes: NMW.Z.2000.020.0008, NMW.Z.2000.020.0010-0017 |
| *Magelona fasciata Mortimer, Kongsrud & Willassen (2021)* | Holotype: ZMBN132144<br>Paratypes: ZMBN; AC-NMW |
| *M. fauchaldi* Shakouri, Mortimer & Dehani, 2017 | Holotype: NMW.Z.2015.012.0002a<br>Paratypes: NMW.Z.2015.012.0001-0002;<br>NMW.Z.2010.037.0002-0006 |
| *M. filiformis Wilson, 1959* | Holotype: BMNH 1959.4.2.1<br>Paratypes: BMNH 1959.4.2.2-10 |
| *M. gemmata Mortimer & Mackie, 2003* | Holotype: NMW.Z.2000.020.0019<br>Paratypes: NMW.Z.2000.020.0018; NMW.Z.2000.020.0020 |
| *Magelona guineensis Mortimer, Kongsrud & Willassen (2021)* | Holotype: ZMBN132137<br>Paratypes: ZMBN; AC-NMW |
| *M. hartmanae Jones, 1978* | Original description |
| *M. hobsonae Jones, 1978* | Original description |
| *M. johnstoni Fiege, Licher & Mackie, 2000* | Cited material from original description from NMW collections |
| *M. jonesi Hartmann-Schröder, 1980* | Original description |
| *M. lenticulata* Gallardo, 1968 | Original description |
| *M. lusitanica* Mortimer, Gil & Fiege, 2011 | Holotype: SMF 9246/1<br>Paratypes: MB29-000176-000181;<br>NMW.Z.2010.010.0001-0006; SMF 9245/1 |

**Table 2** (*continued*)

| Species | Material |
| --- | --- |
| *Magelona mackiei* Mortimer, Kongsrud & Willassen (2021) | Holotype: ZMBN107309<br>Paratypes: ZMBN; AC-NMW |
| *M. magnahamata* Aguado & San Martín, 2004 | Original description |
| *M. mahensis* Mortimer & Mackie, 2006 | Holotype: NMW.Z.2000.020.0188<br>Paratypes: NMW.Z.2000.020.0176-0187;<br>NMW.Z.2000.020.0189-0193 |
| *M. marianae* Hernández-Alcántara & Solís-Weiss (2000) | Original description |
| *M. minuta* Eliason, 1962 | Holotype: NHMG Polych. 11491<br>Additional material: USNM 52510 |
| *M. mirabilis* (Johnston, 1865) | Cited material from original description from NMW collections |
| *M. montera* Mortimer, Cassà, Martin & Gil, 2012 | Holotype: MNHN A895 |
| *Magelona nanseni* Mortimer, Kongsrud & Willassen (2021) | Holotype: ZMBN132141<br>Paratypes: ZMBN; AC-NMW |
| *M. nonatoi* Bolívar & Lana, 1986 | Original description |
| *M. obockensis* Gravier, 1905 | Syntypes, MNHN Type 1357 |
| *M. pacifica* Monro, 1933 | Syntypes: BMNH Type 1933.7.10.65/70 |
| *M. papillicornis* F. Müller, 1858 | Redescription (Jones, 1977) |
| *M. parochilis* Zhou & Mortimer, 2013 | Holotype: ECSFRI100532<br>Paratypes: ECSFRI100533-535;<br>NMW.Z.2012.033.0001 |
| *M. paulolanai* Magalhães, Bailey-Brock & Watling, 2018 | Original description plus photos provided by the authors |
| *M. pettiboneae* Jones, 1963 | Original description |
| *M. phyllisae* Jones, 1963 | Original description |
| *Magelona picta* Mortimer, Kongsrud & Willassen, 2021 | Holotype: ZMBN107338<br>Paratype: ZMBN115737 |
| *M. pitelkai* Hartman, 1944 | Original description, Jones (1978) |
| *M. polydentata* Jones, 1963 | Original description |
| *M. posterelongata* Bolívar & Lana, 1986 | Original description |
| *M. pulchella* Mohammad, 1970 | Holotype: BMNH 1969.391 |
| *M. pygmaea* Nateewathana & Hylleberg, 1991 | Paratype: PMBC 4220<br>Additional material: PMBC 4234; PMBC 4227; PMBC 4241 |
| *M. riojai* Jones, 1963 | Original description |
| *M. sacculata* Hartman, 1961 | Holotype: NHMLA: LACM-AHF POLY 596 |
| *M. sinbadi* Mortimer, Cassà, Martin & Gil, 2012 | Holotype: NMW.Z.2010.037.0001 |
| *M. spinifera* (Hernández-Alcántara & Solís-Weiss, 2000) | Original description |
| *M. symmetrica* Mortimer & Mackie, 2006 | Holotype: NMW.Z.2000.020.0175 |
| *M. tehuanensis* Hernández-Alcántara & Solís-Weiss, 2000 | Original description |
| *M. tinae* Nateewathana & Hylleberg, 1991 | Paratypes: PMBC 3180; PMBC 4251; PMBC 4253; PMBC 4254 |
| *M. uebelackerae* (Hernández-Alcántara & Solís-Weiss, 2000) | Original description |
| *M. variolamellata* Bolívar & Lana, 1986 | Original description |
| *M. wilsoni* Glémarec, 1967 | Paratypes: MNHN Poly Type 1415 |

**Table 2** (*continued*)

| Species | Material |
| --- | --- |
| *Magelona* sp. A | *Uebelacker & Jones (1984)* |
| *Magelona* sp. B | *Uebelacker & Jones (1984)* |
| *Magelona* sp. C | *Uebelacker & Jones (1984)* |
| *Magelona* sp. D | *Uebelacker & Jones (1984)* |
| *Magelona* sp. E | *Uebelacker & Jones (1984)* |
| *Magelona* sp. F | *Uebelacker & Jones (1984)* |
| *Magelona* sp. G | *Uebelacker & Jones (1984)* |
| *Magelona* sp. H | *Uebelacker & Jones (1984)* |
| *Magelona* sp. I | *Uebelacker & Jones (1984)* |
| *Magelona* sp. J | *Uebelacker & Jones (1984)* |
| *Magelona* sp. K | *Uebelacker & Jones (1984)* |
| *Magelona* sp. L | *Uebelacker & Jones (1984)* |
| *O. bizkaiensis Aguirrezabalaga, Ceberio & Fiege, 2001* | Holotype: MNCN 16.01/6887 Paratype: SMF 10025 |
| *Octomagelona* sp. (West Africa) | AC-NMW |

*Wilson, 1959*; *Jones, 1963*; *Jones, 1971*; *Jones, 1978*; *Uebelacker & Jones, 1984*; *Bolívar & Lana, 1986*; *Blake, 1996c*; *Mortimer, 2019*). According to *Uebelacker & Jones (1984)*, the general shape and relative dimensions are fairly constant within each species, something which the current authors are in agreement. Consequently, the ratio of width to length was used to recognise three prostomial characters: prostomium longer than wide [character 7(0), Figs. 4A, 4F, 5B and 5D], as wide as long [character 7(1), Figs. 4B, 4D and 4G], and wider than long [character 7(2), Figs. 4C, 4H and 5E]. Members of three species within the outgroups (*Spio filicornis, Prionospio lighti* and *P. ehlersi*) are considered to possess prostomia which are longer than wide [character 7(0)], whilst members of *Phyllochaetopterus limicolus* and *Laonice cirrata* have prostomia which are wider than long [character 7(2)].

Magelonid prostomia carry paired "muscular", dorsal longitudinal ridges (**subject 8**), which extend from the base of the prostomium towards the distal tip [character 8(1)]. *Jones (1968)* stated that their "inner surfaces, particularly dorsally and ventrally, are provided with longitudinal muscles," describing them as hollow, cylindroid structures, which he presumed were fluid filled. The number of prostomial ridges (**subject 9**) varies from one pair [character 9(0), Fig. 4H], such as members of *Magelona equilamellae* or two pairs [character 9(1), Figs. 4A–4C, 4E and 5B], as observed among members of *M. pacifica* (Fig. 5A). The ridges may vary in size and distinctiveness, the outer pair among members of some species being more difficult to discern, *e.g.*, *M. alleni* (Fig. 4G), to the distinct and transversely ridged structures seen in, for example, members of *M. wilsoni* (Fig. 4C). Prostomial ridges are not present among members of the outgroups [character 8(0)] and thus subject 9 is not applicable.

### Characters of the palps (subjects 10–13)

Magelonids possess two long palps, considered to be peristomial, arising from the larval prototroch (*Wilson, 1982*). There has been much discussion about palp origin (**subject 10**),

**Table 3  Characters, as observation statements represented by subject-predicate relations, not 'characters' and 'states' (*Fitzhugh, 2006c*, *Fitzhugh, 2008c*), used in this study.**

1. Shovel-shape prostomium: (0) absent; (1) present.

2. Nuchal organs: (0) absent; (1) present.

3. Prostomium distal margin: (0) smooth; (1) crenulate; (2) medially indented.

4. Prostomium distal shape: (0) triangular; (1) rounded; (2) straight.

5. Prostomial horns (including rudimentary): (0) absent; (1) present.

6. Shape of prostomial horns [cf. character 5(1)]: (0) rudimentary; (1) distinct.

7. Prostomium dimensions: (0) longer than wide; (1) as wide as long; (2) wider than long.

8. Prostomial ridges: (0) absent; (1) present.

9. Number of prostomial ridges [cf. character 8(1)]: (0) a pair; (1) two pairs.

10. Palp origin: (0) dorsal; (1) ventral.

11. Palp surface: (0) non-papillate; (1) papillate.

12. Number of rows of proximal palp papillae [cf. character 11(1)]: (0) 2; (1) 4–8; (2) 10–14.

13. Number of rows of distal palp papillae [cf. character 11(1)]: (0) 2; (1) 4; (2) more.

14. Burrowing organ: (0) absent; (1) present.

15. Body regionation: (0) non-magelonid; (1) magelonid-like.

16. Number of anterior body region chaetigers [cf. character 15(1)]: (0) 8; (1) 9.

17. Notopodial postchaetal lamellae: (0) absent; (1) present.

18. Development of notopodia along thorax: (0) all similar; (1) different in some chaetigers.

19. Neuropodial postchaetal lamellae: (0) absent; (1) present.

20. Development of neuropodia along thorax: (0) all similar; (1) different in some chaetigers.

21. Chaetigers 1–7 position of notopodial lamellae relative to notochaetae: (0) postchaetal; (1) subchaetal.

22. Chaetigers 1–7 notopodial lamellae shape: (0) filiform; (1) foliaceous.

23. Chaetigers 1–7 margins of foliaceous notopodial lamellae [cf. character 22(1)]: (0) smooth; (1) crenulate; (2) bilobed.

24. Chaetigers 1–7 dorsal superior lobes: (0) absent; (1) present.

25. Superior dorsal lobes [cf. character 24(1)]: (0) absent on some chaetigers between 1–7; (1) present on chaetigers 1–7.

26. Chaetigers 1–7 neuropodial lamellae position relative to neurochaetae: (0) postchaetal; (1) subchaetal; (2) prechaetal.

27. Chaetigers 1–7 neuropodial subchaetal lamellae position relative to neurochaetae [cf. character 26(1)]: (0) same position along thorax; (1) varying in position along thorax.

28. Chaetigers 1–7 neuropodial lamellae shape: (0) filiform; (1) foliaceous.

29. Chaetigers 1–7 neuropodial filiform lamellae shape distal ends [cf. character 28(0)]: (0) pointed; (1) distally expanded/scoop shaped.

30. Chaetigers 1–7 lengths of noto- and neuropodial lamellae: (0) equivalent; (1) notopodial longer; (2) neuropodial longer.

31. Chaetigers 1–8 limbations of capillary chaetae: (0) unilimbate; (1) bilimbate.

32. Chaetigers 1–8 margins of bilimbate capillary chaetae [cf. character 31(1)]: (0) smooth; (1) irregular blade.

33. Chaetigers 1–8 lengths of noto- and neuropodial capillary chaetae: (0) equivalent; (1) notopodial longer; (2) neuropodial longer.

34. Chaetiger 8 notopodial lamellae position relative to notochaetae: (0) postchaetal; (1) subchaetal.

35. Chaetiger 8 notopodial lamellae shape: (0) filiform; (1) foliaceous.

**Table 3** (*continued*)

36. Chaetiger 8 margins of foliaceous notopodial lamellae [cf. character 35(1)]: (0) smooth; (1) crenulate; (2) bilobed.

37. Chaetiger 8 superior dorsal lobes: (0) absent; (1) present.

38. Chaetiger 8 neuropodial lamellae position: (0) postchaetal; (1) subchaetal; (2) prechaetal.

39. Chaetiger 8 neuropodia: (0) without additional postchaetal expansion; (1) with an additional postchaetal expansion.

40. Chaetiger 8 neuropodial lamellae shape: (0) filiform; (1) foliaceous.

41. Chaetiger 9 notopodial lamellae positions: (0) postchaetal; (1) subchaetal.

42. Chaetiger 9 notopodial lamellae height [cf. character 41(0)]: (0) low; (1) elongate.

43. Chaetiger 9 notopodial lamellae shape: (0) filiform; (1) foliaceous.

44. Chaetiger 9 notopodial foliaceous lamellae margins [cf. character 43(1)]: (0) smooth; (1) crenulate; (2) bilobed.

45. Chaetiger 9 superior dorsal lobes: (0) absent; (1) present.

46. Chaetiger 9 neuropodial lamellae position: (0) postchaetal; (1) subchaetal; (2) prechaetal.

47. Chaetiger 9 neuropodial lamellae height [cf. character 46(0)]: (0) low; (1) elongate.

48. Chaetiger 9 neuropodial lamellae shape: (0) filiform; (1) foliaceous.

49. Chaetiger 9 neuropodia: (0) without additional postchaetal expansion; (1) with additional postchaetal expansion.

50. Lengths of chaetiger 9 fascicles relative to chaetigers 1–8 fascicles: (0) same length as chaetae in chaetigers 1–8; (1) shorter than chaetae in chaetigers 1–8; (2) longer than chaetae in chaetigers 1–8.

51. Distal ends of chaetae in chaetiger 9: (0) gently tapered, similar to chaetigers 1–8; (1) mucronate; (2) pennoned.

52. Shape of thoracic interparapodial margins: (0) straight; (1) rounded, bulbous.

53. Dorso-lateral grooves, anterior chaetigers: (0) absent; (1) present.

54. Dimensions of post-chaetiger 9 segments: (0) as long as wide; (1) longer than wide.

55. Post-chaetiger 9 lamellae shape: (0) without basal constriction; (1) with basal constriction.

56. Post-chaetiger 9 dorsal and ventral medial lobes: (0) absent; (1) present.

57. Post-chaetiger 9 dorsal and ventral medial lobes sizes [cf. character 56(1) partim]: (0) smaller than hooks; (1) longer than hooks.

58. Post-chaetiger 9 postchaetal expansion behind chaetal rows: (0) absent; (1) present.

59. Relative sizes of post-chaetiger 9 noto- and neuropodial postchaetal lamellae: (0) same size; (1) different size.

60. Post-chaetiger 9 lateral pouches: (0) absent; (1) present.

61. Post-chaetiger 9 lateral pouch arrangement [cf. character 60(1)]: (0) paired; (1) unpaired; (2) both paired and unpaired.

62. Direction of lateral pouch openings [cf. character 60(1)]: (0) posteriorly; (1) posteriorly and anteriorly.

63. Post-chaetiger 9 posteriorly open lateral pouch arrangement [cf. character 62(0)]: (0) on consecutive segments; (1) on alternating segments; (2) on both consecutive and alternating segments.

64. Post-chaetiger 9 lateral pouches distribution [cf. character 60(1)]: (0) throughout most abdominal chaetigers; (1) median and posterior abdominal chaetigers only; (2) posterior abdomen only.

65. Margins of posteriorly-open pouches [cf. character 62(0)]: (0) smooth; (1) medially split.

66. Post-chaetiger 9 noto- and neuropodial hooded hooks: (0) absent; (1) present.

67. Post-chaetiger 9 noto- and neuropodial hooded hooks dentition [cf. character 66(1)]: (0) bidentate; (1) tridentate; (2) polydentate.

**Table 3** (*continued*)

68. Post-chaetiger 9 hooded hooks adjacent to notopodial subchaetal or neuropodial suprachaetal lamellae [cf. character 66(1)]: (0) same size; (1) smaller than rest.

69. Post-chaetiger 9 enlarged hooded hooks: (0) absent; (1) present.

70. Post-chaetiger 9 enlarged hooded hooks distal ends [cf. character 69(1)]: (0) recurved; (1) spine-like; (2) enlarged normal hook.

71. Post-chaetiger 9 number of hooded hooks per ramus [cf. character 66(1)]: (0) 8 or more; (1) less than 8.

72. Post-chaetiger 9 arrangement of hooded hooks [cf. character 66(1)]: (0) vis-à-vis; (1) vis-à- dos.

73. Neuropodial sabre chaetae: (0) absent; (1) present.

74. Post-chaetiger 9 noto- and neuropodial internal 'aciculae': (0) absent; (1) present.

75. Pigmentation in posterior thorax: (0) absent; (1) present.

76. Pigmentation pattern in posterior thorax [cf. character 75(1): (0) discreet band; (1) dispersed, not forming band.

77. Granular bodies on surfaces of segments: (0) absent; (1) present.

78. Granular bodies within segments as abdominal interparapodial patches: (0) absent; (1) present.

79. Tube construction: (0) absent; (1) present.

since they are ventrally inserted [character 10(1), Figs. 4F and 5G], rather than dorsally [character 10(0)], the latter of which is seen among members of all other polychaete groups with palps, including the outgroups. In addition, the palps do not have the characteristic feeding groove seen among members of other taxa such as spionids and *Myriowenia*. However, an inconspicuous ventral line devoid of papillae is generally present. According to *Orrhage (1966)*, palps among members of the Spionidae and Magelonidae are homologous because they are innervated by nerves from the same part of the brain. More recently *Beckers, Helm & Bartolomaeus (2019)* stated that two nerves innervate the palps of members of the Magelonidae, Spionidae, Chaetopteridae and Oweniidae, indicating the purported plesiomorphic condition in Annelida.

Magelonid palp surfaces (**subject 11**) are uniquely papillated, with ampulliform to digitiform papillae arranged in rows along the longitudinal axis [character 11(1), Figs. 4H, 5B and 5F–5H]. Members of the outgroups, along with all other polychaetes possess non-papillated palps [character 11(0)]. Although the number of rows of papillae varies among members of magelonid species, it is relatively stable among members of any particular species. As papillae are involved in obtaining food (*Jones, 1968*; *Fauchald & Jumars, 1979*; *Mortimer & Mackie, 2014*), variations in number of papillae may reflect different feeding strategies. In general, the number of rows of papillae are fewer distally in comparison to proximally (*Jones, 1963*; *Jones, 1971*; *Jones, 1978*). The number of rows of papillae in the proximal region of the palp (**subject 12**) were coded as three characters: two rows of papillae [character 12(0)], 4–8 rows [character 12(1)], and 10–14 rows [character 12(2)]. The number of rows of papillae at the distal end (**subject 13**) were coded for 2 [character 13(0)], 4 [character 13(1)], or more than 4 rows [character 13(2)]. Where a range was observed among members of a species, the maximum number of rows was the value used. As with palps in other polychaete groups, those of magelonids are easily lost upon collection. Many descriptions do not include information about them, whilst others do not detail differences in number of papillae along the palp length. Many authors

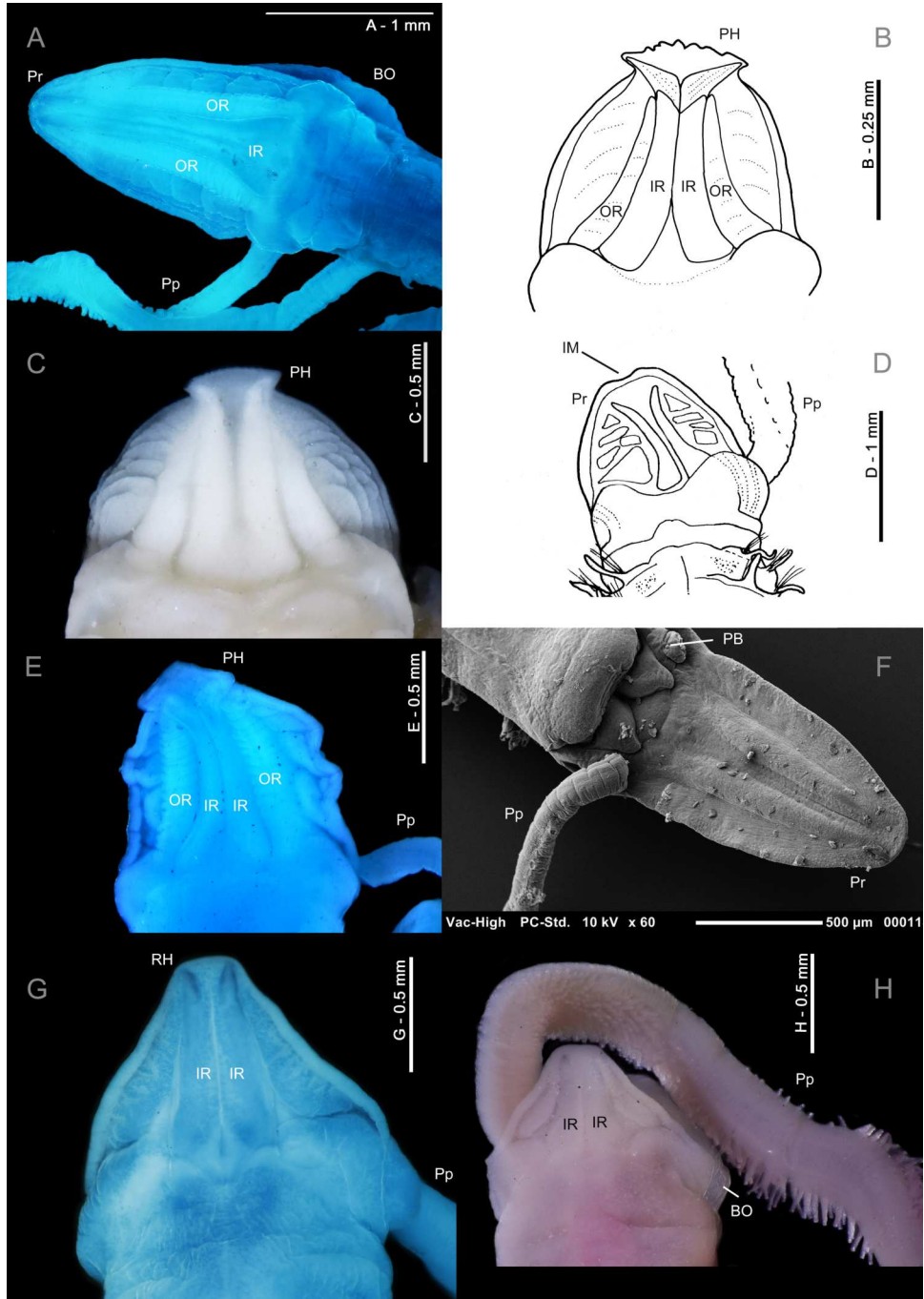

**Figure 4** **Magelonid prostomial characters.** (A) smooth distal margin of *Magelona obockensis* (Kuwait, BMNH:ZB. 1971.54; photo by James Turner); (B) distinctly crenulate distal margin of *M. crenulifrons* from Hong Kong (NMW.Z.2007.033.0001); (C) minutely crenulate margin of *M. wilsoni* from the Gulf of Lions (NMW.Z.2010.010.0008); (D) medially indented distal margin of *M. symmetrica* from the Seychelles, holotype, from Mortimer & Mackie, 2006); (E) triangular distal margin of *M. sinbadi* from Iran (holotype, NMW.Z.2010.037.0001); (F) rounded distal margin of *M. johnstoni* from Berwick-upon-Tweed (NMW.Z.2013.037.0008); (G–H) straight distal margins (continued on next page...)

**Figure 4 (…continued)**
(often termed 'rudimentary horns') of *M. alleni* from Morocco (NMW.Z.2021.001.0001) and *M. equil-amellae* from Ebro Delta, Catalonia. A–E, G, H, dorsal views; F, ventral view; A, E, G, stained with methyl green; H, stained with Rose Bengal. BO = burrowing organ, IM = indented margin, IR = inner ridges, OR = outer ridges, PB = palp base, PH = prostomial horns, Pp = palp, Pr = prostomium, RH = rudimentary horns.

have undervalued the importance of palp characteristics in the descriptions of members of species. These two factors present the largest problem in coding these subjects. It should be noted that the number of rows of papillae may vary in animals with regenerating palps, particularly at the distal tips, thus affecting the ranges observed. Subjects 12 and 13 are not applicable to members of the outgroups since papillated palps are unique to magelonids.

### Burrowing organ (subject 14)

Magelonids have long been described to possess a ventral eversible proboscis. A heart-shaped sac (when fully everted; oval when only partially everted), carrying longitudinal ridges (*Mortimer, 2019*: fig. 4.2.4). However, recent studies show that this structure plays no part in feeding (and is not connected to the buccal region) but functions in burrowing only (*Mortimer, 2019*). The term "burrowing organ" should be applied to this structure. The burrowing organ is present in members of all known magelonid species [character 14(1)] but not present in any members of the outgroups [character 14(0)].

### Characters of body regionation (subjects 15–16)

The body of magelonids (**subject 15**) is divided into an anterior region comprising the head and "thorax" (the latter of which is characterised by the presence of capillary chaetae), and an "abdomen" comprising many segments with hooded hooks [character 15(1)] (Fig. 6A). As an aside, the terms thorax and abdomen are ill-defined when applied to polychaetes overall, other than to vaguely indicate anterior-posterior distinctions along the body. Unfortunately, the characters associated with these regions are not uniformly applied across all polychaetes.

The body of members of the outgroups do not have this regionation [character 15(0)]. Behind the head region (**subject 16**), the thorax carries an achaetous first segment, followed by either eight [character 16(0), *Octomagelona*; Fig. 6] or nine [character 16(1), *Magelona*] chaetigers. Subject 16 does not apply to members of the outgroups.

### Characters of thoracic parapodia (subjects 17–30)

Unique terminology introduced and later modified by *Jones (1963)*, *Jones (1971)* and *Jones (1978)* has been applied to the family, particularly related to parapodial structures (Fig. 7). *Brasil* (*2003*: fig. 4) provided a comprehensive review of this terminology and problems related to inconsistencies in applying terms. Consequently, several authors have suggested its abandonment (*Rouse, 2001*; *Mortimer & Mackie, 2003*). Magelonid chaetigers possess biramous parapodia with lamellae in both the noto- and the neuropodia. Lamellae are generally postchaetal, flattened structures that range from filiform to foliaceous in shape. They encircle the chaetal bundle, attaching to a low, triangular or rounded prechaetal lamella, almost cuff-like (Fig. 8A). However, among members of some species,

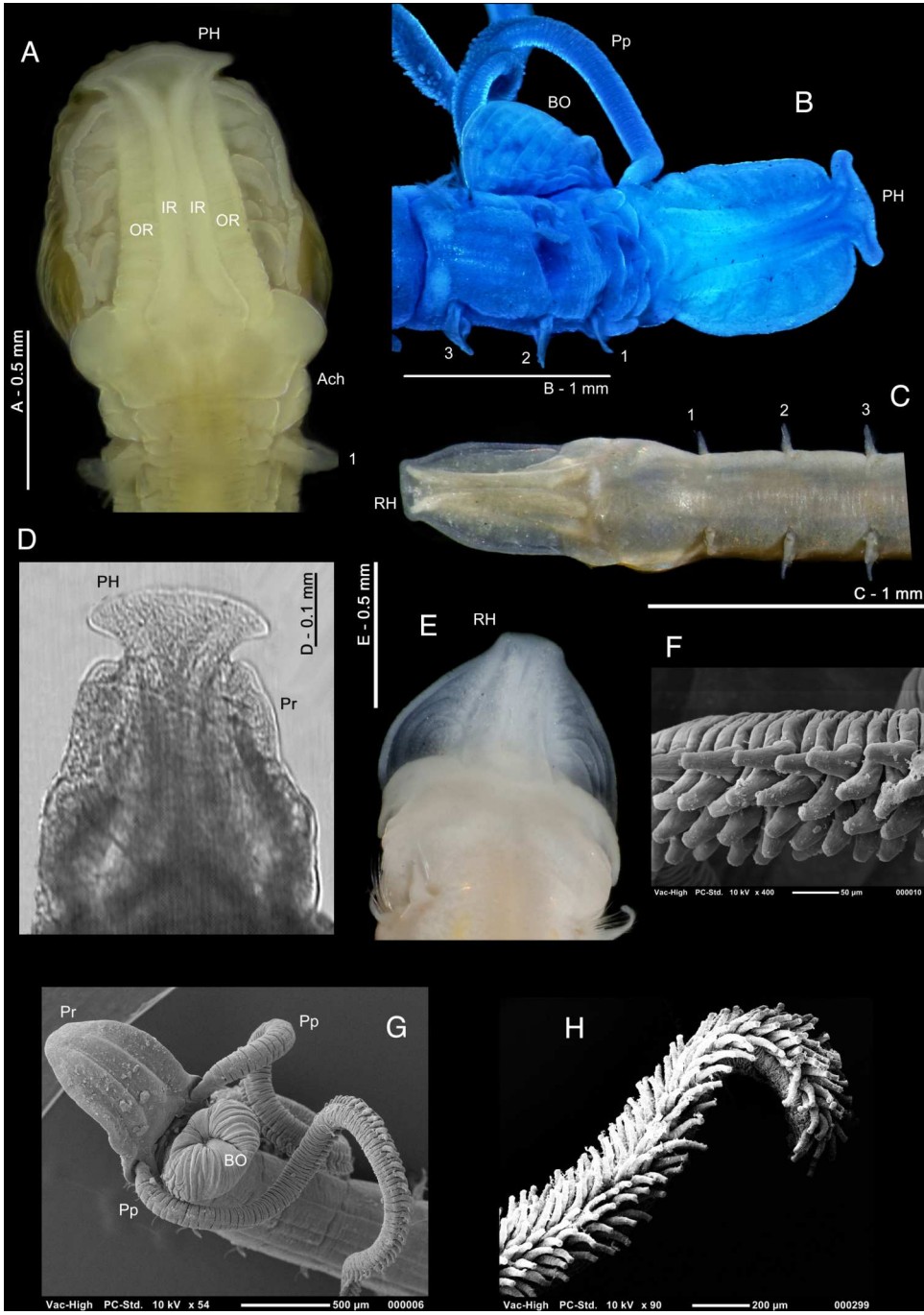

**Figure 5 Magelonid anterior end and palp characters.** (A) prostomium and first chaetiger of *Magelona pacifica* from Panamá (syntype, BMNH Type 1933. 7.10.65/70, dorsal view); (B) prostomium and first three chaetigers of *M. montera* from Gulf of Aqaba (holotype MNHN TYPE 1548, dorso-lateral view, stained with methyl green, photo by James Turner); (C) prostomium and first three chaetigers of *M. filiformis* from Salcombe, England (paratype, BMNH 1959.4.2.6–10, dorso-lateral view); (D) prostomium of *M. posterelongata* from Paranaguá Bay, Brazil (dorsal view); (continued on next page...)

**Figure 5 (…continued)**
(E) prostomium and first chaetiger of *Magelona nanseni* from Nigeria (Holotype, ZMBN132141, dorsal view); (F) mid-palp region of *M. johnstoni* from Berwick-upon-Tweed (NMW.Z.2013.037.0008c); (G) prostomium and anterior thorax of *M. johnstoni* from Berwick-upon-Tweed, showing ventrally inserted papillate palps, and partially everted burrowing organ (NMW.Z.2013.037.0008b); (H) distal palp end of *M. alleni* from Swansea (NMW.Z.2012.022.0001, SEM by Kimberley Mills). Ach = achaetous first segment, BO = burrowing organ, IR = inner ridges, OR = outer ridges, PH = prostomial horns, Pp = palp, Pr = prostomium, RH = rudimentary horns. Numbers indicate chaetiger.

the lamellae are somewhat subchaetal in position, underneath the chaetal bundle and appearing somewhat U-shaped when viewed laterally (Figs. 8B, 8D). All members of all species in the ingroup and outgroups have postchaetal noto- and neuropodial lamellae in thoracic/anterior chaetigers (**subjects 17** and **19**, respectively) except for *Phyllochaetopterus limicolus*, in which they are absent [characters 17(0) and 19(0)].

The development of notopodial (**subject 18**) and neuropodial lamellae (**subject 20**) along the anterior chaetigers may be similar in all chaetigers [characters 18(0) and 20(0), respectively], such as among members of *Magelona papillicornis*, and *M. minuta* (Fig. 7A), or vary along the thorax [characters 18(1) and 20(1), respectively]. Subjects 18 and 20 are considered inapplicable to members of all species in the outgroups. The variation in degree of development can be easily observed among members of *M. obockensis* (Fig. 7C), and *M. conversa* (Fig. 7D), in both the noto- and neuropodia, whilst among members of *Octomagelona bizkaiensis* it is evident only in the neuropodia (Fig. 6E). The lamellae of chaetigers eight and nine often vary in comparison to those of the first seven chaetigers, *e.g.*, *M. crenulifrons*, and *M. johnstoni* (Fig. 9I). Thus, characters of the lamellae of the first seven chaetigers were coded separately (**subjects 21–30**) to those for chaetiger eight (**subjects 34–40**), and chaetiger nine (**subjects 41–49**).

The notopodial lamellae in chaetigers 1–7 (**subject 21**) may be postchaetal [character 21(0); noted also for members of the outgroups, except *Phyllochaetopterus limicolus*, for which the subject is not applicable] or subchaetal [character 21(1)], whilst neuropodial lamellae (**subject 26**) may be postchaetal [character 26(0); observed for members of the outgroups, except *P. limicolus*, for which the subject is not applicable], subchaetal [character 26(1)] or prechaetal [character 26(2)]. Prechaetal neuropodial lamellae, such as are observed among members of *Magelona conversa* and *M. mirabilis* (Figs. 7D and 8C), are less frequent than both post- (Fig. 7A) or subchaetal (Fig. 9E). The position of the thoracic neuropodial lamellae (**subject 27**) may be consistent throughout the thorax [character 27(0)], or may vary [character 27(1)], *e.g.*, being prechaetal on chaetiger 1, occurring in a ventral position in the mid-thorax, and distinctly postchaetal in the posterior thorax (Figs. 8F and 8G). This subject was not considered applicable to members of the outgroups.

Thoracic notopodial lamellar shape (**subject 22**) may be filiform, having a long, tapering structure [character 22(0), Figs. 9A, 9B and 9G] or more foliaceous [character 22(1), Figs. 9C–9F]. The margins of the latter (**subject 23**) may be smooth [character 23(0)], crenulate [character 23(1), *e.g.*, *M. montera*, Figs. 9D and 9E] or bilobed [character 23(2), *e.g.*, *M. obockensis*, Fig. 9F]. All members of the outgroups are considered to have foliaceous [character 22(1)], smooth-edged [character 23(0)] anterior notopodia, except for members

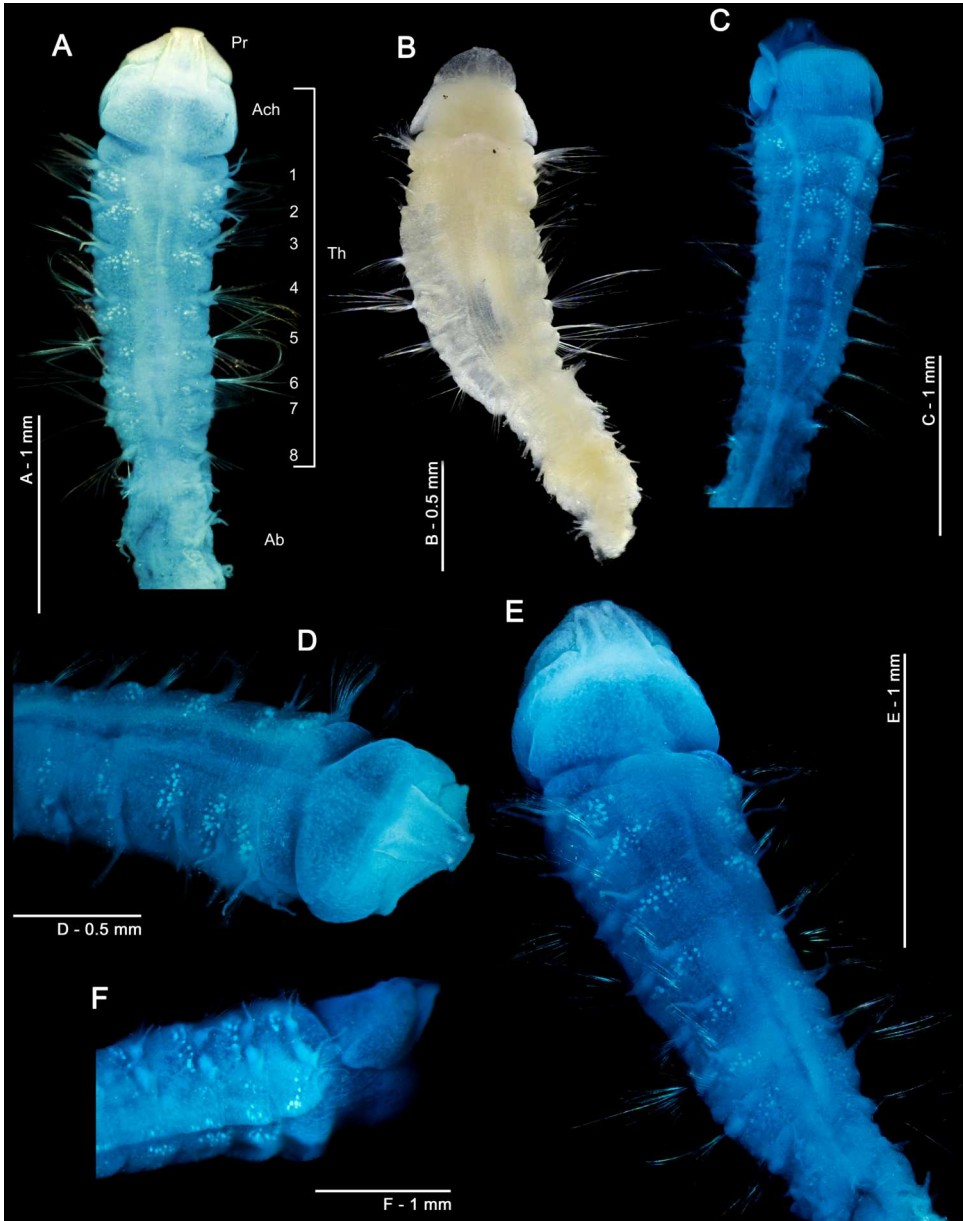

**Figure 6** **_Octomagelona._** _Octomagelona_: (A, C–F) anterior of an undescribed species from West Africa (dorsal, ventral, antero-dorsal, postero-dorsal and lateral views respectively); (B) _Octomagelona bizkaiensis_ from Capbreton Canyon (holotype, MNCN 16. 01/6887, dorsal view). Ab, abdomen; Ach, achaetous first segment; Pr, prostomium; Th, thorax.

of _Phyllochaetopterus limicolus_ for which subjects 22 and 23 are not applicable. Above the notopodial lamellae and in a slightly prechaetal position, superior dorsal lobes (**subjects 24, 25, 37, 45**) (dorsal medial lobes _sensu_ Jones) may be present (Figs. 7C, 7D, 8D, 8E and 9C–9F) or absent [characters 24(0), 37(0), 45(0)]. These structures are generally smaller than the notopodial lamellae and somewhat cirriform to digitiform in shape. However, among members of some species they may be larger and distinctly foliaceous. When

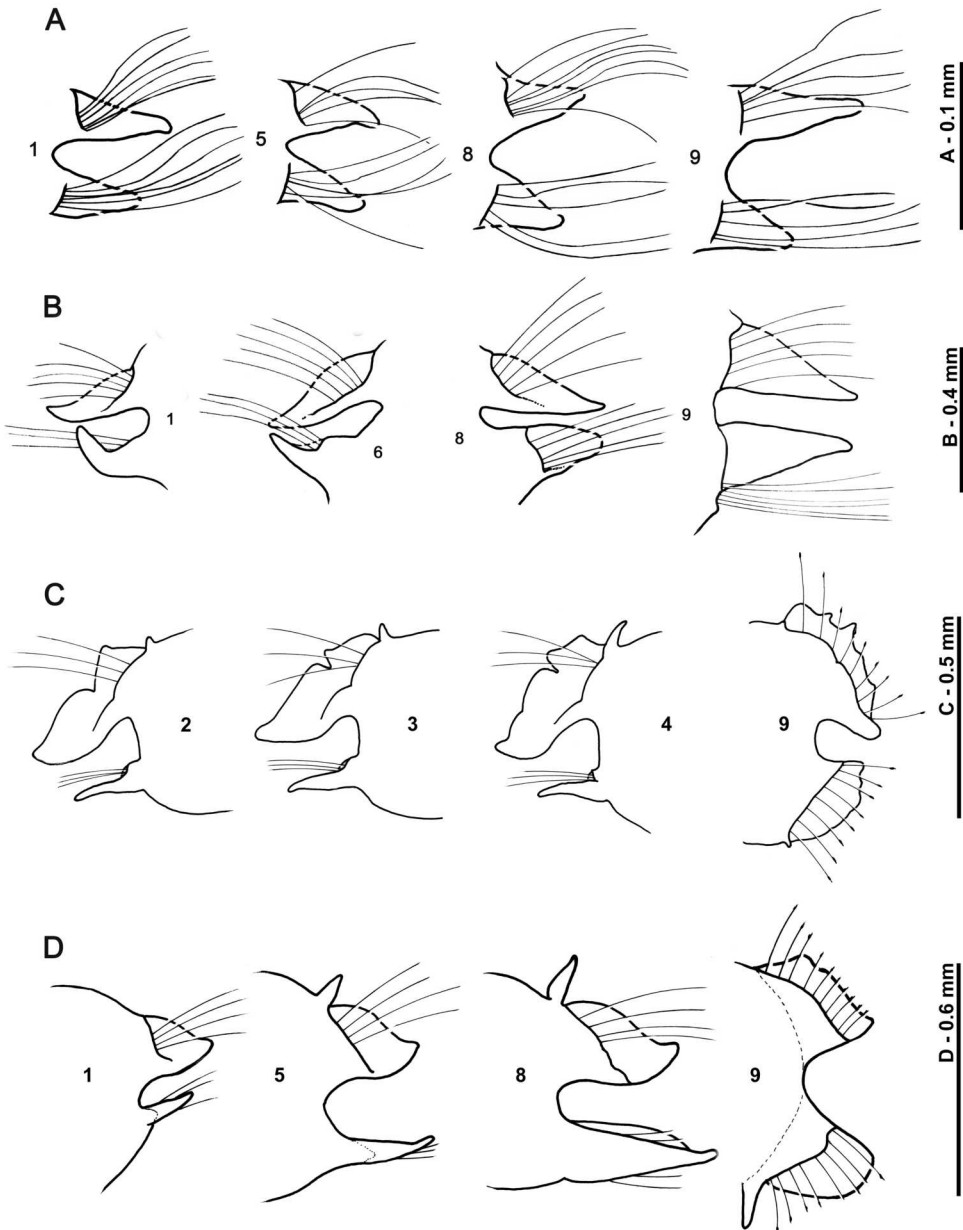

**Figure 7** **Development of magelonid thoracic parapodia.** (A) left-hand parapodia of chaetigers 1, 5, 8 and 9 of *Magelona minuta* from Öresund, Sweden (USNM 52510, anterior views); (B) parapodia of chaetigers 1, 6, 8 and 9 of *M. equilamellae* from France (syntype, SMF 4675, anterior views); (C) parapodia of chaetigers 2, 3, 4 and 9 of *M. obockensis* from Obock, Red Sea (syntype, MNHN Type 1357, anterior views); (D) left-hand parapodia of chaetigers 1, 5, 8 and 9 of *M. conversa* from Iran, NMW.Z.2010.037.0007, anterior views).

present, superior dorsal lobes may be present on all thoracic chaetigers [characters 25(1), 37(1), 45(1)], on chaetigers 1–8 only [characters 25(1), 37(1), 45(0)], or only occurring on a small number of thoracic chaetigers [often starting in the mid-thoracic region, *e.g.*, *M. johnstoni* or *M. conversa*, character 25(0), Fig. 7D]. Superior dorsal lobes are absent among
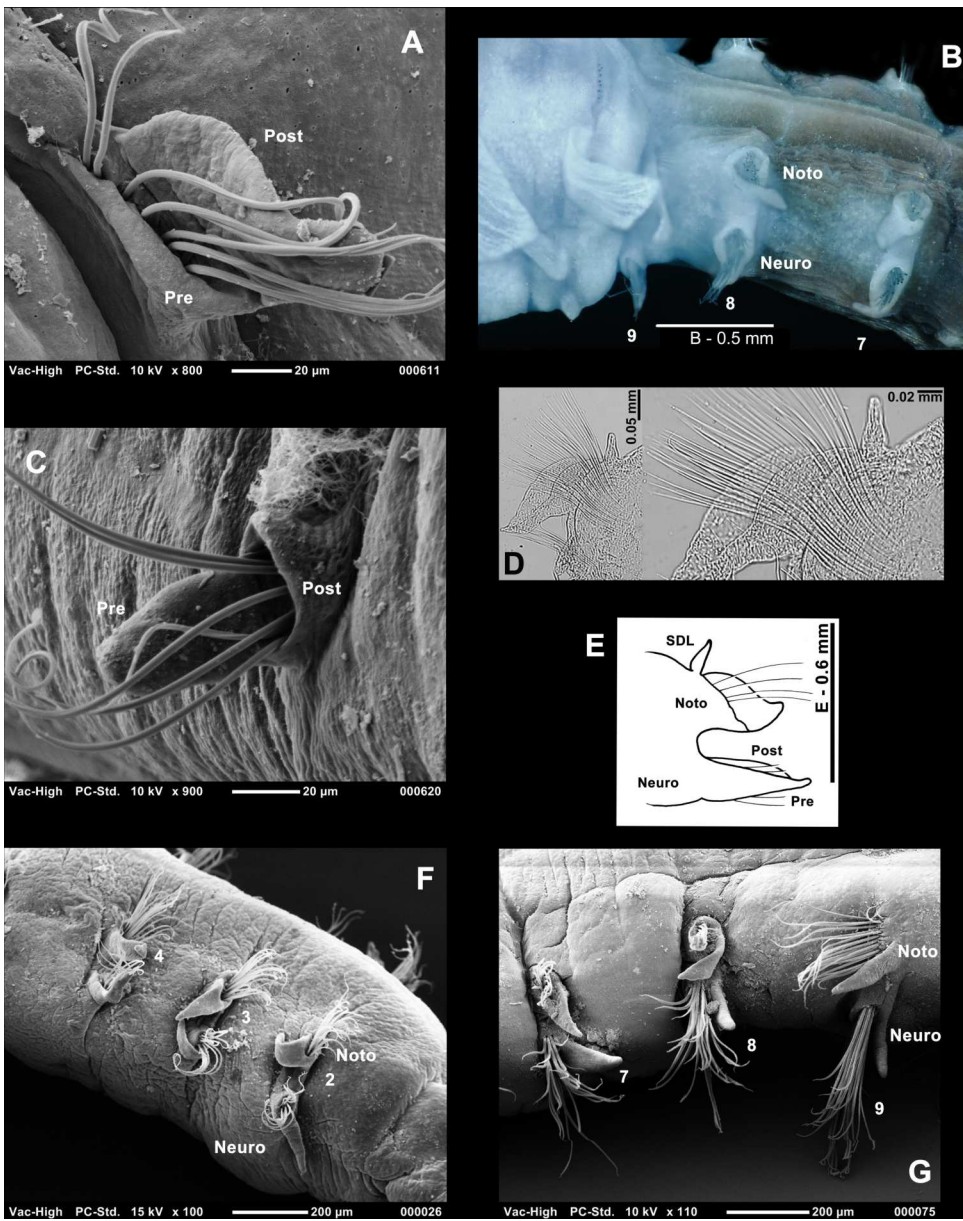

**Figure 8** **Magelonid thoracic parapodia.** (A) Thoracic postchaetal notopodial lamellae of *Magelona mirabilis* from Berwick-upon-Tweed (anterior view, SEM by Kimberley Mills); (B) chaetigers 7–10 of *M. alleni* from Morocco (NMW.Z.2021.001.0001, lateral view, stained with methyl green); (C) thoracic prechaetal neuropodial lamellae of *M. mirabilis* from Berwick-upon-Tweed (posterior view, SEM by Kimberley Mills); (D) subchaetal notopodial lamellae of *M. riojai* from Corora Grande, Rio de Janeiro, Brazil; (E) left-hand parapodia of chaetiger 8 of *M. conversa* from Iran, (NMW.Z.2010.037.0007, anterior view); (F) right-hand parapodia of chaetigers 2–4 of *M. equilamellae* from Ebro Delta, Catalonia (NMW.Z.2019.100.0012–14, lateral view, SEM by Kimberley Mills); (G) left-hand parapodia of chaetigers 7–9 of same specimen (SEM by Kimberley Mills). Noto, notopodial; Neuro, neuropodia; Pre, prechaetal; Post, postchaetal; SDL, superior dorsal lobe. Numbers indicate chaetiger.

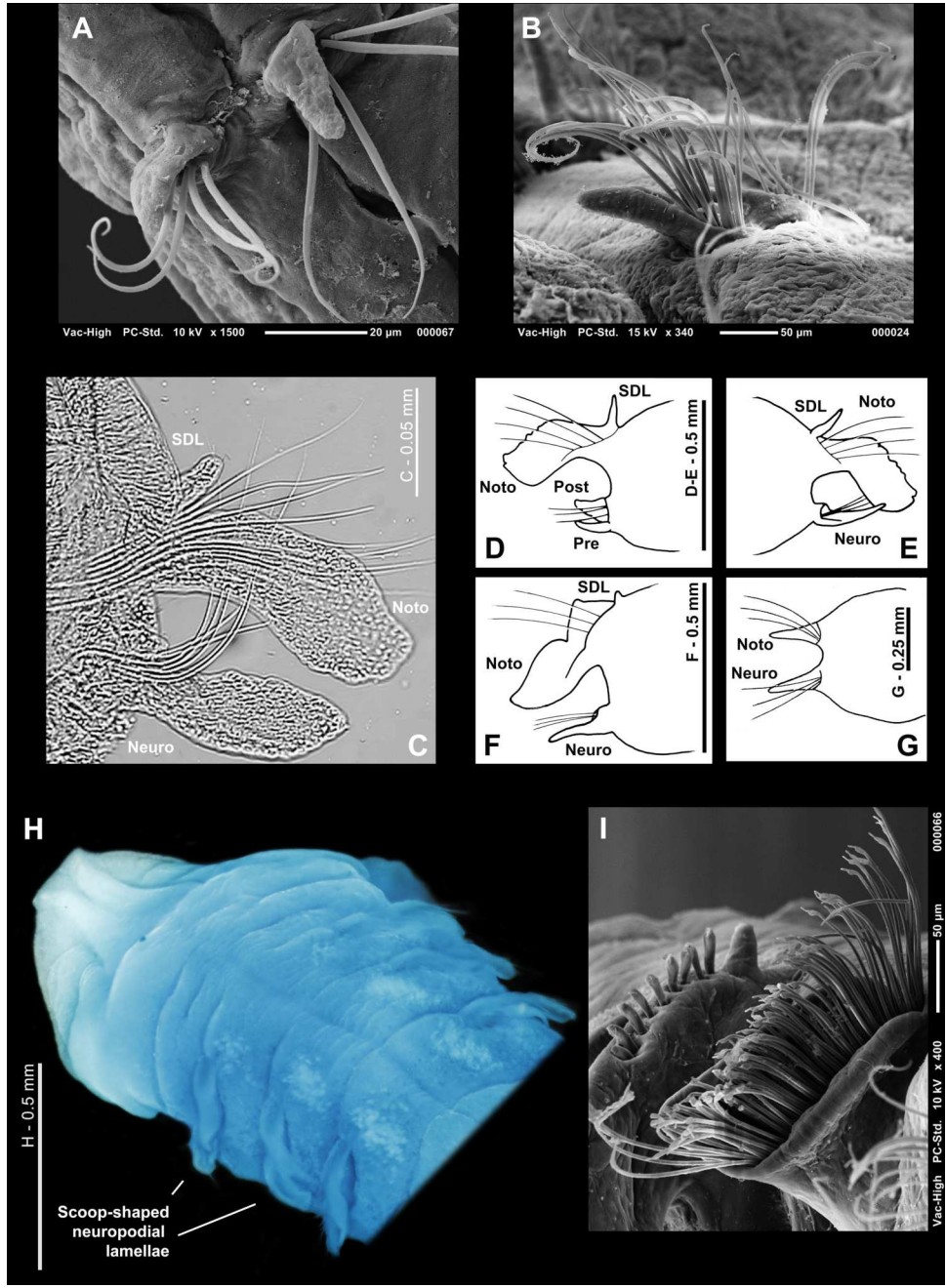

**Figure 9  Magelonid thoracic parapodia.** (A) Right-hand parapodia of chaetiger 4 of *Magelona fauchaldi* from Iran (paratype, NMW.Z.2015.012.0002b, antero-lateral view); (B) right-hand notopodia of *M. equil-amellae* from Ebro Delta, Catalonia (NMW.Z.2019.100.0012–14, anterior view, SEM by Kimberley Mills); (C) foliaceous thoracic lamellae of *M.* sp. 2 from the Brazilian coast (from *Brasil, 2003*, anterior view); (D–E) parapodia of chaetigers 8 and 4 of *M. montera* respectively from Gulf of Aqaba (holotype, MNHN TYPE 1548, anterior views); (F) right-hand parapodia of (continued on next page...)

**Figure 9 (…continued)**
chaetiger 2 of *M. obockensis* from Obock, Red Sea (syntype, MNHN Type 1357, anterior view); (G) right-hand parapodia of chaetiger 8 of *M. symmetrica* from the Seychelles (holotype, NMW.Z.2000.020.0175, anterior view); (H) distally expanded and scoop-shaped neuropodial lamellae of chaetigers 1–3 of *Magelona* sp. cf. *M. cincta* from Iran, NMW.Z.2015.012.0004, posterolateral view, stained with methyl green; (I) notopodia of chaetigers 9 and 10 of *M. johnstoni* from Berwick-upon-Tweed (anterior view). Noto, notopodia; Neuro, neuropodia; Pre, prechaetal; Post, postchaetal; SDL, superior dorsal lobe.

all members of the outgroups on chaetigers 1–9 (thus subject 25 is not applicable), and subjects 24, 25, 37 and 45 are considered inapplicable for *Phyllochaetopterus limicolus*.

The shape of thoracic neuropodial lamellae (**subject 28**) may be filiform [character 28(0), Figs. 7C, 8F and 8G] or foliaceous [character 28(1), Fig. 9C, also seen in members of the outgroups, excluding *Phyllochaetopterus limicolus*, for which the subject is not applicable]. The distal ends (**subject 29**) of which may be pointed [character 29(0)], as among members of *M. equilamellae* (Figs. 8F and 8G), or distally expanded and scoop-shaped [character 29(1)] as observed among members of *M. cincta* (Fig. 9H). The latter subject is not applicable to members of the outgroups. The lengths of noto- and neuropodial lamellae (**subject 30**) may be equivalent [character 30(0), Fig. 9G], the notopodial lamellae may be longer [character 30(1), Figs. 9D–9F], or the neuropodial lamellae may be longer [character 30(2), Fig. 7D], *e.g.*, chaetiger 8 of *M. conversa*. The thoracic notopodial lamellae among members of the outgroups, *Prionospio lighti* and *P. ehlersi* are longer than neuropodial lamellae [character 30(1)], whilst for members of *Spio filicornis* and *Laonice cirrata* they are equivalent [character 30(0)]. Subject 30 is not applicable to *P. limicolus*.

### Characters of the chaetae from chaetigers 1–8 (subjects 31–33)

Capillary chaetae (**subject 31**) occur only in the thorax of magelonids (Figs. 6, 8, 9, 10 and 11), but many authors have undervalued the differences between them, describing them simply as limbate capillary chaetae (*Hartman, 1944*; *Hartman, 1961*; *Hartman, 1965*; *Nateewathana & Hylleberg, 1991*). *Jones (1977)* was one of the few authors who referred to the presence of unilimbate [character 31(0), Fig. 10C] and bilimbate capillary chaetae [character 31(1)] in descriptions of members of species. For some individuals, both unilimbate and bilimbate chaetae have been recorded, however, the predominate form in each case was coded in the current analysis. *Bolívar & Lana (1986)* and *Blake (1996c)* additionally noted some members of species possess bilimbate chaetae (**subject 32**) with an irregular blade [character 32(1)], such as *Magelona variolamellata* (Fig. 10E), and *Brasil (2003)* additionally noted it to occur among members of species such as *M. riojai* (Fig. 10D), *M. pacifica* and *M. pitelkai*. Members of the outgroups possess unilimbate capillary chaetae [character 31(0)], whilst subject 32 is not applicable. Additional to the limbations of capillary chaetae, the relative lengths of noto- and neurochaetae may vary (**subject 33**). Either being equivalent in length [character 33(0)] *e.g.*, *M. nonatoi*, longer in the notopodia [character 33(1)], *e.g.*, *M. sacculata*, or longer in the neuropodia [character 33(2)], *e.g.*, *M. variolamellata*. For members of *Phyllochaetopterus limicolus*, *Spio filicornis* and *Prionospio lighti*, chaetae are longer in the notopodia [character 33(1)], whilst for members of *Laonice cirrata* and *P. ehlersi* they are equivalent in length [character 33(0)]. In some members

of magelonid species chaetae of all thoracic chaetigers are similar, however, in others the chaetae of the ninth chaetiger are modified. For this reason, the chaetae were coded collectively for the first eight chaetigers, and separately for chaetiger nine (see below).

### Characters of parapodia of chaetiger 8 (subjects 34–40)

As described above, development of lamellae in the magelonid thorax may be similar on all chaetigers or vary between them. For members of species in which they vary, the parapodia of chaetigers 8 and 9 in particular may show larger differences in comparison to preceding chaetigers. However, the same characters that were coded for the first seven chaetigers can be applied to the posterior-most thoracic chaetigers. The notopodial lamellae (**subjects 34–36**) of chaetiger 8 may be postchaetal [character 34(0)] or subchaetal [character 34(1)] in position relative to the chaetal bundle, filiform [character 35(0)] or foliaceous [character 35(1)] in shape, with smooth [character 36(0)], crenulate [character 36(1)] or bilobed [character 36(2)] margins. Subjects 34–36 are considered not applicable to members of *Phyllochaetopterus limicolus.* However, remaining members of the outgroups possess postchaetal [character 34(0)], foliaceous [character 35(1)] parapodia with smooth margins [character 36(0)]. Superior dorsal lobes (**subject 37**) may be absent [character 37(0)] or present [character 37(1)]. The neuropodial lamellae (**subjects 38–40**) of chaetiger 8 may be postchaetal [character 38(0)], subchaetal [character 38(1)] or prechaetal [character 38(2)] in position relative to the chaetal bundle, and may be filiform [character 40(0)] or foliaceous [character 40(1)] in shape. However, an additional postchaetal expansion (**subject 39**), often triangular in shape, as is seen among members of *Magelona montera* (Fig. 9D) is sometimes observed in the neuropodia of chaetiger 8 (and 9, see below) [character 39(1)]. Subjects 38–40 are considered not applicable to members of *Phyllochaetopterus limicolus.* Members of *Spio filicornis, Prionospio lighti, P. ehlersi* and *Laonice cirrata* have postchaetal [character 38(0)], foliaceous [character 40(1)] neuropodia in chaetiger 8, without an additional postchaetal expansion [character 39(0)].

### Characters of the parapodia of chaetiger 9 (subjects 41–49)

For chaetiger 9, observed characters were coded in the same way as for chaetiger 8. The notopodial lamellae (**subjects 41, 43–44**) of chaetiger 9 may be postchaetal [character 41(0)], or subchaetal [character 41(1)] in position in comparison to the notochaetae, and may be filiform [character 43(0)] or foliaceous [character 43(1)] in shape, with smooth [character 44(0)], crenulate [character 44(1)] or bilobed [character 44(2)] margins. Members of the outgroups possess postchaetal [character 41(0)] *e.g.*, notopodia of chaetiger 99 which are foliaceous [character 43(1)], with smooth margins [character 44(0)]. However, subjects 41, 43 and 44 are not applicable to *Phyllochaetopterus limicolus.* Superior dorsal lobes (**subject 45**) may be present [character 45(1)] or absent [character 45(0)].

The neuropodial lamellae (**subjects 46, 48**) of chaetiger 9 may be postchaetal [character 46(0)], subchaetal [character 46(1)] or prechaetal [character 46(2)] in position in comparison to the chaetal bundle, and filiform [character 48(0)] or foliaceous [character 48(1)] in shape. As with chaetiger 8, an additional postchaetal expansion (**subject 49**) may be absent [character 49(0)] or present [character 49(1)]. Members of the outgroups possess foliaceous [character 48(1)], postchaetal [character 46(0)] lamellae without an additional

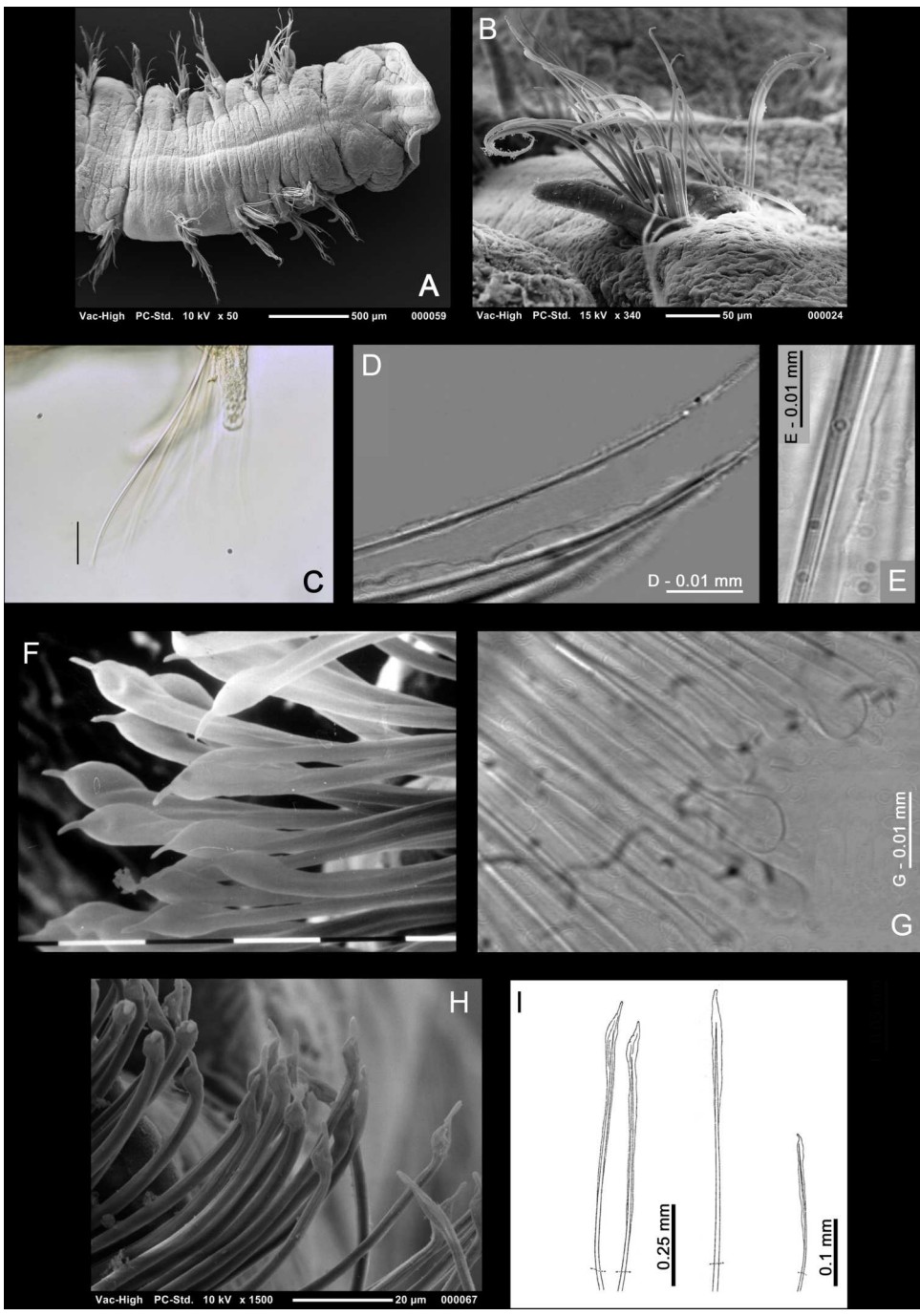

**Figure 10 Magelonid thoracic chaetae.** (A) First five chaetigers, showing thoracic capillary chaetae in noto- and neuropodia, of *Magelona equilamellae* from Ebro Delta, Catalonia; dorsal view. SEM by Kimberley Mills; (B) right-hand notopodia of chaetiger 5 (anterior view) of *M. equilamellae* from Ebro Delta, Catalonia, SEM by Kimberley Mills; (C) unilimbate capillary chaetae of *M. minuta* from Öresund, Sweden (USNM 52510); (D) bilimbate capillary chaetae with irregular blades from chaetiger 4 of *M. riojai* from Coroa Grande, Rio de Janeiro, Brazil; (E) bilimbate capillary chaeta (continued on next page...)

**Figure 10 (…continued)**
with irregular blade from chaetiger 2 of *M. variolamellata* from Paranaguá Bay, Brazil; (F–G) mucronate chaetae from chaetiger 9 of *M. riojai* from Coroa Grande, Rio de Janeiro, Brazil; (H) the same from *M. johnstoni* from Berwick-upon-Tweed (NMW.Z.2013.037.0011b); (I) pennoned chaetae from chaetiger 9 of *M. pitelkai, M. hobsonae, M. hartmanae* (respectively), modified from *Jones (1978)*.

postchaetal expansion [character 49(0)], except for *P. limicolus* for which these subjects are not applicable.

The parapodia in chaetiger 9 among members of some magelonid species may be markedly different in comparison to preceding chaetigers. For example, the lamellae in chaetiger 9 among members of *Magelona johnstoni* (Fig. 9I) are much broader and lower [characters 42(0) and 47(0)] in comparison to the elongate lamellae of chaetigers 1–8 in both the noto- and neuropodia, and the elongate lamellae of chaetiger 9 amongst members of other species, *e.g.*, *M. equilamellae* [characters 42(1) and 47(1), Fig. 7B]. For this reason, additional subjects were added to accommodate the height of lamellae in both rami of chaetiger 9 (**subjects 42 and 47**). As members of *Octomagelona* species only possess eight thoracic chaetigers these subjects were coded as inapplicable for these taxa and additionally for members of *Phyllochaetopterus limicolus*. Members of *Spio filicornis* possess broad and low parapodia in chaetiger 9 [characters 42(0) and 47(0)], whilst for members of *Prionospio lighti, P. ehlersi* and *Laonice cirrata* they are more elongate [characters 42(1) and 47(1)].

### Characters of the chaetae of chaetiger 9 (subjects 50–51)

The chaetae of chaetiger 9 in magelonids may be similar to or vary from those of chaetigers 1–8 (**subjects 50 and 51**). They may be the same length [character 50(0), as in the outgroups], shorter than [character 50 (1)], or longer than those of chaetigers 1–8 [character 50(2)]. The distal ends of chaetiger 9 chaetae may be gently tapered as is seen in the preceding chaetigers [character 51(0), Figs. 8F and 8G] (observed also for members of the outgroups), or have distinctly mammiform, expanded, mucronate tips [character 51(1), Figs. 7C, 7D, 9I and 10F–10H] such as among members of *Magelona johnstoni* and *M. mirabilis*. In members of other species, the distal ends are pennoned [character 51(2), Fig. 10I], such as *M. pitelkai, M. hobsonae*, and *M. hartmanae*.

### Characters of the thoracic region (subjects 52–53)

Whilst many authors have noted the often-marked difference between thoracic and abdominal regions in magelonids (Fig. 8B), other characteristics of the thoracic region have been largely overlooked. *Mortimer (2019)* noted that among members of certain magelonid species, the thoracic interparapodial margins (**subject 52**) may be characteristically bulbous and rounded [character 52 (1), Fig. 11A], as seen among members of *Magelona alleni*, whilst among members of others, such as *M. johnstoni*, they are straight [character 52(0), Fig. 11B]. Members of the outgroups possess straight thoracic interparapodial margins [character 52(0)].

*Uebelacker & Jones (1984)* first coined the phrase "oblique lateral slits" (**subject 53**) in relation to members of their undescribed species *Magelona* sp. I from the Gulf of Mexico. Later, *Bolívar & Lana (1986)* described "sulcos latero-dorsais" (dorso-lateral slits) and

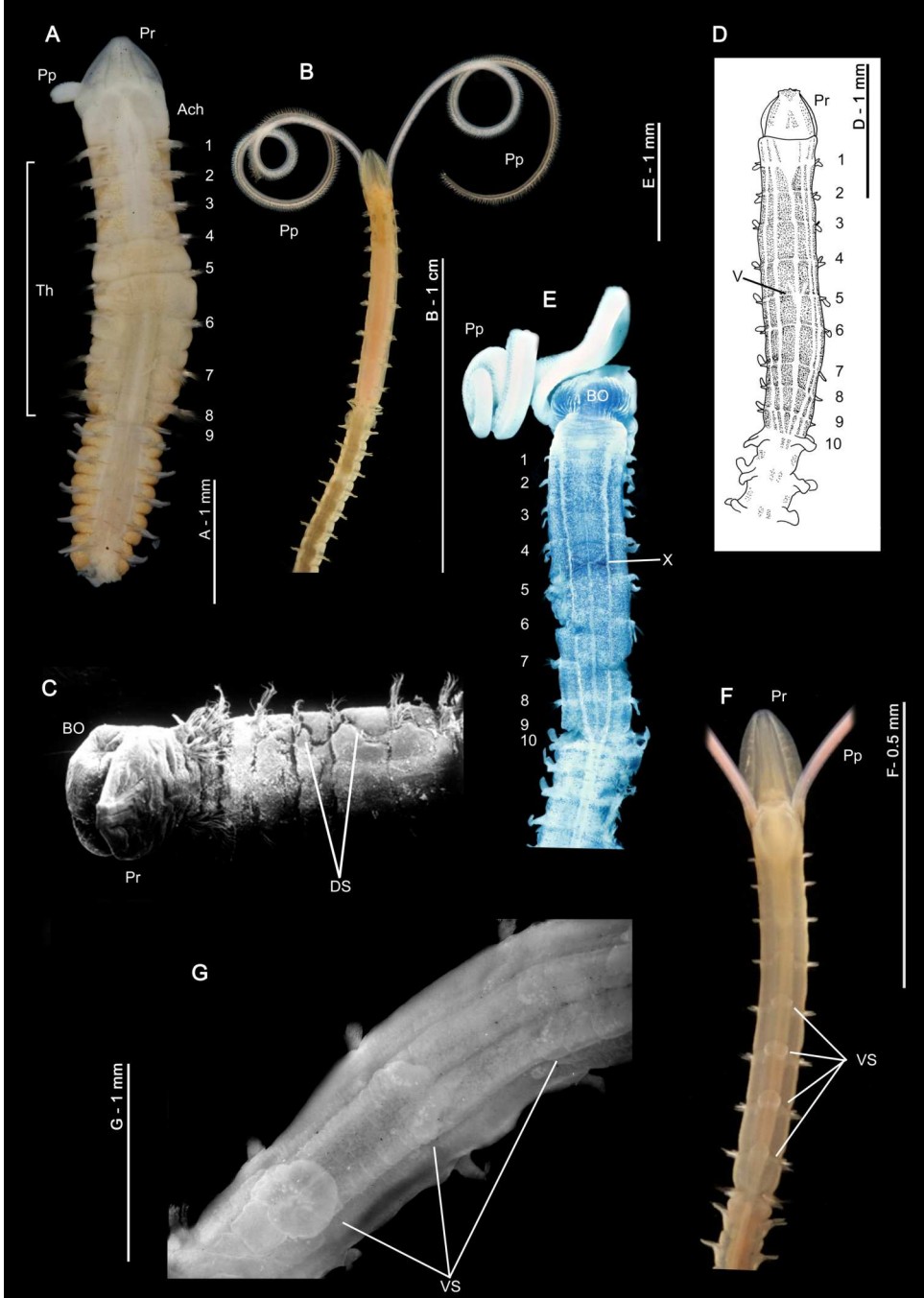

**Figure 11 Magelonid thoracic region.** (A) Characteristically bulbous and rounded thoracic chaetigers of *Magelona alleni* from Rame Head, England (holotype, BMNH 1958. 5.2.1, dorsal view); (B) anterior end of *M. johnstoni* from Berwick-upon-Tweed (NMW.Z. 2013.037.0001, dorsal view, photo by Andrew S.Y. Mackie) showing straight thoracic interparapodial margins; (C) anterior end of *M. variolamellata* from Paranaguá Bay, Brazil (dorsal view) showing oblique lateral slits; (D) anterior end of *M. crenulifrons* from Hong Kong (ventral view, from *Mortimer & Mackie, 2009*) (continued on next page…)

**Figure 11 (…continued)**
showing V-shaped ventral marking in the mid-thoracic region and methyl green staining pattern; (E) anterior end of *M. pulchella* from Kuwait (holotype, BMNH 1969.391, ventral view) showing X-shaped ventral marking in the mid-thoracic region, stained with methyl green; (F) thoracic ventral pads/swellings of *M. mirabilis* from Berwick-upon-Tweed in the mid to posterior thorax (NMW.Z.2013.037.0020, ventral view, photo by Andrew S.Y. Mackie); (G) posterior thorax of *M. obockensis* from Obock, Red Sea (syntype, MNHN Type 1357, ventral view) showing ventral swellings/pads. Ach, achaetous first segment; BO, burrowing organ; DS, dorsal slits; Pp, palp; Pr, prostomium, Th, thorax; V, ventral V-shaped marking; VS, ventral swellings; X, ventral X-shaped marking.

"sulcos transversais" (transverse slits) for members of *M. variolamellata* (Fig. 11C). *Brasil (2003)* used the term dorsal furrows/grooves, highlighting them to be present also among members of *M. polydentata* [character 53(1)], whilst herein, the term dorso-lateral grooves is utilised. *Brasil (2003)* stated that they vary in number, position, and location, and they were noted to be absent in the majority of members of magelonid species [character 53(0)] and are additionally absent among members of the outgroups.

### Dimensions of post-chaetiger 9 segments (subject 54)

*Uebelacker & Jones (1984)* described posterior segments that are much longer than wide [character 54(1), Fig. 12A] among members of an undescribed species, *Magelona* sp. H, from the Gulf of Mexico. Later, *M. posterelongata* was the first species with individuals formally described with this feature. Many authors have ignored this characteristic, a problem compounded by the difficulty in collecting entire specimens. A further problem in coding this character is that abdominal chaetigers may vary in proportion along the length of the posterior region. It is believed, however, that the majority of magelonid species possess abdominal chaetigers equal in length and width, or only marginally longer than their width; a characteristic shared with members of the outgroups [character 54(0)].

### Characters of the parapodia of post-chaetiger nine segments (subjects 55–59)

The abdominal region of magelonids consists of many chaetigers carrying biramous parapodia. In contrast to the thorax, the abdominal lamellae of the notopodia and neuropodia are generally symmetrical and of similar size and shape in both rami. The abdominal region starts at chaetiger 9 for members of *Octomagelona* (Fig. 6) and chaetiger 10 for members of *Magelona* (Fig. 11A) and is marked by a change in chaetal type, from capillary chaetae to hooded hooks. Abdominal lamellar shape **(subject 55)** varies among members of species, from triangular to rounded. They may be basally constricted [those with rounded lamellae, character 55(1), Figs. 12C, 12H], or without a basal constriction, being broad based [character 55(0), Fig. 12B]. The latter situation being generally observed among members of species possessing more triangular lamellae, tapering to pointed tips (Fig. 12D). Subject 55 is not applicable to members of *Phyllochaetopterus limicolus*, however all other members of the outgroups have lamellae without a basal constriction [character 55(0)].

At the inner margins of chaetal rows, triangular to digitiform processes **(subject 56)** may be present [character 56(1), Figs. 12C, 12F, 12H, 12I]. Jones termed these structures dorsal medial lobes (DML) for notopodia and ventral medial lobes (VML) for neuropodia.

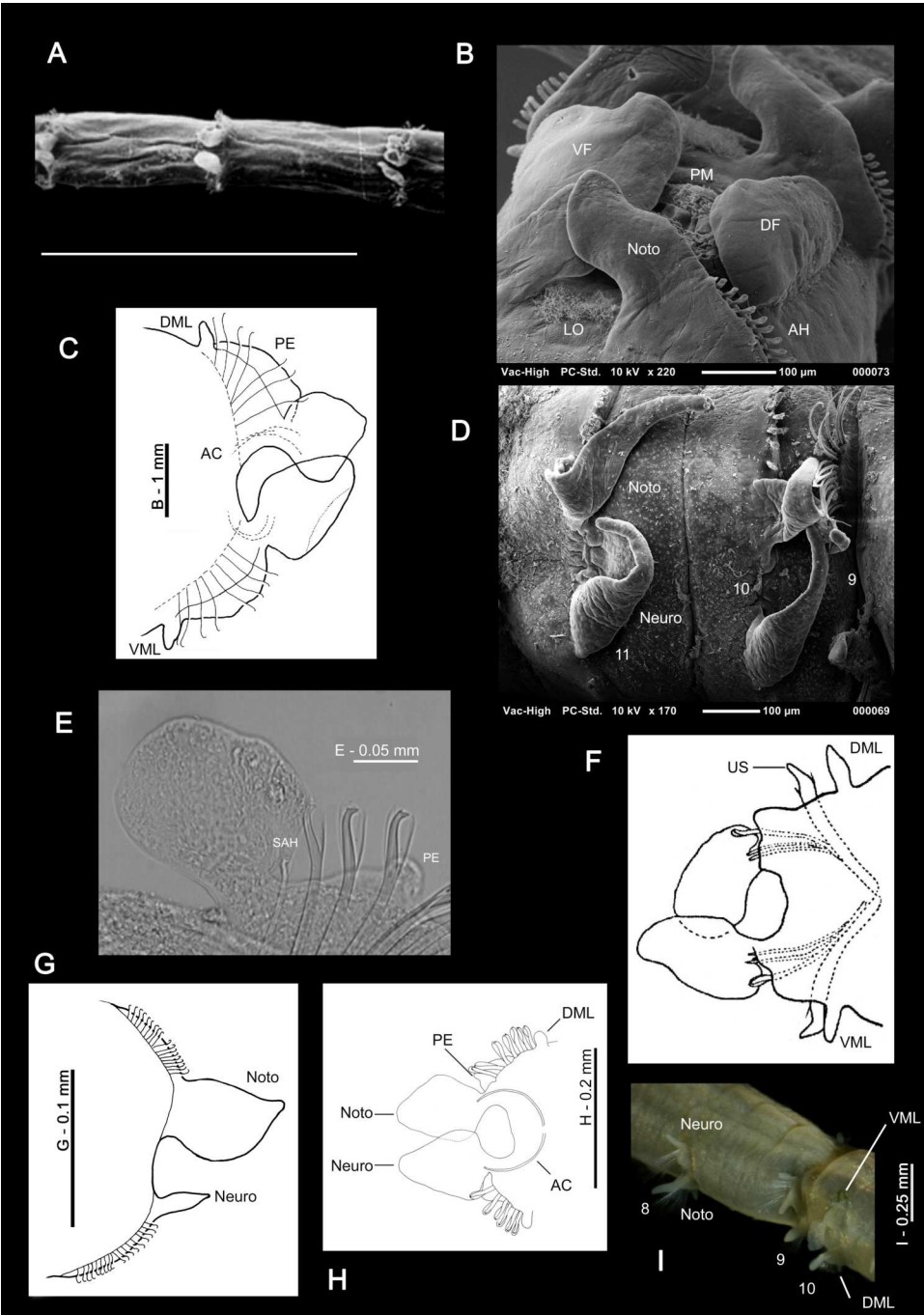

**Figure 12  Magelonid abdominal chaetigers.** (A) Wider than long abdominal chaetigers of *Magelona posterelongata* from Paranaguá Bay, Brazil (lateral view, scale one mm); (B) right-hand parapodia of chaetigers 10 and 11 of *M. johnstoni* from Berwick-upon-Tweed (NMW.Z.2013.037.0011b, anterior view), showing anteriorly open pouch; (C) left-hand parapodia of chaetiger 10 of *M. crenulifrons* from Iran (NMW.Z.2010.037.0034b, anterior view); (D) right-hand (continued on next page…)

*Uebelacker & Jones (1984)* believed that they may occur universally throughout the family, but this is a viewpoint not shared by the current authors. They are absent [character 56(0)] in members of the outgroups, however, this subject is considered not applicable to *Phyllochaetopterus limicolus*. In members of magelonid species, the size of the medial lobes in the abdomen varies widely, from minute and difficult to discern, to long and conspicuous structures, *e.g.*, among members of *Magelona montera*. For comparison, the medial lobes were coded depending on their height relative to abdominal hooded hooks (**subject 57**): character 57(0) for members of species in which they are smaller than the hooks and character 57(1) when they are larger than the hooks. The latter subject is not applicable to the outgroups.

The abdominal lamellae may extend behind chaetal rows (**subject 58**). This postchaetal expansion (previously termed the interlamella) is often more conspicuous in the anterior abdomen and when present [character 58(1), Figs. 12C, 12E and 12H] it may be rounded to triangular. Among members of species in which it is absent [character 58(0), Figs. 12B, 13D and 14B], the abdominal hooded hooks arise from a distinct ridge (Figs. 12B and 13E). Postchaetal expansions are absent among members of the outgroups, whilst in members of *Phyllochaetopterus limicolus* the subject is considered inapplicable.

As stated above, abdominal lamellae (**subject 59**) in magelonids are generally symmetrical in size and shape [character 59(0), Figs. 12D and 12H] as in members of the outgroups. However, in some magelonids, such as members of *Magelona alleni*, the lamellae are sub-equal between the two rami [character 59(1), Fig. 12G]. This subject is inapplicable to members of *Phyllochaetopterus limicolus*.

### Characters of lateral abdominal pouches (subjects 60–65)

Lateral pouches (**subject 60**) are present among members of some species of magelonids, located laterally between the parapodia of certain abdominal chaetigers [character 60(1)]. Their presence [character 60(1)] or absence [character 60(0)] is often a key diagnostic feature, although their function is currently unknown (*Jones, 1968*; *Mortimer & Mackie, 2014*). Pouches vary in morphology, direction of opening, number, and location. However, descriptions of these structures have generally been limited, often only indicated as being present or absent. *Fiege, Licher & Mackie (2000)* described two types of lateral pouch: Σ-shaped occurring in the anterior abdomen, generally paired, and opening anteriorly, whilst C-shaped pouches were noted to occur on median and posterior abdominal chaetigers,

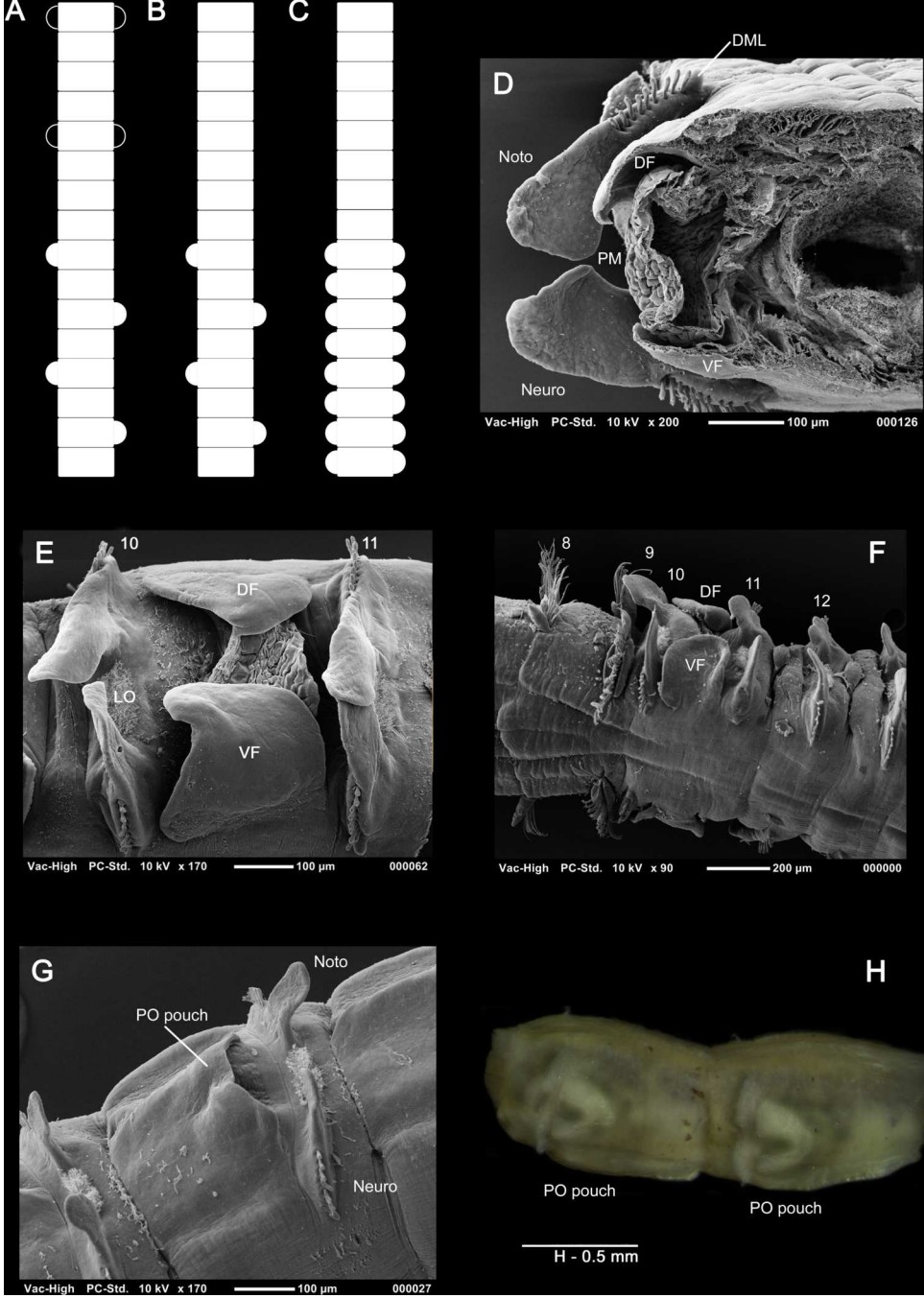

**Figure 13 Magelonid lateral abdominal pouches.** (A) Schematic diagram showing a 'magelonid' with paired anteriorly open pouches in the anterior abdomen, followed by unpaired posteriorly open pouches in the mid and posterior abdomen, on alternate chaetigers; (B) schematic diagram of a magelonid with unpaired posteriorly open pouches, on alternate chaetigers; (C) schematic diagram of a magelonid with paired posteriorly open pouches, on consecutive chaetigers; (D) transverse section through the body and an anteriorly opening pouch situated between chaetigers (continued on next page...)

**Figure 13 (…continued)**
10 and 11 of *Magelona johnstoni* from Berwick-upon-Tweed (posterior half of pouch and parapodia of chaetiger 11 visible) (NMW.Z.2013.037.0010c, anterior view); (E) left-hand anteriorly opening pouch between chaetigers 10 and 11 of *M. johnstoni* from Berwick-upon-Tweed (NMW.Z.2013.037.0011b, lateral view); (F) the same from another specimen collected from the same locality, ventrolateral view (NMW.Z.2013.037.0008c); (G) posteriorly open pouch of the same specimen (ventrolateral view); (H) posteriorly open pouches on consecutive segments of *M. pacifica* from Panamá (syntype, BMNH Type 1933.7.10.65/70, lateral view), pouches showing distinct medial splits. DF, dorsal flap; DML, dorsal medial lobe; LO, lateral organ; Neuro, neuropodia; Noto, notopodia; PM, pouch membrane; PO pouch, posteriorly opening pouch; VF, ventral flap. Numbers indicate chaetiger.

opening posteriorly and occurring in pairs or singly. *Mortimer (2010)* suggested these terms should be abandoned to enable variations to be better understood, and subsequently, *Mortimer & Mackie (2014)* and *Mortimer (2019)* detailed further pouch morphologies. Following the latter authors, lateral pouches when present, were coded based on six subjects: arrangement of pouches (**subject 61**), direction of pouch openings (**subject 62**), posteriorly open lateral pouch arrangement (**subject 63**), lateral pouch distribution (**subject 64**), and margins of posteriorly open pouches (**subject 65**). Firstly, pouches are either all paired [character 61(0), Fig. 13C] or unpaired [character 61(1), Fig. 13B], or both types are present [character 61(2), Fig. 13A]. Secondly, both anteriorly and posteriorly opening pouches are present [character 62(1), Figs. 13A, 13E and 13G], *e.g.*, members of *Magelona johnstoni*, or only posteriorly open pouches are present [character 62(0), Figs. 13B and 13C]. Posteriorly open pouches may occur on alternate segments [character 63(1), Fig. 13B], or on consecutive segments [character 63(0), Fig. 13C], or alternatively, they may be present on both consecutive and alternating chaetigers at varying points along the abdomen [character 63(2)], *e.g.*, members of *M. pulchella*. *Fiege, Licher & Mackie (2000)* described the first occurrence of lateral pouches on the body as an important distinguishing character between members of species and is relatively easy to observe. In members of some species, lateral pouches are present throughout most of the abdomen [character 64(0)], whilst in others they are present only on median and posterior chaetigers [character 64(1)], and in others restricted to the posterior abdomen only [character 64(2)] (*Mortimer & Mackie, 2014*). Whilst the margins of most posteriorly open pouches are smooth [character 65(0)], in members of some species, such as *M. pacifica*, they are medially split [character 65(1), Fig. 13H]. One of the biggest problems in coding these characters is the lack of information known about the posterior regions of many magelonids, whilst some pouches are present throughout most of the abdomen and are therefore likely to be recorded even in posteriorly incomplete specimens. For members of species in which pouches only occur in the extreme posterior region, pouches are likely to be recorded as absent if individuals are described from anterior fragments only. Lateral pouches are present [character 60(1)] in members of outgroups, *Laonice cirrata* and *Prionospio ehlersi*. These are paired [character 61(0)], posteriorly open [character 62(0)] and with smooth margins [character 65(0)]. They occur on consecutive segments [character 63(0)] on median and posterior chaetigers [character 64(1)]. Lateral pouches are absent [character 60(0)] in the remaining members of the outgroups.

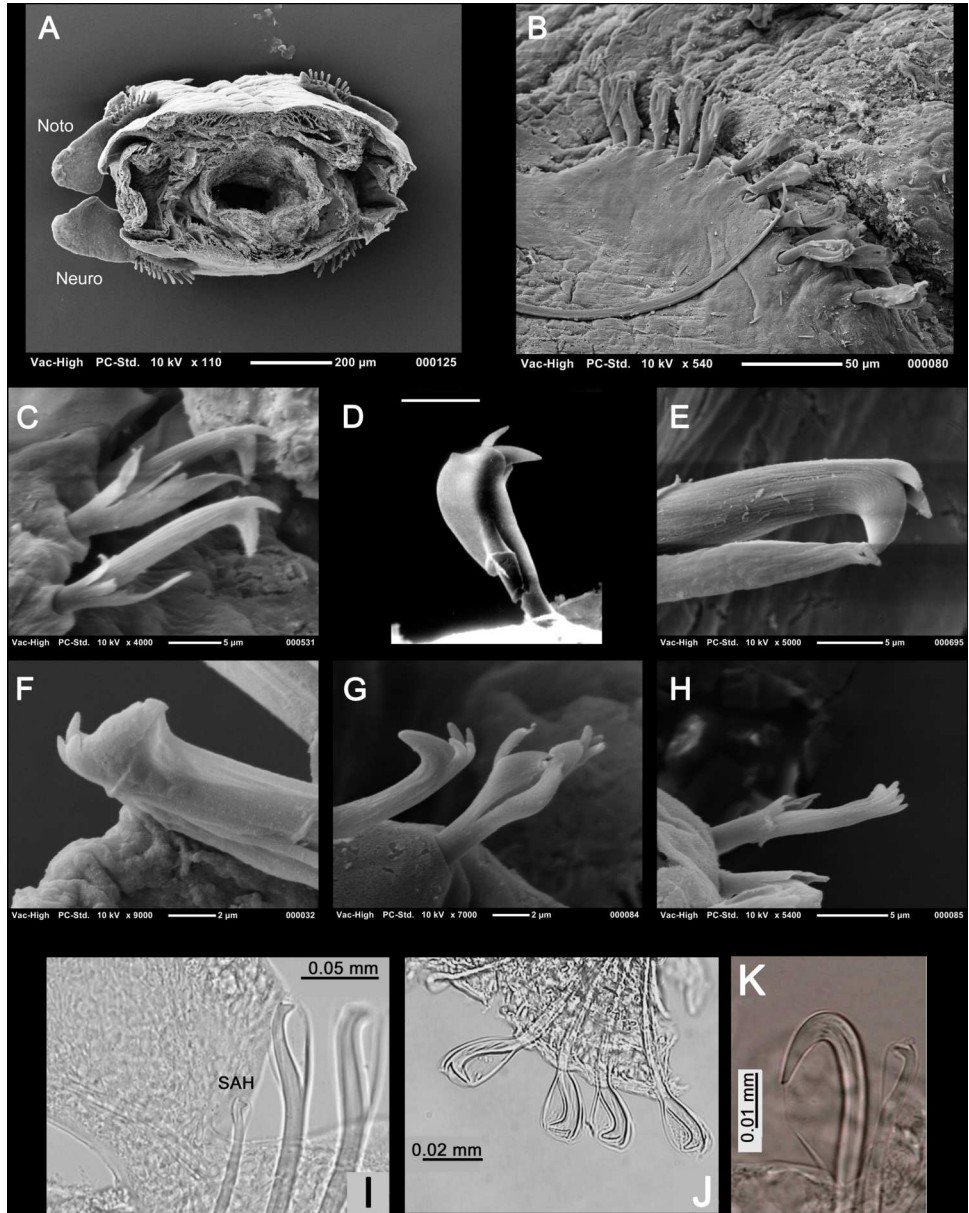

**Figure 14** **Magelonid abdominal hooded hooks.** (A) Transverse section of the body of *Magelona johnstoni* from Berwick-upon-Tweed, between chaetigers 10 and 11 (posterior half of pouch and parapodia of chaetiger 11 visible) (NMW. Z.2013.037.0010c, anterior view); (B) row of hooded hooks arranged vis-à-vis of *M. equilamellae* from Ebro Delta, Catalonia; (C) bidentate hooded hooks from chaetiger 29 of *M. minuta* from the Irish Sea (NMW.Z.1991.075.1584a, lateral view); (D) bidentate hooded hook from *M. papillicornis* from Brazilian coast (lateral view, scale = 0.01 mm); (E) tridentate hooded hook from *M. alleni*; (F–H) quadridentate, pentadentate and hexodont hooded hooks from *M. fauchaldi* from Iran (paratypes, NMW.Z.2015.012.0002d–e); (I) small hooded hook adjacent to the lateral lamellae of *M.* sp. 3 from the Brazilian coast (from *Brasil, 2003*, anterior view); (J) bidentate hooded hooks of *M. papillicornis* from Brazilian coast, arranged vis-à-vis; (K) unidentate enlarged hook of *M.* sp. 3 from the Brazilian coast (from *Brasil, 2003*). C, E–H, hoods broken. Photos B, C, E by Kimberley Mills. Neuro, neuropodia; Noto, notopodia; SAH, small abdominal hook.

### Characters of post-chaetiger 9 chaetae (subjects 66–74)

Magelonids possess hooded hooks in the abdominal region (**subject 66**), located between the lamellae and medial lobes (when present). They are present [character 66(1)] in members of all magelonid taxa and members of the outgroup except *Phyllochaetopterus limicolus* [character 66(0)]. In magelonids, hooded hooks generally occur in one row per ramus, but recent studies suggest that they can occur in two rows, at least in the medial part of the rami (*Mortimer et al., 2020*) (Fig. 14B for members of *M. equilamellae*). Although, this may prove to be an important characteristic in future studies, due to lack of information for the majority of species this was not considered in this study.

The dentition of abdominal hooks (**subject 67**) may be bidentate [character 67(0), *e.g.*, members of *Magelona minuta* and *M. papillicornis*, Figs. 14C and 14D] with one secondary tooth above the man fang, tridentate [character 67(1), Fig. 14E], with two secondary teeth above, or polydentate [character 67(2)], having three or more secondary teeth above the main fang. Quadridentate, pentadentate, and hexodont hooks (Figs. 14F–14H) have been recorded among Magelonidae, with some polydentate individuals carrying more than one type of hook. Where members of species have been recorded with more than one, the dentition which predominates was coded. For example, members of *M. minuta* are recorded to possess the odd sporadic tridentate hook (*Mills & Mortimer, 2018*), but carry predominately bidentate hooks. Members of the outgroup *Spio filicornis* possess bidentate [character 67(0)] hooded hooks, whilst members of *Prionospio lighti*, *P. ehlersi* and *Laonice cirrata* possess polydentate hooded hooks [character 67(2)]. This subject is not applicable to members of *Phyllochaetopterus limicolus*, which have uncini.

*Jones (1971)* was the first author to draw attention to the fact that in members of some species the hook at the base of the lamellae (**subject 68**) may be smaller than the others [character 68(1), Figs. 12E and 14I]. Later it has been suggested that not only their size but also their appearance may indicate groups of closely related species (*Magelona pitelkai, M. hobsonae, M. hartmanae, M. dakini, M. filiformis, M. capensis*). However, *Brasil (2003)*, upon re-examination of the holotypes of the first four species, observed no differences in shape, only differences in size. Among members of other magelonid species, the hook at the base of the lamellae is of equivalent size to the rest [character 68(0), Fig. 13D]. The subject is not applicable to members of the outgroups.

*Uebelacker & Jones (1984)* first reported the presence of enlarged chaetae (**subject 69**) in the abdominal region of members of four undescribed species from the Gulf of Mexico; *Magelona* sp. C, possessing an enlarged recurved hook with a minute apical tooth in each ramus; *Magelona* sp. D, with enlarged hooded apical spines (Fig. 12F); *Magelona* sp. E, possessing large bidentate hooks akin to "ordinary" hooks; and lastly *Magelona* sp. H, with large unidentate recurved spines. *Hernández-Alcántara & Solís-Weiss (2000)* formally described members of two of these species (*Magelona* spp. D and H) as *Meredithia spinifera* and *M. uebelackerae*, respectively, erecting the new genus based on this distinctive feature. The authors stated that "although this character is the only one that differentiates this group from the other species described in this monogeneric group, it is sufficiently distinctive to make it a valid generic level character." However, *Mortimer & Mackie (2003)* felt that stronger evidence for recognising other genera within the family was needed, preferring
to follow *Uebelacker & Jones (1984)* by including species with enlarged abdominal chaetae in *Magelona*, and synonymising *Meredithia* with *Magelona*. Two additional species with enlarged abdominal chaetae have since been described: *Magelona magnahamata* and *Magelona falcifera*. For the current analysis, the presence [character 69(1)] and absence [character 69(0)] of enlarged chaetae in the abdomen was noted as well as the morphology of the distal ends (**subject 70**), *i.e.*, recurved [character 70(0), Fig. 14K], spine-like [character 70(1)], or an enlarged normal hook [character 70(2)]. Enlarged hooks are not present in members of the outgroups [character 69(0)], therefore subject 70 is not applicable (together with subject 69 for *Phyllochaetopterus limicolus*).

The number of hooded hooks per ramus in abdominal chaetigers (**subject 71**) is variable and characters distinguished as eight or more hooks per ramus [character 71(0), *e.g.*, members of *Magelona equilamellae*, Fig. 14B, and members of outgroups *Prionospio lighti, P. ehlersi* and *Laonice cirrata*], and less than eight [character 71(1), Fig. 14J], such as among members of *M. papillicornis* and members of outgroup *Spio filicornis*. This separation in number was based on the examinations of specimens from a number of different species and assessing several chaetigers of the same specimen. As variations in the number of hooks have been observed along the length of the abdomen (*e.g.*, members of *M. papillicornis* have a higher number of hooks in the mid-abdominal region), the number of hooks was counted in the anterior abdomen. Subject 71 is not applicable for members of *Phyllochaetopterus limicolus*. Hooks may be arranged in two formations (**subject 72**): occurring in two groups per ramus with main fangs arranged face-to face (vis-à-vis) [character 72(0), Figs. 12C, 12G, 12H, 14B and 14J], or in unidirectional rows (vis-à-dos) with main fangs pointing in one direction [character 72(1), Fig. 13D]. Vis-à-vis orientation is more common among members of the Magelonidae, whilst vis-à-dos is present for all members of the outgroups (subject 72 is not applicable to *P. limicolus*).

Neuropodia with sabre chaetae (**subject 73**) are present [character 73(1)] among members of the spionid outgroups. Such chaetae are absent [character 73(0)] among members of *Phyllochaetopterus limicolus* and all ingroup species.

Internal support chaetae, or aciculae (**subject 74**), have been reported among members of several magelonid species. These curved and slender chaetae support the bases of the lamellae, and it is possible that they are more prevalent among members of species with larger abdominal lamellae. The proximal ends of the aciculae may overlap in the junction between the noto- and neuropodia (Figs. 12C and 12H). Notes on their presence [character 74(1)] or absence [character 74(0)], as well as their description, are poorly reported in the literature. *Jones (1978)* stated that all magelonid abdominal chaetigers are supported by these structures, although this is not an observation supported by the current authors. Aciculae are generally more conspicuous in larger specimens, whilst slide preparations of parapodia may be necessary for smaller individuals. The relative obscurity of aciculae may be a factor in the limited inclusion in descriptions of specimens. Aciculae are absent in all members of the outgroups [character 74(0)].

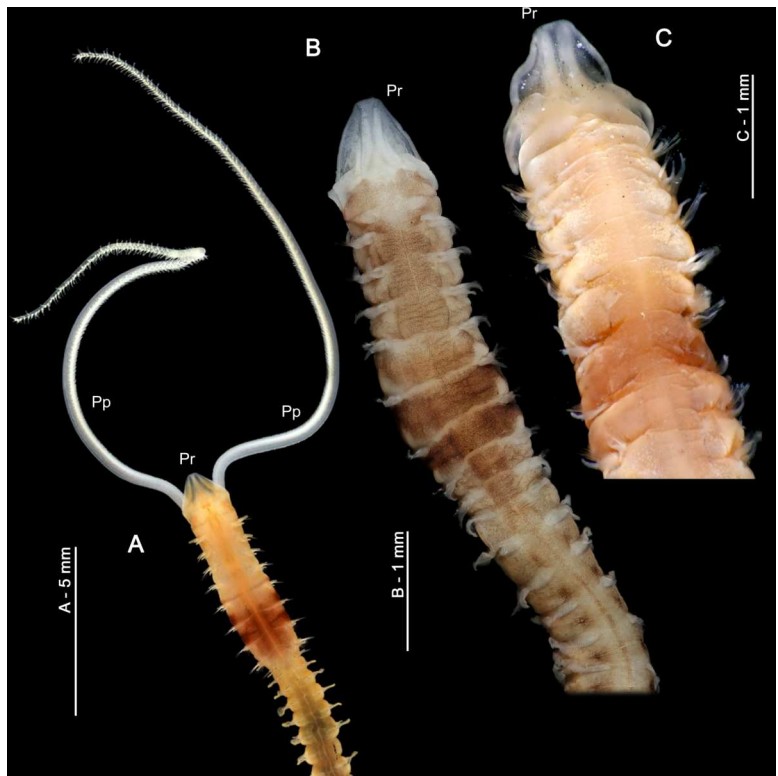

**Figure 15  Magelonid pigmentation.** (A) Anterior end of a live *Magelona alleni* (dorsal view) from Plymouth, showing thoracic pigment band on posterior thorax (photo by Andrew S.Y. Mackie); (B) anterior end (dorsal view) of *M. fasciata* from Senegal, West Africa (NMW.Z.2021.001.0009), showing distinct stripy pigmentation characteristic of the species; (C) pigment band of posterior thorax of *M. equilamellae* from Ebro Delta, Catalonia (MNCN 16.01/18576, photo by Kimberley Mills). Pp, palp; Pr, prostomium.

### Characters of body pigmentation (subjects 75–76)

Among members of some species pigmentation in the posterior thorax (**subject 75**) is evident even in fixed material [character 75(1)], often deep reddish to brown in colour. Very little is known about this character beyond presence or absence [character 75(0)]. When pigment is present, there are two patterns of distribution (**subject 76**). Among members of some species, such as *Magelona alleni* and *M. equilamellae*, the pigment forms a distinct band [character 76(0), Figs. 15A and 15C], often from chaetigers 4/5 to 8; among members of other species only light, dispersed pigmentation in posterior chaetigers is noted [character 76(1)], *e.g.*, members of *M. symmetrica*. Moreover, members of *Magelona fasciata Mortimer, Kongsrud & Willassen (2021)* from West Africa are noted to have distinct stripy pigmentation over much of the body (Fig. 15B). Pigmentation on the palps of members of some species, such as *M. mirabilis*, has been noted (see *Mortimer 2019*: fig. 4.2.6). However, insufficient information is currently available to enable coding of this character. Relevant pigmentation is absent in members of the outgroups [character 75(0)].

### Characters of granular bodies (subjects 77–78)

The surfaces of segments can have granular bodies (**subject 77**) appearing as spots, distributed in a variety of patterns on both the thorax and abdomen, and remain upon fixation and preservation of material [character 77(1), Figs. 6A, 9H and 15C]. *Jones (1971)* felt that this characteristic had no systematic significance. However, patterns appear to be species specific, particularly those of the thoracic region. The latter may appear as transverse stripes (Fig. 16A) or distinct smaller circular regions adjacent to the parapodia (Fig. 16B). Abdominal interparapodial areas commonly have granular bodies occurring in patches (**subject 78**) [character 78(1), Figs. 16C and 16D] among members of Magelonidae and may cover a large part of the lateral region between parapodia of adjacent chaetigers. Both thoracic and abdominal bodies may stain with dyes such as Rose Bengal, or be highlighted by methyl green, in which the granules appear white against the contrasting stain (Figs. 16B and 16C). Whilst staining patterns can be extremely useful in separating members of species (Figs. 11D, 11E and 16A), it was not included in the current study. However, it may prove useful in the future, as more methyl green patterns are described. Granular bodies and abdominal interparapodial areas are absent in members of the outgroups [character 77(0), character 78(0)].

### Tube construction (subject 79)

In general, magelonids are relatively motile, burrowing more or less continually through sediments (*Jumars, Kelly & Lindsay, 2015*), without constructing tubes [character 79(0)]. However, members of several species are known to build distinct multi-layered tubes covered in sand [character 79(1), Fig. 16E]. *Mills & Mortimer (2019)* investigated the permanency of these tube-lined burrows among members of the European species, *Magelona alleni*. Members of some species, such as *M. filiformis*, have been described living in fragile tubes of a secretion to which sand grains adhere. These are not considered as tubes *per se* but are believed to be the worm's response to removal from its habitat. Members of the outgroups, *Spio filicornis* and *Phyllochaetopterus limicolus*, are known to build distinct tubes [character 79(1)], whilst the situation is unknown for members of *Prionospio lighti*, *P. ehlersi* and *Laonice cirrata*.

## Characters not included, but of possible future consideration
### Prostomial features

Magelonid prostomia may carry distinct markings on either side of the dorsal muscular ridges (Figs. 4A, 4C–4E, 5A, 5B and 5E), the pattern of which is generally species specific, whilst members of other species show no obvious markings (Figs. 4G and 4H). Although this can be an important diagnostic feature, this was not included in the current analysis due to difficulties in adequately describing this character.

### Thoracic chaetae

*Mills & Mortimer (2018)* highlighted differences in the number of thoracic chaetae among members of different species, and additionally observed variations in number of chaetae on different chaetigers of the same animal for members of several magelonid species. They noted that members of some species, such as *Magelona cincta*, have distinctly splayed

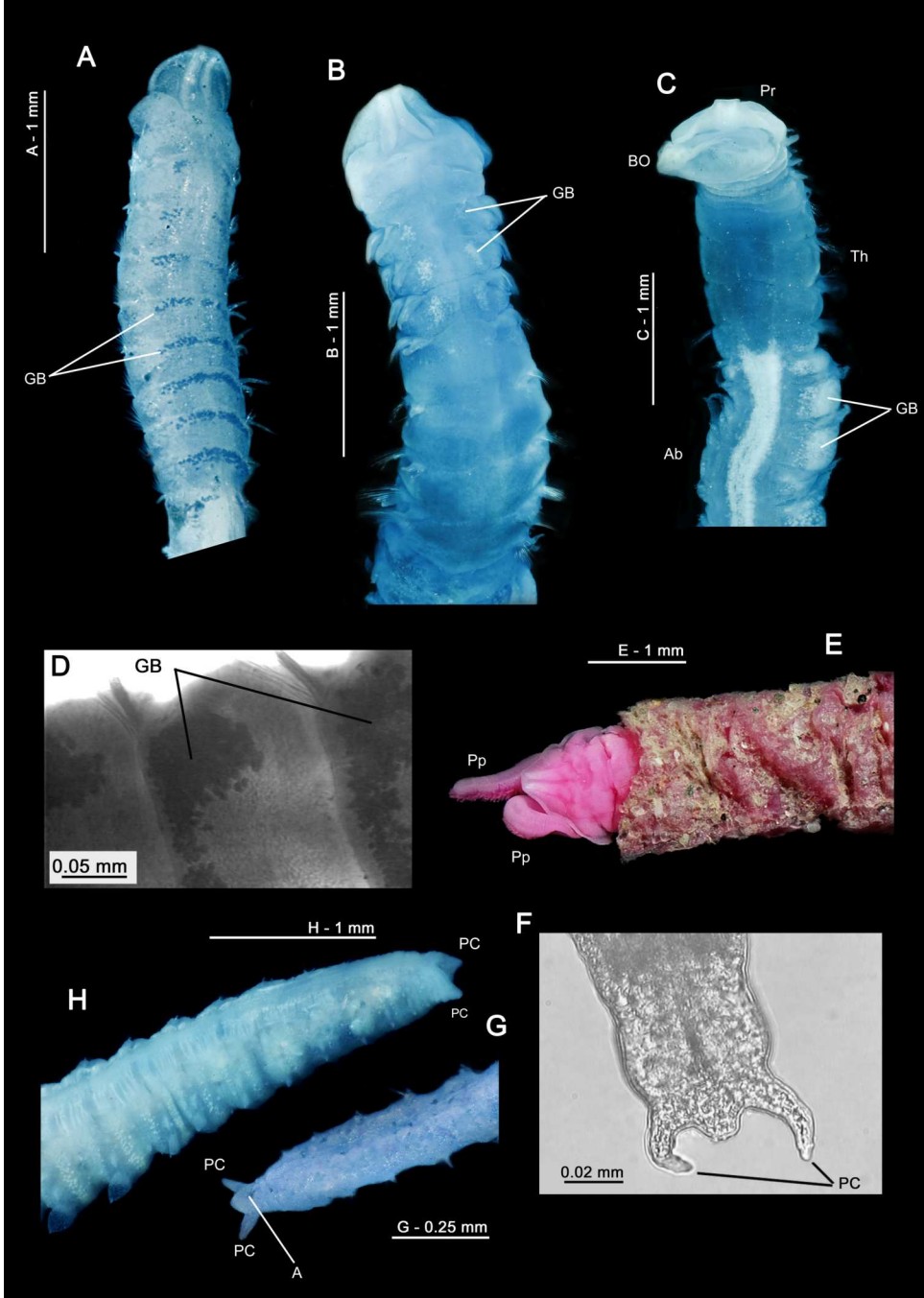

**Figure 16  Magelonid pigmentation, granular bodies, tube, pygidial cirri.** (A) Anterior region of *Magelona minuta* from the Celtic Sea (NMW.Z.2005.014.0111, dorsal view; photo by Kimberley Mills) showing distinct transverse thoracic stripes; (B) anterior region of *Magelona* sp. cf. *M. cincta* from Iran (NMW.Z.2015.012.0004, dorsal view) showing distinct circular regions of granular bodies adjacent to the parapodia; (C) same specimen showing abdominal interparapodial patches and granular bodies along the mid-ventral line of the abdomen (ventral view); (D) granular bodies of the dorsal surface of the thoracic region of *M. papillicornis* from Brazilian coast; (E) distinct (continued on next page…)

**Figure 16 (…continued)**
multi-layered tube covered in sand of *M. alleni* from the Irish Sea (NMW.Z.1969.104.1094; photo by Kimberley Mills); (F) digitiform pygidial cirri either side of the pygidium (dorsal view) from Brasil (2003); (G) posterior region of *M. minuta* from the Outer Bristol Channel (NMW.Z.2003.047.5939, ventral view; photo by Kimberley Mills); (H) posterior region of *M. alleni* from Plymouth, England (paratype, BMNH 1958.5.2.1, ventro-lateral view). A–C, G, H, stained with methyl green; E, stained with Rose Bengal. A, anus; Ab, abdomen; BO, burrowing organ; GB, granular bodies; PC, pygidial cirri; Pp, palp; Pr, prostomium; Th, thorax.

chaetae in the posterior thorax when compared to members of other species. Whilst these three characteristics may prove to be useful, limited information is currently available, prohibiting their use in the current study.

### Features of the thoracic region

More recently, several additional characteristics of the thoracic region have been noted. *Mortimer & Mackie (2009)* and *Mortimer (2019)* described distinct V- and X-shaped ventral markings in the mid-thoracic region of members of species such as *Magelona crenulifrons* and *M. pulchella* (Figs. 11D and 11E). Among members of other magelonid species, thoracic ventral pads/swellings have been recorded, such as those observed among members of *M. mirabilis* (*Mortimer & Mackie, 2014*) or *M. obockensis* (*Mortimer, 2010*). These swellings may be oval to reniform in shape (Figs. 11F and 11G). However, occurrence of these two features in members of the majority of species is unknown.

Although, the constriction at the thorax/abdomen junction can be marked and may vary between species, the extent to which it is constricted can be greatly influenced by preservation and fixation and was not considered herein. Despite this, however, it is a character worthy of further study, and should be detailed more fully in future descriptions. *Brasil (2003)* indicated that members of *Octomagelona* present a constriction despite lacking a 9th "thoracic" chaetiger, suggesting that the character may not simply indicate a transition between the thorax and the abdomen.

Whilst average length and the average number of chaetigers appear to be species specific, the lack of information for members of many species prohibits this as a character in the analysis at the present time. Many individuals of magelonid species are described from anterior fragments only. The breadth of individuals, however, may prove an important character. Members of species like *Magelona minuta,* true to their name, rarely attain widths greater than 0.5 mm, being somewhat long and slender. Members of species such as *M. alleni* are distinctly broad and stout, often measuring over one mm in width (*Mortimer, Kongsrud & Willassen, 2021*). Additionally, *Mills & Mortimer (2019)* indicate that stouter magelonid species generally are shorter in comparison to width and have a fewer number of chaetigers. This is something which warrants further investigation.

### Abdominal hooded hooks

*Mills & Mortimer (2018)* highlighted differences in the angle between the main fang and secondary teeth in abdominal hooded hooks of members of several magelonid species and further suggested characteristics such as the width and roundedness of the main fang, and

the angle between it and the axis of the hook shaft, as noted for example by *Jones (1963)* and *Jones (1977)*, may prove useful in future studies.

### *Pygidium*

Magelonids possess a pair of digitiform pygidial cirri on either side of the pygidium (Figs. 16F and 16G). *Uebelacker & Jones (1984)* noted the presence of three pygidial cirri on individuals of an undescribed species, *Magelona* sp. B, from the Gulf of Mexico, however, it is believed that the third cirrus may actually represent the elongated, rounded tip of the pygidium itself. As this is the only magelonid that has been described with this feature, the number of pygidial cirri was not included in this study. *Mills & Mortimer (2019)* noted that pygidial cirri of members of *M. alleni* were more truncate and triangular (Fig. 16H) in comparison to other magelonids, however, this is something that warrants further investigation.

Whilst *Rouse (2001)* stated that the magelonid anus is terminal, early illustrations from *McIntosh (1878)* of members of *Magelona mirabilis* (possibly *M. johnstoni*; see *Fiege, Licher & Mackie, 2000*) clearly show the anus in a distinctly ventral position. Unfortunately, for members of many species the pygidium is unknown, or the amount of information provided by authors has been relatively limited. When included in descriptions, the pygidium is often drawn from a dorsal perspective, thus conclusions about the position of the anus are often difficult to make. *Mills & Mortimer (2019)* investigated the situation among members of five European species, concluding that the anus was ventral in members of four species, whilst in the latter, *M. alleni*, it was distinctly terminal in position (Fig. 16H). Subsequently, *Mortimer, Kongsrud & Willassen (2021)* have noted that members of *Magelona fasciata* from West Africa also have a terminal anus. Whilst this may prove to be a valuable character in subsequent studies, the lack of information among individuals of many species precludes inclusion in the current study.

### Inferring phylogenetic hypotheses

Phylogenetic inferences were performed using PAUP* 4.0b10 (*Swofford, 2001*), with all observations weighted equally and multiple subject–predicate relations ("multistate characters") treated as non-additive (Table S1). The following command string was executed (*Larkin, Neff & Simpson, 2006*; *Fitzhugh, 2010b*; *Fitzhugh et al., 2015*; *Nogueira, Fitzhugh & Rossi, 2010*; *Nogueira, Fitzhugh & Hutchings, 2013*): hsearch enforce = no start = stepwise addseq = random nreps = 100000 nchuck = 5 chuckscore = 1; hsearch enforce = no start = current chuckscore = no. As noted earlier in **Methodological considerations**, the commands used are not tantamount to what has been incorrectly called "parsimony analysis." Rather, implementation of these commands is consistent with abductive reasoning for the purpose of causally accounting for differentially shared characters by way of common causes as fully as possible. Character transformation series were examined using Mesquite 3.61 (*Maddison & Maddison, 2011*).

## RESULTS

The phylogenetic inference using data in Table S1 produced 2,417,600 cladograms of 404 steps each. The consistency index for each is 0.243 and retention index is 0.744. The strict consensus tree for all cladograms is shown in Fig. 17. What is notable is the mix of clades and grades within the monophyletic Magelonidae. A consequence of the overall arrangements of these phylogenetic hypotheses is that there can be no generic-level phylogenetic hypotheses that can be formally named without also incurring paraphyletic taxa. The result, as will be pointed out in the **DISCUSSION**, is that the only (composite) phylogenetic hypothesis that can be formally recognised is Magelonidae. The Magelonidae clade involves seven phylogenetic hypotheses referring to the following respective characters:

1. Shovel-shaped prostomium:  absent (0) → *present* (1)
2. Nuchal organs:  present (1) → *absent* (0)
8. Prostomial ridges:  absent (0) → *present* (1)
10. Palp origin:  dorsal (0) → *ventral* (1)
11. Palp surface:  non-papillate (0) → *papillate* (1)
14. Burrowing organ:  absent (0) → *present* (1)
15. Body regionation:  non-magelonid (0) → *magelonid-like* (1).

The present results indicate that *Octomagelona* cannot be maintained, given that the clade is nested within *Magelona*. And while the arrangements of the phylogenetic hypotheses among all the cladograms impose severe limits that only allow for formally recognising Magelonidae, there are no phylogenetic hypotheses to which the type genus *Magelona* refer. The implications of these results will be addressed in the **DISCUSSION**.

In addition to the Magelonidae, there are 20 additional clades, indicated by letters **a** through **t** in Fig. 17, from which phylogenetic hypotheses deserve mention. None of these clades can be formally named, given the various grade groups also present among the cladograms. Whilst serving as synapomorphies for these clades, those characters that are homoplasious (including reversals and convergence) are denoted by an asterisk (*):

Clade a–  22. Chaetigers 1–7 notopodial lamellae shape: foliaceous (1) → *filiform* (0)*
26. Chaetigers 1–7 neuropodial lamellae position relative to neurochaetae: postchaetal (0) → *subchaetal* (1)*
28. Chaetigers 1–7 neuropodial lamellae shape: foliaceous (1) → *filiform* (0)*

Clade b–  5. Prostomial horns: present (1) → *absent* (0)*
20. Development of neuropodia along thorax: different in some chaetigers, (1) → *all similar* (0)*
26. Chaetigers 1–7 neuropodial lamellae position relative to neurochaetae: subchaetal (1) → *postchaetal* (0)*
30. Chaetigers 1–7 lengths of noto- and neuropodial lamellae: notopodial longer (1) → *equivalent* (0)*

Clade c–  4. Prostomium distal shape: straight (2) → *triangular* (0)*

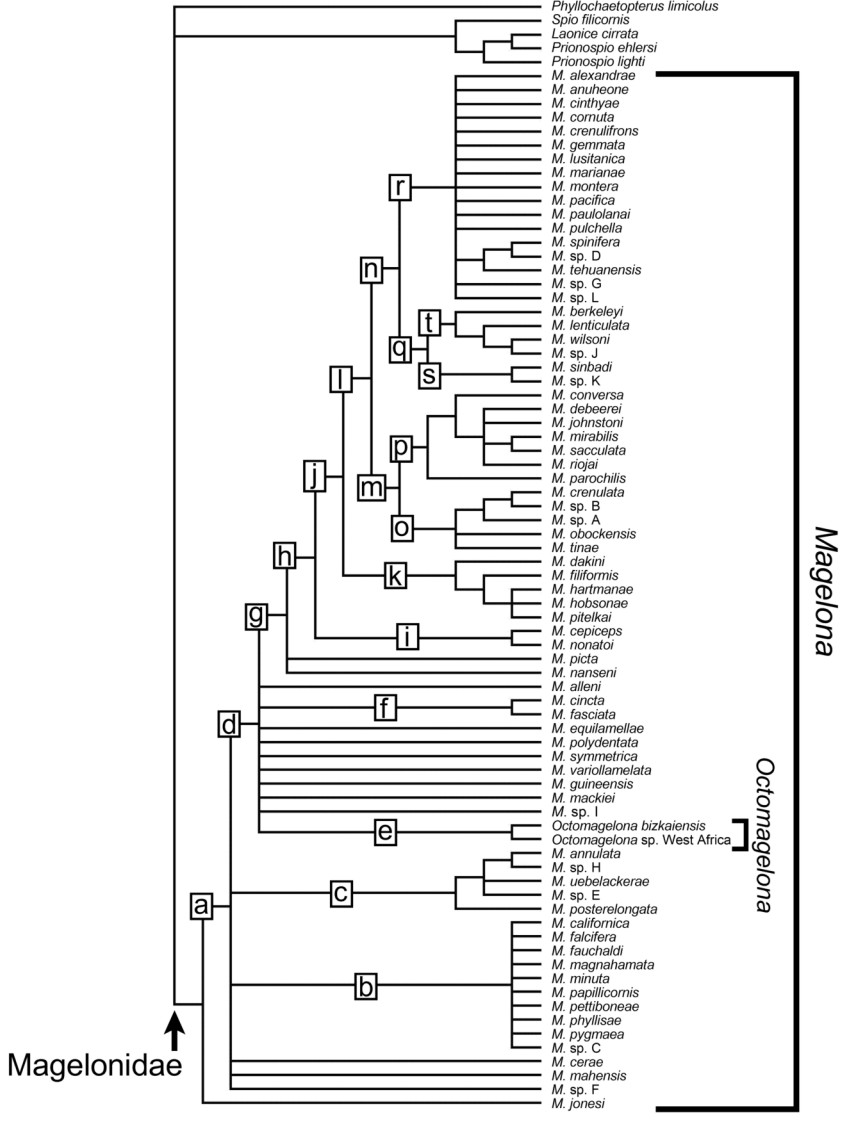

**Figure 17 Phylogenetic results. Strict consensus tree for 2,417,600 cladograms produced from data matrix in Table S1.** Note that *Magelona* is paraphyletic relative to *Octomagelona*. Letters **a–t** indicate clades for which some character transformation series are presented in the Results.

Clade d–

6. Shape of prostomial horns: rudimentary (1) → *distinct* (0)*

46. Chaetiger 9 neuropodial lamellae position: postchaetal (0) → *subchaetal* (1)*

67. Post-chaetiger 9 noto- and neuropodial hooded hooks dentition: bidentate (0) → *tridentate* (1)*

Clade e–

16. Number of anterior body region chaetigers: nine (1) → *eight* (0)

| Clade f– | 18. Development of notopodia along thorax: all similar (0) → *different in some chaetigers* (1)* |
| | 29. Chaetigers 1–7 neuropodial filiform lamellae shape distal ends: pointed (0) → *distally expanded/scoop shaped* (1) |
| | 60. Post-chaetiger 9 lateral pouches: absent (0) → *present* (1)* |
| Clade g– | 22. Chaetigers 1-7 notopodial lamellae shape: filiform (0) → *foliaceous* (1)* |
| | 35. Chaetiger 8 notopodial lamellae shape: filiform (0) → *foliaceous* 35(1)* |
| | 55. Post-chaetiger 9 lamellae shape: without basal constriction (0) → *with basal constriction* (1)* |
| Clade h– | 12. Number of rows of proximal palp papillae: 10–14 (2) → *4–8* (1)* |
| | 43. Chaetiger 9 notopodial lamellae shape: filiform (0) → *foliaceous* (1)* |
| | 75. Pigmentation in posterior thorax: present (1) → *absent* (0)* |
| Clade i– | 18. Development of notopodia along thorax: different in some chaetigers (1) → *all similar* (0)* |
| | 45. Chaetiger 9 dorsal superior lobes: absent (0) → *present* (1)* |
| Clade j– | 38. Chaetiger 8 neuropodial lamellae position: subchaetal (1) → *prechaetal* (2)* |
| | 60. Post-chaetiger 9 lateral pouches: absent (0) → *present* (1)* |
| Clade k– | 22. Chaetigers 1-7 notopodial lamellae shape: foliaceous (1) → *filiform* (0)* |
| | 57. Post-chaetiger 9 dorsal medial lobes length: smaller than hooks (0) → *longer than hooks* (1)* |
| | 68. Post-chaetiger 9 hooded hooks adjacent to notopodial subchaetal or neuropodial suprachaetal lamellae: same size (0) → *smaller than rest* (1)* |
| Clade l– | 46. Chaetiger 9 neuropodial lamellae position: subchaetal (1) → *prechaetal* (2)* |
| Clade m– | 4. Prostomium distal shape: straight (2) → *rounded* (1) |
| | 5. Prostomial horns: present (1) → *absent* (0)* |
| | 42. Chaetiger 9 notopodial lamellae height: elongate (1) → *low* (0)* |
| | 50. Lengths of chaetiger 9 fascicles relative to chaetigers 1–8 fascicles: same length as chaetae in chaetigers 1–8 (0) → *shorter than chaetae in chaetigers 1–8* (1) |

|   |   |
|---|---|
|   | 51. Distal ends of chaetae in chaetiger 9: gently tapered, similar to chaetigers 1–8 (0) → *mucronate* (1) |
|   | 55. Post-chaetiger 9 lamellae shape: with basal constriction (1) → *without basal constriction* (0)* |
|   | 61. Post-chaetiger 9 lateral pouch arrangement: unpaired (1) → *both paired and unpaired* (2)* |
|   | 62. Direction of lateral pouch openings: posteriorly (0) → *posteriorly* and anteriorly (1)* |
| Clade n– | 13. Number of rows of distal palp papillae: four (1) → *two* (0)* |
| Clade o– | 36. Chaetiger 8 margins of foliaceous notopodial lamellae: smooth (0) → *bilobed* (2) |
|   | 44. Chaetiger 9 notopodial foliaceous lamellae margins: smooth (0) → *crenulate* (1)* |
| Clade p– | 72. Post-chaetiger 9 arrangement of hooded hooks: vis-à-vis (0) → *vis-à-dos* (1)* |
| Clade q– | 52. Shape of thoracic interparapodial margins: straight (0) → *rounded, bulbous* (1)* |
|   | 60. Post-chaetiger 9 lateral pouches: present (1) → *absent* (0)* |
| Clade r– | 26. Chaetigers 1–7 neuropodial subchaetal lamellae position relative to neurochaetae: varying in position along thorax (1) → *same position along thorax* (0)* |
|   | 41. Chaetiger 9 notopodial lamellae positions: postchaetal (0) → *subchaetal* (1)* |
| Clade s– | 57. Post-chaetiger 9 dorsal medial lobes length: smaller than hooks (0) → *longer than hooks* (1)* |
| Clade t– | 7. Prostomium dimensions: longer than wide (0) → *wider than long* (2)* |
|   | 33. Chaetigers 1-8 lengths of noto- and neuropodial capillary chaetae: equivalent (0) → *neuropodial longer* (2)* |

## DISCUSSION

A point mentioned in the **RESULTS** is that only one composite phylogenetic hypothesis can be formally recognised, *i.e.,* Magelonidae. There can be no phylogenetic hypothesis(es) to which the name *Magelona* refers that is not redundant with Magelonidae. From a nomenclatural, as opposed to strictly scientific perspective, this assertion is contrary to what is required by the International Code of Zoological Nomenclature (*International Commission on Zoological Nomenclature, 1999*): at least one formal name at the rank of genus must be established relative to the ranks of family and species. The present results offer another good example (see also *Fitzhugh, 2008a*; *Fitzhugh, 2010b*; *Nogueira, Fitzhugh & Rossi, 2010*; *Nogueira et al., 2017*) of the conflict that can occur between the

science of biological systematics and a mandated international nomenclature system that is not entirely aligned with the goal of that science. Compromise between these two positions currently favours the nomenclatural system, but by properly acknowledging that formal taxon names should be defined in terms of being explanatory hypotheses (cf. **Methodological considerations**), the conflict between naming and scientific practice can be somewhat mitigated. Emendations of the formal names Magelonidae, *Magelona*, and *Octomagelona* in relation to the phylogenetic hypotheses inferred in this study are presented below.

## Morphological characters in future specimen descriptions

Various authors have discussed the "crucial morphological characters" in differentiating members of magelonid species. *Jones (1963)*, in his review of magelonids of the Gulf of Mexico listed six: (1) the fine structure of the hooded hooks of the posterior region (subject 67); (2) the presence or absence of prostomial horns (subject 5); (3) the presence or absence of medial lamellae in the posterior region (subject 56); (4) the presence or absence of specialised chaetae of various types on chaetiger 9 (subject 51); (5) the morphology of the anterior lateral lamellae (subjects 18–30, 34–49); and (6) the relative dimensions of the prostomium (subject 7). *Blake (1996c)* listed seven "principal diagnostic characters important in differentiating species," adding to Jones' list the following: (7) presence or absence of dorsal median lobes on thoracic notopodia (subjects 24, 37, 45, superior dorsal lobes); (8) presence and location of lateral pouches between abdominal segments (subjects 60, 64); and (9) the presence/absence and form of interlamellae on abdominal parapodia (herein termed the postchaetal expansion behind chaetal rows, subject 58). *Brasil* (*2003*: fig23) concluded that the position and form of noto- and neuropodial lamellae, and the division of the thorax into three different blocks of segments (*i.e.,* A - all anterior chaetigers with identical morphology; B - first eight chaetigers being identical but possessing a ninth chaetiger having a distinct morphology; and C - in which the first seven identical chaetigers differ from chaetigers eight and nine) reflected the relationships found within clades and sub-clades within the family.

Of the characters previously mentioned, the current results suggest those of particular importance to be included in future descriptions of members of species are: (1) the presence/absence of prostomial horns (in addition to their form, *i.e.,* distinct or rudimentary); (2) the relative dimensions of the prostomium; (3) the morphology of the anterior lamellae, including presence/absence of superior dorsal lobes; (4) the presence/absence of specialised chaetae on chaetiger 9; and (5) the presence of lateral abdominal pouches. As highlighted by *Brasil (2003)*, the thoracic lamellae are extremely diagnostic. Great care must be taken to fully describe and illustrate all thoracic chaetigers in descriptions, taking into account the size, shape and position of thoracic lamellae and whether/how they vary along the thorax. Often variations in position of the lamellae may be subtle between thoracic parapodia, particularly that of filiform subchaetal neuropodial lamellae. It is particularly important to examine the parapodia of chaetigers 8 and 9.

In addition to the characters already highlighted, the prostomial distal shape (subject 3), the number of prostomial ridges (subject 9), the shape of thoracic interparapodial

margins (subject 52), the arrangement of abdominal lateral pouches (subject 63) and the arrangement of abdominal hooded hooks (subject 72) warrant further investigation and inclusion in descriptions.

## Systematics

**Magelonidae Cunningham & Ramage, 1888 (1865), emended**

Type genus. *Magelona* F. Müller, 1858, by monotypy.

**Definition.** A composite phylogenetic hypothesis (Fig. 17), causally accounting for (a) presence of a shovel-shaped prostomium, (b) absence of nuchal organs, (c) presence of prostomial ridges, (d) ventral palps with (e) papillate surfaces, (f) presence of a burrowing organ, and (g) magelonid-like body regionation.

Within a reproductively isolated population of individuals in the past, the following causal events occurred: (a') the presence of a shovel-shaped prostomium [character 1(1)] originated by unspecified mechanism(s) among individuals with a rounded prostomium [character 1(0)], subsequent to which the novel character became fixed in the population by an unspecified mechanism(s); (b') the loss of nuchal organs [character 2(1)] originated by unspecified mechanism(s) among individuals with nuchal organs [character 2(0)], subsequent to which the novel character became fixed in the population by an unspecified mechanism(s); (c') the presence of prostomial ridges [character 8(1)] originated by unspecified mechanism(s) among individuals with no ridges [character 8(0)], subsequent to which the novel character became fixed in the population by an unspecified mechanism(s); (d') the presence of ventrally inserted palps [character 10(1)] originated by unspecified mechanism(s) among individuals with dorsal palps [character 10(0)], subsequent to which the novel character became fixed in the population by an unspecified mechanism(s); (e') the presence of palps with papillae [character 11(1)] originated by unspecified mechanism(s) among individuals with smooth palps [character 11(0)], subsequent to which the novel character became fixed in the population by an unspecified mechanism(s); (f') the presence of a burrowing organ [character 14(1)] originated by unspecified mechanism(s) among individuals without such an organ [character 14(0)], subsequent to which the novel character became fixed in the population by an unspecified mechanism(s); (g') the magelonid-like body [character 15(1)] originated by unspecified mechanism(s) among individuals without such body regionation [character 15(0)], subsequent to which the novel character became fixed in the population by an unspecified mechanism(s). Following character origin/fixation events (a')–(g') was a population splitting event by unspecified mechanism(s) leading to individuals to which subsequent phylogenetic and specific hypotheses refer.

**Remarks.** Results of the phylogenetic inferences by *Brasil* (*2003*: figs. 45 and 46; Fig. 18) acknowledged the monophyly of Magelonidae based on three of the synapomorphies also referred to in this study (cf. **RESULTS**): presence of a spade-shaped prostomium [character 1(1); phylogenetic hypothesis a' in the above **Definition**] and ventral insertion of papillated palps [characters 10(1), 11(1); phylogenetic hypotheses d' and e', respectively, in the above **Definition**]. *Fauchald & Rouse (1997)*: 103) stated evidence for monophyly of the Magelonidae as "Palps with rounded cross-section and a subdistal expanded area

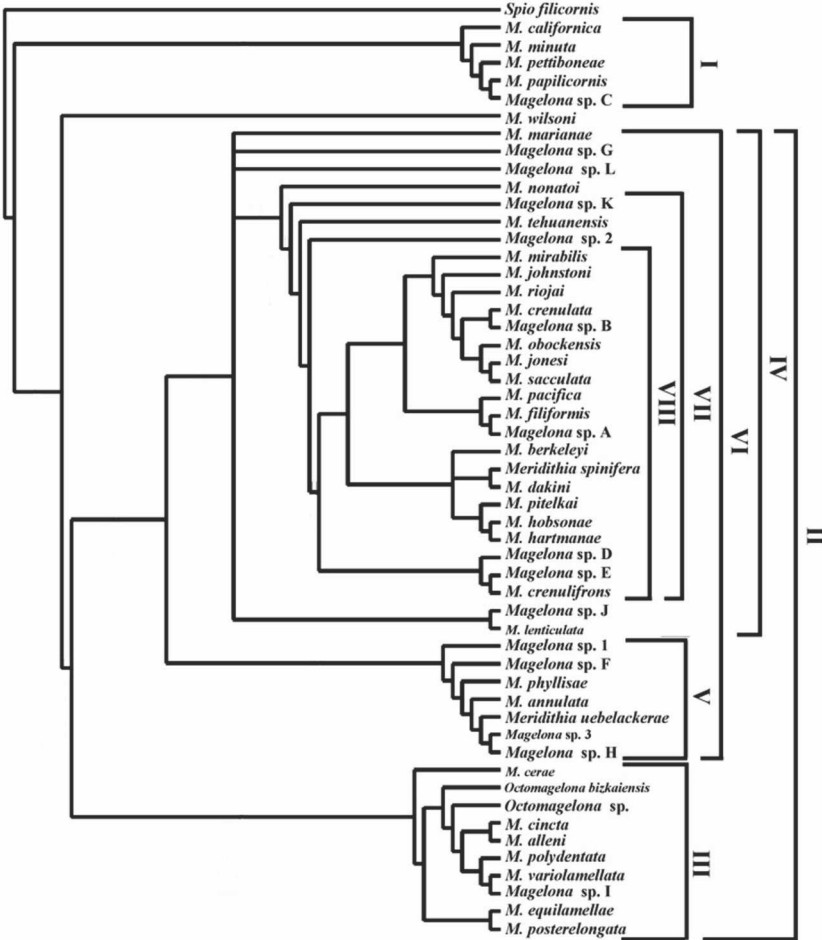

**Figure 18 Phylogenetic results.** Fig. 46 of the strict consensus tree for nine cladograms presented by *Brasil (2003)* of Magelonidae phylogenetic hypotheses. Clades referred to in that study are indicated by the Roman numerals. Reproduced as shown in *Brasil (2003)*.

covered with papillae. Prostomium shovel-shaped." They additionally added that "chaetiger 9 often modified with unusual postchaetal lobes or different chaetae from those segments in front or behind it or both." This does not, however, follow for all members of species within the Magelonidae.

The most basal group in the phylogenetic study by *Brasil (2003)* (Fig. 18: **clade I**) was that containing members of five species: *Magelona californica*, *M. minuta*, *M. pettiboneae*, *M. papillicornis* and *Magelona* sp. C of *Uebelacker & Jones (1984)*. Members of these species possess postchaetal lamellae [characters 21(0), 26(0), 34(0), 38(0), 41(0), 46(0)] of equivalent size [character 30(0)] in both rami of thoracic chaetigers. **Clade b** in the current study (Fig. 17) contains the same five species, along with members of five additional species: *M. falcifera*, *M. fauchaldi*, *M. magnahamata*, *M. phyllisae* and *M. pygmaea*. As well as the characters highlighted by *Brasil (2003)*, individuals in this clade possess smooth [characters 23(0), 36(0), 44(0)], filiform [characters 22(0), 28(0), 35(0), 40(0), 43(0),

48(0)] lamellae without superior dorsal lobes [characters 24(0), 37(0), 45(0)]. They also have prostomia the widths approximately equivalent to their lengths [character 7(1)], with straight [character 4(2)] and smooth [character 3(0)] prostomial anterior margins, but only one pair of prostomial ridges [character 9(0)]. As the names of two species within the clade highlight, *i.e., M. minuta* and *M. pygmaea*, members of these species rarely attain the sizes seen in members of other magelonid species and are slender and thread-like. No lateral abdominal pouches have been recorded among members of these species [character 66(0)], although as noted above (cf. **Character descriptions**) this should be accepted with caution since lateral abdominal pouches are likely underreported within the family. As noted by *Mills & Mortimer (2018)* for members of *M. minuta*, granular bodies [subject 77(1)] appear as distinct transverse stripes in the thoracic region, and spots or stripes of this kind are certainly reported in over half of these species. This is a character worthy of further investigation.

The analysis by *Brasil* (*2003*: fig. 47; Fig. 18) highlighted a **clade VIII.2** containing members of eight species: *Magelona mirabilis, M. johnstoni, M. riojai, M. crenulata, Magelona* sp. B of *Uebelacker & Jones (1984), M. obockensis, M. jonesi* and *M. sacculata.* The synapomorphies of the clade include the presence of mucronate chaetae [character 51(1)] and abdominal hooded hooks in a vis-à-dos orientation [character 72(1)], the latter of which is uncommon amongst members of Magelonidae. **Clade m** (Fig. 17) in the present study includes all the above species (except *M. jonesi* which will be discussed below) and additionally members of *M. conversa, M. debeerei, M. parochilis, Magelona* sp. A of *Uebelacker & Jones (1984)* and *M. tinae.* Members of these species possess longer than wide prostomia [character 7(0)], with rounded [character 4(1)], smooth anterior margins [character 3(0)], carrying two pairs of prostomial ridges [character 9(1)] but lacking prostomial horns [character 5(0)]. The development of lamellae along the thorax is different [characters 18(1), 20(1)], particularly the lamellae of chaetiger 9 which are low and broad [characters 42(0), 47(0)] in comparison to the elongate lamellae of preceding chaetigers. In addition to the chaetae of chaetiger 9 being mucronate, they are shorter than those occurring on chaetigers 1–8 [character 50(1)]. Lateral abdominal pouches are always present [character 60(1)] and whilst members of several species within the clade possess both anteriorly and posteriorly open pouches [characters 62(1)], only the latter is present in members of other species [character 62(0)]. **Clade m** is further divided into **clades o** and **p** (Fig. 17), separating the species based on arrangement of abdominal hooks (**clade o** with vis-à-vis and **clade p** with vis-à-dos). **Clade m** contains members of the species referred as a "*Magelona mirabilis*" group noted by several previous authors (*Mortimer & Mackie, 2003*; *Clarke et al., 2010*; *Zhou & Mortimer, 2013*). *Clarke et al. (2010)* additionally suggesting the arrangement of abdominal hooks separated members of species within the group.

As noted above, *Magelona jonesi* in the present study is plesiomorphic to all magelonid species in **clade a** (Fig. 17). However, despite possessing tapering chaetae in chaetiger 9 [character 51(0)], similar to preceding chaetigers, in the analysis of *Brasil (2003)* members of *M. jonesi* occurred in **clade VIII.4**, the synapomorphies of which were the presence of mucronate chaetae [character 51(1)] and abdominal hooded hooks in vis-à-dos orientation. Members of this species are in need of redescription and this may help to resolve the

situation in a future study. It is perhaps the orientation of the abdominal hooded hooks which is in most need of examination. Whilst *Hartmann-Schröder (1980)* illustrated the abdominal hooks in a vis-à-dos orientation, she made no mention of that orientation in her description. Looking at her neuropodial figures (*Hartmann-Schröder, 1980*: figs. 116 and 117), several hooks are drawn facing anteriorly. It has been recently highlighted by *Mortimer et al. (2020)* that *M. filiformis*, a species also originally recorded as possessing vis-à-dos hooks and occurring in **clade VIII.3** of *Brasil* (*2003*: fig. 47; Fig. 18) along with *M. pacifica*, has in fact a vis-à-vis orientation. This misinterpretation of orientation can happen in members of species in which the hooks are not split equally between the two groups of a ramus, particularly if the quality of the material is poor. This warrants further clarification for *M. jonesi* and also *M. pacifica*. Unfortunately for the latter species, the abdominal hooks in the type material are mostly all broken (*Mortimer et al., 2012*) making it extremely difficult to discern. This may explain why these three species, *M. jonesi*, *M. filiformis* and *M. pacifica* were part of **clade VIII.2** with members of species possessing mucronate chaetae in *Brasil*'s (*2003*) study.

Clade k in the current results (Fig. 17) contains members of five species: *Magelona dakini*, *M. filiformis*, *M. hartmanae*, *M. hobsonae* and *M. pitelkai*. They share similarities in possessing prostomia which are longer than wide [character 7(0)], with straight [character 4(2)], smooth anterior margins [character 3(0)] formed into rudimentary horns [character 6(0)], and which carry two pairs of prostomial ridges [character 9(1)]. Individuals have similar development of lamellae along the thorax in the notopodia [character 18(0)] but varying development in the neuropodia [character 20(1)]. They possess smooth edged [characters 23(0), 36(0), 44(0)], filiform thoracic lamellae [characters 22(0), 28(0), 35(0), 40(0), 43(0), 48(0)] which are larger in the notopodia [character 31(1)], and basally constricted abdominal lamellae [character 55(1)]. Posteriorly open lateral abdominal pouches are reported [character 60(1)] in all but one species, *M. hobsonae*, however members of this species are only known from short anterior fragments, so this character needs verification. Members of three species, *M. hartmanae*, *M. hobsonae* and *M. pitelkai*, within this clade also possess pennoned chaetae of chaetiger 9 [character 51(2)].

Perhaps worth of noting is **clade n** (Fig. 17), which includes members of species possessing prostomia with distinct horns [character 6(1)], triangular anterior margins [character 4(0)] and two pairs of prostomial ridges [character 9(1)]. Specimens also possess foliaceous notopodial thoracic lamellae [characters 22(1), 35(1), 43(1)] which are larger than the neuropodial lamellae [character 30(1)]. Superior dorsal lobes are present on chaetigers 1–8 [characters 24(1), 37(1)]. The thoracic neuropodial lamellae are filiform [characters 28(0), 40(0),48(0)] with additional postchaetal expansions on chaetigers 8 and 9 [characters 39(1), 49(1)] (N.B. the latter needs verifying in *Magelona anuheone*). Abdominal lamellae are rounded with basal constrictions [character 55(1)]. Whilst nothing more can be concluded from **clade r** (Fig. 17), which contains numerous polytomies, characters such as prostomial width may prove to be important characters in other clades. **Clade t** contains members of four species (Fig. 17), *M. berkeleyi*, *M. lenticulata*, *M. wilsoni* and *Magelona* sp. J of *Uebelacker & Jones (1984)*, which all have wide prostomia

[character 7(2)], rounded bulbous thoracic interparapodial margins [character 52(1)], and lack abdominal lateral pouches [character 60(0)].

### *Magelona* F. Müller, 1858, emended

*Maea* Johnston, 1865 (see below)
*Rhynophylla* Carrington, 1865
*Meredithia* Hernández-Alcántara & Solís-Weiss, 2000
*Octomagelona* Aguirrezabalaga, Ceberio & Fiege, 2001

**Definition.** Pursuant to ICZN (*International Commission on Zoological Nomenclature, 1999*) Article 12.1, the definition of *Magelona* is identical to that provided for the emended family-rank name Magelonidae, provided above (Fig. 17).

**Remarks.** The monotypic name *Magelona* can only be defined as referring to the same phylogenetic hypotheses that define the name Magelonidae, in accordance with ICZN (*International Commission on Zoological Nomenclature, 1999*) Article 12.1's requirement that all names published before 1931 "must be accompanied by a description or a definition of the taxon that it denotes." This means that from a scientific perspective the name *Magelona* is an empirically empty placeholder that is only recognised to satisfy the requirement that species-rank names, as epithets, be binomial.

Hernández-Alcántara & Solís-Weiss (2000) introduced the genus *Meredithia* for members of species possessing large hooded recurved spines on abdominal chaetigers [Table 3: character 69(1)], adding the two species *M. spinifera* and *M. uebelackerae*. However, the nature of the enlarged chaetae differ among members of the two species; members of *M. spinifera* possess enlarged spines [Table 3: character 70(1)], whilst members of *M. uebelackerae* possess unidentate, enlarged re-curved hooks [Table 3: character 70(0)]. The descriptions of specimens with enlarged abdominal chaetae, but which differ greatly in other aspects (*e.g.*, lacking prostomial horns and in the nature of the thoracic lamellae), called the validity of the genus into question (*Mortimer & Mackie, 2003*). Subsequently in the phylogenetic analysis performed by *Brasil (2003)* it was concluded that *Meredithia* is paraphyletic, which is in agreement with the current results. Herein, members of species possessing enlarged chaetae appear in three separate clades (Fig. 17: **clades b**, **c** and **r**), and the two species originally assigned to *Meredithia* are split between **clades c** and **r**. *Magelona falcifera*, *M. magnahamata* and *Magelona* sp. C of *Uebelacker & Jones (1984)* (**clade b**) are all species with members lacking prostomial horns, having similar thoracic development with postchaetal, filiform lamellae in both the noto- and neuropodia but in lacking both thoracic superior dorsal lobes and abdominal lateral pouches, whilst *M. uebelackerae*, *Magelona* sp. E of *Uebelacker & Jones (1984)* (**clade c**) and *M. spinifera* (**clade r**) all possess distinct prostomial horns, have different thoracic development, with variations between the noto- and neuropodia in the thoracic region, but which possess subchaetal neuropodial lamellae. The present results therefore further support the synonymisation of *Meredithia* with *Magelona*.

Results of the phylogenetic inferences by *Brasil* (*2003*: fig. 45) additionally shed doubt on the validity of *Octomagelona*, with *O. bizkaiensis* and an undescribed species from Mexico (not considered in this study) as a paraphyletic group nested within *Magelona* and closely related to species such as *M. variolamellata*, *M. equilamellae*, *Magelona* sp. I of *Uebelacker & Jones (1984)* and *M. polydentata*. This finding is in agreement with the current analysis, indicating that *Octomagelona* cannot be maintained (Fig. 17). Whilst members of *Octomagelona* may differ from other members of *Magelona* in possessing only eight thoracic chaetigers, they share similarities with members of some species in terms of possessing a prostomium which is wider than long [character 7(2)], with rudimentary horns [character 6(0)], but with only one pair of prostomial ridges [character 9(0)], filiform subchaetal thoracic lamellae in the noto- and neuropodia [characters 21(1), 22(0), 26(1), 28(0)], without superior dorsal lobes [characters 24(0), 37(0), 45(0)] and abdominal lamellae without basal constrictions [character 55(0)].

As discussed above *Johnston (1865)* erected the genus *Maea* for the species *M. mirabilis* (*Johnston, 1865*), later referred to *Magelona*. Whilst only one composite phylogenetic hypothesis can be formally recognised from the current results, the type species of the genus, *Maea mirabilis*, appears in both consensus trees (that of the current results and *Brasil, 2003*) in a clade clearly different from the one including the type species of *Magelona*, *M. papillicornis* F. Müller 1858. If, in future studies, **clade m** (a '*Magelona mirabilis*' group) is formally recognised, then this may see the re-establishment of *Maea*.

# ACKNOWLEDGEMENTS

The authors would like to thank Kimberley Mills (Amgueddfa Cymru–National Museum Wales and School of Earth and Environmental Sciences, Cardiff University), Andrew Mackie and James Turner (Amgueddfa Cymru–National Museum Wales) for provision of images used within this publication. Special thanks to the three reviewers, who offered highly constructive comments that greatly improved the quality of the manuscript.

### Funding

The authors received no funding for this work.

### Competing Interests

The authors declare there are no competing interests.

### Author Contributions

- Kate Mortimer and Kirk Fitzhugh conceived and designed the experiments, performed the experiments, analyzed the data, prepared figures and/or tables, authored or reviewed drafts of the paper, and approved the final draft.
- Ana Claudia dos Brasil conceived and designed the experiments, performed the experiments, prepared figures and/or tables, authored or reviewed drafts of the paper, and approved the final draft.

- Paulo Lana conceived and designed the experiments, performed the experiments, authored or reviewed drafts of the paper, and approved the final draft.

### Data Availability

Data are available as a Supplemental File.

### Supplemental Information

Supplemental information for this article can be found online at http://dx.doi.org/10.7717/peerj.11993#supplemental-information.

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
