# Peer review of "Who’s who in Magelona: phylogenetic hypotheses under Magelonidae Cunningham & Ramage, 1888 (Annelida: Polychaeta)"

_PeerJ, doi:10.7717/peerj.11993_

## Round 0.1 · original submission · Minor Revisions

Dear Kate, thanks for submitting this article to PeerJ. Please find attached three 'enthusiastic' reviews, with each offering slightly different suggestions. Most of the suggestions fall into the 'optional' category, so if you choose not to follow them, then a brief justification will be fine. Perhaps the one to give most consideration to is the perceived overly long Materials and methods section, especially the philosophical basis of the study. Placing this at the end of the paper, as an appendix, or referencing prior studies, are both good suggestions. Finally, one reviewer found the figure labelling and captions needed some attention. Improving this aspect would be good as PeerJ has a very wide readership that I suspect appreciates visual material. Thanks again for submitting such a nice ms, and I look forward to your revised version, best Chris

·

Basic reporting

The submitted manuscript is very well written, and structured conforming to all professional standards. The text includes sufficient information both in the Introduction and Material and Methods referencing all relevant prior literature. In this last section, the “Methodological considerations” deserves special mention and gives the work added value. The Results are relevant to the hypothesis and the Discussion includes everything previously published on this field. Literature used is also relevant and well referenced. Figures are all necessary to the content of the article, of very high quality, and appropriately described and labelled. Raw data supplied in a supplemental file are robust.

Experimental design

The submission describes original primary research, clearly defining the research question, which is undoubtedly relevant and meaningful. The investigation has been conducted rigorously and to a high technical standard. Methods are described and widely justified with a lot of information. The research has been conducted in conformity with the prevailing ethical standards in the field.

Validity of the findings

The data on which the conclusions are based are robust. The Conclusions are appropriately stated, connected to the original question investigated, and limited to those supported by the results.

Additional comments

I think the manuscript of Mortimer and colleagues represents an outstanding work, which I believe should be published as is. Although I am not a specialist in Magelonidae and my experience in phylogenetic analysis is limited, as it undoubtedly is in the case of the authors, my long experience in the study of the morphology and taxonomy of polychaetes allows me to evaluate it as a very relevant work.

Reviewer 2 ·

Basic reporting

The manuscript is well written and each section has been carefully constructed. The references are thorough and sufficient historical information is provided. The illustrations are well done and in particular those that depict the critical morphology are excellent.

Experimental design

This study, unlike most phylogenetic papers deals with morphology and not molecular results. The reasons for this are presented up front and although unnecessarily long and drawn out do capture most of current problems and vagaries with current phylogenetic studies. To be honest, it is refreshing to see morphological characters used directly in an analysis instead of mapped on to a tree produced from molecular data. The large number of characters and states is impressive and is what is be expected from biologists who are thoroughly familiar with the morphology and biology of their organisms.

Methods are described in great detail. However, the philosophical basis of abduction and the approach seems overly long for a paper like this and most of the methods and arguments have been previously dealt with by Fitzhugh in numerous papers. I recommend that the large up front section be condensed somewhat or moved to an appendix.

Validity of the findings

The results as presented are well explained and quite clear. I am not so worried about the fact that Magelonidae is a stand-alone taxon. I think every user will understand that the genus Magelona and family Magelonidae have the same meaning and that both terms will be available for use by ecologists and systematists as they prefer. This situation is no different from Apistobranchidae--Apistobranchus or Longosomatidae--Heterospio albeit the latter has a problem with nomenclature and priority. Perhaps the authors could provide a few additional examples such as these.

I would be interested in knowing more about how the various clades relate to currently perceived species groups such as the so-called M. mirabilis group (no frontal horns, etc). I was a little surprised to see that species having the large curved hooks were in different parts of the tree. In those instances where certain characters are somewhat unique, some further discussion would be of interest regarding their placement.

Additional comments

As noted in the other comments, it is refreshing to see a phylogenetic analysis that uses the actual morphology of the animals instead of mapping these characters onto a tree based on molecular results.

I recommend but do not require that the large methods section on Scientific Inquiry be moved to an Appendix. The methods are already long given all the space provided for text and figures documenting the characters being analyzed.

·

Basic reporting

Overall, this is an excellent article. It is very well written and structured according to the standard formats, with a perfect scientific English and the most accurate terminology for the subjects covered. It is very well illustrated, with figures relevant to the topics discussed, and all appropriate raw data are provided. The relevant hypotheses and results are all discussed, making an appropriate unit of publication. If something is missing, is a possible “What next?” final section, pointing possible steps to be followed in the future from the point where the present article ends. This is partially achieved in the section “Morphological characters in future specimen descriptions”, but these statements do not seem to be sufficiently highlighted in the work or abstract, and end somehow lost due to the following sections of the text . In its whole, the work is a pleasant and easy reading for those interested in the subject, and a step forward in the field.

Parts of the article, with a special emphasis in the “Methodological considerations”, can be used as a reference text, being profusely supported with numerous and relevant citations.

Examples are provided to illustrate some criticism discussed in the text, but this is always done in accordance to the professional standards of courtesy and respect expected inter pars, and in the spirits that Science should be open to discussion and avoid dogmatic standings, specially when criticism or the different points of view are perfectly argued and performed in a positively.

The article is abundantly backed with literature references, but I miss the references concerning the original authorship of some taxa cited in the text (see General Comments for the Author, below).

The purpose of the article is clear and well backgrounded, and while it could be argued that the Material and Methods section is probably too long and that it covers issues previously published elsewhere, it is clearly necessary to keep it that way, as the Authors are avoiding some of the actual fashions and practices in phylogenetic studies, and need to explain not only how they did their study, but very specially why they did it that way, avoiding those practices. This section covers aspects of the Philosophy of Science that in spite of being fundamental, seem to have been increasingly overlooked in the way that some tools, analyses and methodologies are applied in the field, and how incorrect and unfounded conclusions are being retrieved from the erroneous application of those same tools, analyses and methodologies to the object of study.

The information included in the section “Methodological Considerations” might have been published elsewhere, but I doubt that many of the potential readers of the present article have actually read it. This is a perfect way to resume all that information in one single article, moreover in the field of the systematics of annelids, and to provide the potential readers with the necessary background to follow the present article and to discuss it. I imagine that the present article might cause some discussion, which is also the driving force that makes Science and our knowledge to progress, so before start arguing, everybody interested should be familiar with the Authors’ arguments.

Experimental design

The research question, the establishment of phylogenetic hypotheses under Magelonidae (Annelida), was perfectly established and defined, and the intention to fulfil this important gap in the knowledge of the group was correctly addressed. It followed a correct experimental design, using the methodology proposed and explained in detail, and established according to the initial question and the required needs to approach its study.

The investigation was performed rigorously and following the highest scientific, technical and ethical standards. Moreover, the authors show a perfect knowledge of the group under investigation and of the methods employed. This not only allows them to use all information available, many of it obtained first-hand and including unpublished data, but also to recognize the flaws and gaps in the understanding of the group, to better establish the necessary methodological approach needed to achieve this important contribution.

While reading, it becomes clear that the present article is the result of a long process of maturation and careful deliberation, integrating the results of a previous approach done almost 20 years ago, in an unpublished PhD thesis. Moreover, a previous draft of the article is mentioned in the text, which supports the idea that the present article is the result of multiple critical readings preceding the present version, and explains in part its strength and cohesion. This becomes perfectly clear in some parts of the text, which are almost flawless. As stated previously, a perfectly matured and deliberated article.

All methods and raw data are explained in detail and provided to the readers, so the results and the consequent conclusions can be not only replicated but also used and integrated in future approaches to the same research question.

Validity of the findings

The present article fulfils and important gap in the knowledge of Magelonidae (Annelida), but its scope goes further beyond the interest of specialists in this particular taxonomic group, or in annelids in general. The Authors present an approach to phylogenetic studies which, is spite of being perfectly established and supported for decades, has been somehow disregarded by a scientific community increasingly in the pursuit of quick and poorly debated publications.

The approach used in the article is far from being new, but it will be seen as such by many unfamiliar with the methodology and the philosophy supporting it, and confidently it will create a much-needed debate in some spheres of Systematics, hopefully in a positive and open-minded way. Some arguments are complex and probably uncommon for many of the current taxonomists, as they were for me, but that does not mean that they should not be considered, fully discussed and integrated in the performed studies, always with base on solid and scientific arguments. I see this possibility of opening a much-desirable debate as one of the big strengths of the present article.

The conclusions achieved are sound and appropriately stated, clearly connected with the original question investigated and the methodology used, and strictly limited to those supported by the results. As commented in a different section of this review, I would have appreciated a final and slightly more speculative section, a “What next?” section debating possible consequences, implications and applications of the obtained results, and pointing possible issues to be investigated in the future, resulting from the conclusions of present article. as stated previously, this is partially achieved in the section “Morphological characters in future specimen descriptions”, but these statements do not seem to be sufficiently highlighted. However, the lack of such hypothetical section does not compromise the interest of the present work.

As a possible flaw, in the Discussion section of the article I miss the inclusion of the genus Maea Johnston, 1865 in the debate about the possible validity of the genera previously assigned to the family Magelonidae, similarly to what is done with the genera Meredithia Hernández-Alcántara & Solís-Weiss, 2000, and Octomagelona Aguirrezabalaga, Ceberio & Fiege, 2001. The type species of the genus, Maea mirabilis Johnston, 1865, appears in both consensus trees in a clade clearly different from the one including the type species of Magelona, M. papillicornis F. Müller 1858. The consequences of such placement should be properly discussed in the text, as it is done with the other two genera synonymized with Magelona.

Finally, another strength of the article that I particularly appreciate as a taxonomist, is that after its reading it becomes clearly evident that the current Systematics needs to go one step further from the basic mechanics of describing taxa or delineating fast phylogenies, to address the epistemological sense of what is being done and, very specially, why it is being done. Only by doing this, Systematics can be considered as a fully established scientific discipline.

Additional comments

First, I would like to thank the Authors and the Editor for the given possibility of reading and reviewing the present work. I have read the manuscript with great interest and pleasure, and I think it is a very good work, very well-structured and thoroughly revised before its submission, which must be acknowledged. I did some comments and corrections directly in the provided pdf, as well as a couple of corrections in table S1, which will be provided to the Authors as attachments to this review. Please, check the attached pdf, and notice that the last three pages correspond to table S1, added to the pdf of the manuscript originally provided.

I would like to highlight some comments and corrections done in the provided pdf, as well as to discuss some other questions that I think the Authors need to consider and/or to address in order to improve the present version of the article. There isn’t a real order of importance of these questions, so they will be provided following the order in which they were noticed while reading the manuscript.

1) In lines 56–57 it is stated:

“Magelonids were given the rank of family by Cunningham & Ramage (1888) but their unusual morphology has often led to difficulties in relating them to other annelid groups. Johnston (1865), puzzled by their peculiar external form, placed them at the end of his catalogue, under the family Maeadae.”

This sentence is not completely correct, as the first author to recognize the uniqueness of magelonids in order to give them the rank of family was Johnston (1865), based on his new genus and new species Maea mirabilis, as Maeadae, a family name that would have priority. McIntosh (1877) included magelonids in the Spionidae, after synonymizing Maea Johnston, 1865 with Magelona Müller, 1858, but Cunningham & Ramage (1888) considered the magelonids to form a distinctive family again, this time based on Magelona, and once more relying on its unique morphology: "We have formed a special family for it, as it cannot be admitted into the Spionidae, with which it is most nearly allied, or into any other family."As stated above, the family name Maeadae would have priority over Magelonidae, but this latter name is to be used, according to article 40.2 of the ICZN. There is also a recommendation (Recommendation 40A), to cite the family-group name maintained under the provisions of Article 40.2.1, with its original author and date, followed by the date of its priority as determined by Article 40.2.1 enclosed in brackets. In case the Authors would like to follow this recommendation, or at least to refer it, the name of the family should be referred as Magelonidae Cunningham & Ramage , 1888 (1865).

2) Note that McIntosh’s 1877 paper "On the structure of Magelona" published in the pages of the Annals and Magazine of Natural History, Series 4, 20(116):147-152 is exactly the same work published previously the same year in the Proceedings of the Royal Society of London, 25(178): 559-564. It would be desirable to cite this latter paper, instead. As an alternative, the Authors can cite both papers, in the same way they are citing the German 1879 paper, and its 1911 English translation in the current article.

3) The references with the original descriptions of some taxa cited in the text are missing in the bibliography. Considering that each one of those names reflects a different hypothesis, please, include their full bibliographic reference. I have gone through the text and tables, and compiled the following list of missing references, but please, revise it:

Aguado & San Martín, 2004
Carrington, 1865
Clarke, Paterson, Florence & Gibbons, 2010
Ehlers, 1874
Ehlers, 1908
Eliason, 1962
Fauvel, 1928
Gallardo, 1968
Glémarec, 1967 [note: 1967, not 1966]
Gravier, 1905
Harmelin, 1964
Hartman, 1944
Hartman, 1960
Hartman & Reish, 1950
Hartmann-Schröder, 1962
Maciolek, 1985
Magalhães, Bailey-Brock & Watling, 2018
Malmgren, 1867
Mohammad, 1970
Monro, 1933
Mortimer & Mackie, 2006
Mortimer, Gil & Fiege, 2011
Müller, 1776 [note: O.F. Müller]
Müller, 1858 [note: F. Müller]
Sars, 1851
Sars, 1853
Shakouri, Mortimer & Dehani, 2017
Wesenberg-Lund, 1949
Zhou & Mortimer, 2013

4) Please, revise the in-text citations, as in some cases they do not seem to follow the journal’s reference format. For instance, for three or fewer authors, to list all author names, for four or more, to abbreviate with ‘first author’ et al. Moreover, multiple references to the same item should be separated with a semicolon (;) and ordered chronologically. According to the same reference format, for two or three authors, the last name should be preceded by “&”, instead of “and”. Please, revise also the whole reference section and edit it according to the journal’s reference format, and include the DOIs of the works cited, if available.

5) Outgroups – The work does not provide much information concerning the included outgroups, such as where the specimens were collected or the vouchers deposited, with the corresponding collection numbers. Please, revise this absence and include the corresponding and relevant information in Table 2.

6) In lines 988–989 it is stated:

“Where specimens were not available for observation, the remaining taxa were chosen based on the level of information and detail in the original descriptions.”

For completeness, and to clearly state which taxa were not considered, please, provide the full list of the taxa excluded from the analyses (as well as the full bibliographic reference containing the original description). This list seems to include the following valid taxa, but please revise and complete it, if necessary:

Magelona agoensis Kitamori, 1967
Magelona americana Hartman, 1965
Magelona capax Hartman, 1965
Magelona capensis Day, 1961
Magelona japonia Okuda, 1937
Magelona japonica koreana Okuda, 1937
Magelona kamala Nateewathana & Hylleberg, 1991
Magelona longicornis Johnston, 1901
Magelona methae Nateewathana & Hylleberg, 1991
Magelona mickiminni Nateewathana & Hylleberg, 1991
Magelona noppi Nateewathana & Hylleberg, 1991
Magelona pectinata Nateewathana & Hylleberg, 1991
Magelona petersenae Nateewathana & Hylleberg, 1991
Magelona pettiboneae lanceolata Jones, 1963
Magelona rosea Moore, 1907
Magelona sachalinenis Buzhinskaja, 1985

7) In lines 992–995 it is stated:

“All ingroup taxa included in the analyses are listed in Table 2, including which specimens were observed and the institution from which they were borrowed. Table 2 also indicates whether observations were based solely on published material.”

Some observations were based on type material, while others solely on published material. In what concerns this last alternative, only in one case it is stated that the observations were based on the redescription of the species, while other redescriptions also available apparently were not used (e.g., the redescription of Magelona pitelkai by Jones (1978)). Was there a reason for not using other redescriptions?

8) In table 3, the characters 26 and 27 are described respectively as “Chaetigers 1‒7 neuropodial lamellae position relative to notochaetae” [character 26] and “Chaetigers 1‒7 neuropodial subchaetal lamellae position relative to notochaetae” [character 27]. It seems to exist a lapsus in both character descriptions, and that they should state “relative to neurochaetae”, instead. Please, confirm if this is correct. In the Results section, revise also the discussion concerning clades a, b and r (lines 1589, 1596 and 1658, respectively).

9) This is probably irrelevant, but in several other publications covering the same taxonomic group, the expression “vis-à-vis” is written using Italics. Please, confirm that Italics is disregarded in the present article on purpose.

10) The distribution of the granular bodies present in the surfaces of the segments seem to be a promising character to be further investigated, while they also raise some questions concerning their possible nature and function, as in some occasions they seem to stain differently under the same stain. For instance, in some cases they stain as dark granular bodies with methyl green (e.g. Fig. 16A) while in other cases and with the same stain, they appear as white granular bodies (e.g. Figs. 16B, 16C). It would be interesting to apply different specific stains to the same specimens, in order to identify the molecular nature of those granular bodies and their possible function.

11) The figures are excellent, but in some cases the labels are difficult to see. Please, revise them and confirm that they are all clearly visible and contrasted in relation to the background. Use appropriate types of fonts and sizes and try using bold. Scales seem to be missing in some figures, please revise them and make sure they are always provided. If necessary, an approximative value can probably be stipulated for the scale. Please, try to standardize the type of scales used as much as possible. In some cases, the labels seem to have been partially cut (e.g., check the label “Scoop-shaped neuropodial lamellae” on figure 9H).

12) The figure captions needed to be standardized through the whole text. I tried to do it, according to the structure that seemed to be used most often by the Authors. Please, revise them and confirm that the same structure is uniformly used through the whole article, or change it accordingly to the Authors preferred structure, and use it consistently.

13) In line 1803 it is stated that in Brasil (2003) Magelona jonesi occurred in clade VIII.3, when it occurred in clade VIII.4.

14) In lines 1853–1855 it is stated, in relation to the genus Magelona F. Müller, 1858:

“The consequence is that ICZN (International Commission on Zoological Nomenclature, 1999)
Article 13.1.1, where a name must be ‘accompanied by a description or definition that states in words characters that are purported to differentiate the taxon,’ cannot be reasonably applied.”

However, Article.13.1.1 applies only to names published after 1930, which is not the case of Magelona F. Müller, 1858. The articles that are relevant here, concerning the validity of the generic name, seem to be articles 11 and 12, and the name Magelona F. Müller, 1858 accomplishes both. Thus, the definition of the name Magelona could be the same as the used for Magelonidae, as both seem to represent the same phylogenetic hypothesis (i.e., Magelonidae as a monogeneric family, with Magelona as the only valid genus).

In the list of synonyms of the genus Magelona the genus Maea Johnston, 1865 is missing and should be added. Consider also the possibility of adding the name Rhynophylla Carrington, 1865 to the list.

15) Directly related with the previous point, why not to consider the definition of Magelona to be the same as Magelonidae? This way, it would not be an empty placeholder, as the family is clearly differentiated and defined. According to the results, and if I understood it correctly the same phylogenetic hypothesis defining Magelonidae, also defines Magelona.

16) There is an interesting issue that seems to have been overlooked. The Authors used the descriptions provided by Uebelacker & Jones (1984) of a series of taxa from the Gulf of Mexico which were described but not formally named by those authors.

Two of these species, Magelona sp. D and Magelona sp. H were later redescribed and formally named as new species by Hernández-Alcántara & Solís-Weiss (2001), as Meredithia spinifera Hernández-Alcántara & Solís-Weiss, 2001 [= Magelona sp. D] and Meredithia uebelackerae Hernández-Alcántara & Solís-Weiss, 2001 [= Magelona sp. H].

While the first species was described and named by Hernández-Alcántara & Solís-Weiss (2001) based on a mixture of material from the Gulf of California and the Gulf of Mexico (type locality = Gulf of California), with the material from the Gulf of Mexico being part of the material used by Uebelacker & Jones (1984) in their description, the second species was described based only in material from the Gulf of Mexico (type locality), all of which originally used by Uebelacker & Jones (1984) in their description. While each one of both duplets represents theoretically the same species hypotheses, they are not necessary grouped together in the two consensus trees provided in the work.

It can be argued that in the case of M. spinifera/M. sp. D, the differences obtained in the consensus tree of Brasil (2003) could be due to the mixture of specimens from two different populations isolated geographically in the description of M. spinifera. However, the differences obtained for the second duplet, M. uebelackerae/M. sp. H, can only be the result of the different ways that the same specimens were interpreted and their characters described by the different authors, and so, they can give an idea of the subjectivity introduced by different authors while studying the same subject, and how and where the species descriptions can be improved.

The analysis of the character matrix of the table S1 shows that there are more differences stated for the duplet M. uebelackerae/M. sp. H (using the same specimens in the two descriptions) than for the duplet M. spinifera/M. sp. D (using a different mixture of specimens in the two descriptions). The analysis of these differences and how they affect the way the duplets are grouped in the two consensus trees, can give an idea of how the interpretation of each one of those characters affects the results, and how and where descriptions should be improved in order to provide more accurate and reliable results. The analysis of the differences in the characters of the duplet M. uebelackerae/M. sp. H could be particularly interesting. Many of the observed differences seem to be related with the description of the morphology of the parapodia, the importance of which is precisely highlighted by the Authors in the present article

17) Finally, and as already observed in a previous section of the review, the genus Maea Johnston, 1865, with M. mirabilis Johnston, 1865 as type species, should also be integrated in the discussion concerning the validity of the different genera included in the family Magelonidae, performed after the subsection “Magelona F. Müller, 1858, emended” (line 1849).

The type species of the genus, Maea mirabilis Johnston, 1865, in placed in both consensus trees in a clade different from the one including the type species of the genus Magelona, M. papillicornis F. Müller 1858. The consequences of such placement should be properly discussed in the text as it is done with the other two genera synonymized with Magelona, especially with Meredithia, which synonymy with Magelona is confirmed.

Thank you!

---

## Round 0.2 · accepted · Accept

Dear Kate, thanks for thoroughly addressing the referee's concerns. Looking forward to seeing this published. Best Chris